# Small-molecule-mediated OGG1 inhibition attenuates pulmonary inflammation and lung fibrosis in a murine lung fibrosis model

L. Tanner [1] ✉, A. B. Single[1], R. K. V. Bhongir [1], M. Heusel [2], T. Mohanty [2], C. A. Q. Karlsson[2], L. Pan [3], C-M. Clausson[4], J. Bergwik[1], K. Wang[3], C. K. Andersson[5], R. M. Oommen[6], J. S. Erjefält[4], J. Malmström [2], O. Wallner[6], I. Boldogh[3], T. Helleday [6,7,8], C. Kalderén[6,7] & A. Egesten [1]

Interstitial lung diseases such as idiopathic pulmonary fibrosis (IPF) are caused by persistent micro-injuries to alveolar epithelial tissues accompanied by aberrant repair processes. IPF is currently treated with pirfenidone and nintedanib, compounds which slow the rate of disease progression but fail to target underlying pathophysiological mechanisms. The DNA repair protein 8-oxoguanine DNA glycosylase-1 (OGG1) has significant roles in the modulation of inflammation and metabolic syndromes. Currently, no pharmaceutical solutions targeting OGG1 have been utilized in the treatment of IPF. In this study we show *Ogg1*-targeting siRNA mitigates bleomycin-induced pulmonary fibrosis in male mice, highlighting OGG1 as a tractable target in lung fibrosis. The small molecule OGG1 inhibitor, TH5487, decreases myofibroblast transition and associated pro-fibrotic gene expressions in fibroblast cells. In addition, TH5487 decreases levels of pro-inflammatory mediators, inflammatory cell infiltration, and lung remodeling in a murine model of bleomycin-induced pulmonary fibrosis conducted in male C57BL6/J mice. OGG1 and SMAD7 interact to induce fibroblast proliferation and differentiation and display roles in fibrotic murine and IPF patient lung tissue. Taken together, these data suggest that TH5487 is a potentially clinically relevant treatment for IPF but further study in human trials is required.

Idiopathic pulmonary fibrosis (IPF) is an irreversible disorder characterized by progressive lung scarring with a median survival time of three years post-diagnosis[1–3]. The disease is associated with increasing cough and dyspnea, affecting approximately 3 million people worldwide, with incidence strongly correlated with increasing age[4]. IPF is

defined on the histological and radiological basis of usual interstitial pneumonia[5]. Fibroblast and myofibroblast overactivation/overstimulation results in extracellular matrix (ECM) deposition in alveolar walls, reducing alveolar spaces[6]. Current IPF therapies focus on inhibiting collagen deposition by blocking myofibroblast activation. These

[1]Respiratory Medicine, Allergology, & Palliative Medicine, Department of Clinical Sciences Lund, Lund University and Skåne University Hospital, SE-221 84, Lund, Sweden. [2]Division of Infection Medicine, Department of Clinical Sciences, Lund University, SE-221 84, Lund, Sweden. [3]Department of Microbiology and Immunology, University of Texas Medical Branch at Galveston, Galveston, TX 77555, USA. [4]Division of Airway Inflammation, Department of Experimental Medical Sciences, Lund University, SE-221 84, Lund, Sweden. [5]Respiratory Cell Biology, Department of Experimental Medical Sciences Lund, Lund University, SE-221 84, Lund, Sweden. [6]Science for Life Laboratory, Department of Oncology-Pathology, Karolinska Institutet, SE-171 76, Stockholm, Sweden. [7]Oxcia AB, Norrbackagatan 70C, SE-113 34, Stockholm, Sweden. [8]Weston Park Cancer Centre, Department of Oncology and Metabolism, University of Sheffield, Sheffield S10 2RX, UK. ✉e-mail: lloyd.tanner@med.lu.se

approaches have shown limited success in achieving overall IPF resolution, highlighting the need for novel therapeutic strategies[4,7].

IPF tissues produce increased levels of reactive oxygen species (ROS), resulting in DNA damage and the upregulation of fibrotic-related pathways, ultimately leading to lung architecture collapse[8,9]. The DNA base guanine is particularly prone to oxidation, forming 7,8-dihydro-8-oxoguanine (8-oxoG). 8-oxoG is recognized by the enzyme 8-oxoguanine DNA glycosylase 1 (OGG1), whereby OGG1 binding initiates DNA base excision repair (BER) pathways[10–14].

Following lung injury, fibroblasts transition to myofibroblasts through stimulatory factors such as TGF-β1, which further induce the production of fibrotic markers including α-smooth muscle actin (α-SMA), collagen, and fibronectin[15,16]. The fibroblast to myofibroblast transition (FMT) and migratory activities are well-established hallmarks of IPF[17,18]. Interestingly, siRNA-mediated *Ogg1* knockdown in murine embryonic fibroblast cells revealed decreased levels of tissue-associated α-SMA, a well-defined marker of FMT[19]. Significant focus has been placed on OGG1's role during oxidative stress, not only as an initiator of DNA BER but also as a key to gene expression in various inflammatory processes and metabolic syndromes[20–28]. OGG1-mediated DNA bending due to nucleotide extrusion from double-stranded DNA creates specific DNA structural changes that facilitate transcription factor recognition and binding to consensus motifs[10,29,30]. In addition, OGG1-mediated base excision with or without incision of DNA is also linked to modulation of gene expression including those promoters with potential G-quadruplex (often present in promoter of inflammatory and fibrotic genes)[24,31–35]. These OGG1-driven processes play an extensively important role in host oxidative stress, inflammation, and metabolic homeostasis, ultimately controlling diseases processes[24]. OGG1's implication in fibrogenesis, combined with its role in inflammation, highlights this enzyme as a potential therapeutic target for IPF treatment[22,36].

In this regard, a small molecule inhibitor of OGG1-DNA interactions, TH5487, was shown to decrease in vivo levels of pro-inflammatory gene expression[22]. More specifically, TH5487 lowered DNA occupancy of OGG1 at guanine-rich promoter regions, subsequently impeding tumor necrosis factor-α (TNFα)-induced OGG1-DNA interactions, reducing immune cell recruitment[22]. In our study, a bleomycin-challenged murine model was used to assess TH5487's anti-fibrotic efficacy. This model reproduces several phenotypic features of human IPF, including peripheral alveolar septal thickening, dysregulated cytokine production, and immune cell influx resulting in fibrosis[37,38]. Herein we demonstrate that TH5487 inhibits migratory and proliferative capacities of lung-derived epithelial and fibroblast cells in vitro. In addition, administration of *Ogg1*-targetting small interfering RNA (siRNA) or TH5487, mitigates distinct bleomycin-induced pulmonary immune cell recruitment and fibrotic lung damage. Finally, we confirm the involvement of base/nucleotide excision repair and Mothers against decapentaplegic homolog (SMAD) family of proteins, with OGG1 and several SMAD proteins shown to drive fibroblast proliferation and differentiation[36,39]. Taken together, these data strongly suggest that TH5487 mitigates numerous features of IPF progression and should be tested further in preclinical models.

## Results

### TH5487 diminished migratory capacity of lung-resident cells
The migration of fibroblasts into the lung injury site is a key step in the pathogenesis of pulmonary fibrosis, with fibroblasts contributing the greatest generation of ECM components during disease[40]. Primary human lung fibroblasts (pHLF) and murine-derived fibroblast (MF) cells (both wild-type and *Ogg1*[−/−]) were used in transwell assays. The cells were additionally treated with TH5487 (10 μM), vehicle, nintedanib (10 μM), or dexamethasone (DEX; 10 μM) as comparator drugs. This assay showed the fibroblast transition and migration were significantly decreased

($P = 0.0001$) *following* TH5487/TGF-β1 addition, while indicating the expected pro-migratory effects of TGF-β1 stimulation at 2 ng/mL (Fig. 1a and Supp. Fig. 22). Interestingly, *Ogg1*[−/−] MF and TH5487-treated primary human lung fibroblast cells displayed less cell migration comparable to those treated with nintedanib as shown by ANOVA testing with Dunnet's post hoc analysis ($P = 0.0001$). TGF-β1-driven fibrosis results in the transition of fibroblasts to myofibroblasts, producing pro-fibrotic proteins (Fig. 1b). OGG1 knockdown using transiently transfected PHLF cells resulted in siOGG1 cells having reduced migration ($P < 0.0001$) in a transwell assay compared to scramble control siRNA-treated cells (Fig. 1c). Furthermore, *Ogg1*[−/−] MFs lacked actin polymerization to F-actin (phalloidin staining), in contrast to wild-type cells, which also expressed α-smooth muscle actin (α-SMA) a marker of fibrogenic cells including myofibroblasts (Fig. 1d). TH5487 treatment or genetic ablation of *Ogg1* reduced the production of collagen, fibronectin, vimentin, and α-smooth muscle actin, as measured by Western blot (WB), and qRT-PCR analysis (Fig. 1e–h; Supp. Fig. 1a–g). These results indicate that TH5487 is capable of inhibiting TGF-β1-mediated migratory effects of lung-derived cells, including both human- and mouse-derived fibroblasts (Supp. Fig. 2a, b), potentially limiting the severity of IPF disease.

### *Ogg1* RNA interference protects against fibrotic lung damage
Based on the effects of *Ogg1* knock down and *Ogg1/OGG1* inhibition in cultured cells, we tested whether administration of small interfering RNA (siRNA) targeting *Ogg1* was effective in mitigating in vivo experimental pulmonary fibrotic processes. Following bleomycin administration to male C57BL6/J mice and subsequent fibrotic lung damage as denoted by mouse weight loss at day 14, *Ogg1*-targeting siRNA was administered intratracheally (Fig. 2a). Non-targeting (NT) siRNAs were included as negative controls for this study. To assess potential pharmacological interventions using small *Ogg1*-targeting molecules, TH5487 ((i.p.) administered once daily), and the clinically utilized nintedanib and pirfenidone were also included. *Ogg1*-targeting in this context, using either siRNA or TH5487, appeared to hinder the progression of weight loss (Fig. 2b) and subsequent lung damage, with doses of siRNA and TH5487 well-tolerated within this model (Fig. 2c). Flow cytometric analyses further elucidated significantly decreased neutrophil ($P < 0.0001$), alveolar macrophage ($P < 0.0001$), and inflammatory macrophage ($P < 0.0001$) populations in bronchoalveolar fluid (BALF) of TH5487-treated mice as compared to the non-targeting siRNA/bleomycin control samples (Fig. 2d). Furthermore, immune cell recruitment was decreased in TH5487-treated mice compared to those treated with nintedanib and pirfenidone. Immune cell recruitment was reflected by levels of cytokines, with reductions in IL-4 (BALF: $P = 0.0006$ and lung: $P < 0.0001$), IL-6 (BALF: $P = 0.0042$ and lung: $P < 0.0001$), G-CSF (BALF: $P = 0.0145$ and lung: $P < 0.0001$), and MCP-1 (BALF: $P = 0.0197$ and lung: $P < 0.0001$) across BALF and lung tissue samples, indicative of reduced immune cell recruitment (Fig. 2e and Supp. Figs. 2–5).

Decreased OGG1 protein levels were seen following the use of either *Ogg1*-targetting siRNA or TH5487 (i.p.) compared to bleomycin/NT siRNA controls (Fig. 2f). Furthermore, collagen deposition was significantly reduced by TH5487 treatment compared to bleomycin control animals ($P < 0.0001$) to levels similar to nintedanib and pirfenidone-treated samples, with less of an effect reported for the *Ogg1*-targetting siRNA group (Fig. 2g). Interestingly, histological analyses of *Ogg1*-targetting siRNA and TH5487-treated murine lungs revealed reduced levels of collagen deposition and decreased structural remodeling of the lungs (Supp. Figs. 6 and 7). Given the initial success of this approach, we aimed to further elucidate the mechanisms by which TH5487 ameliorated pulmonary fibrosis.

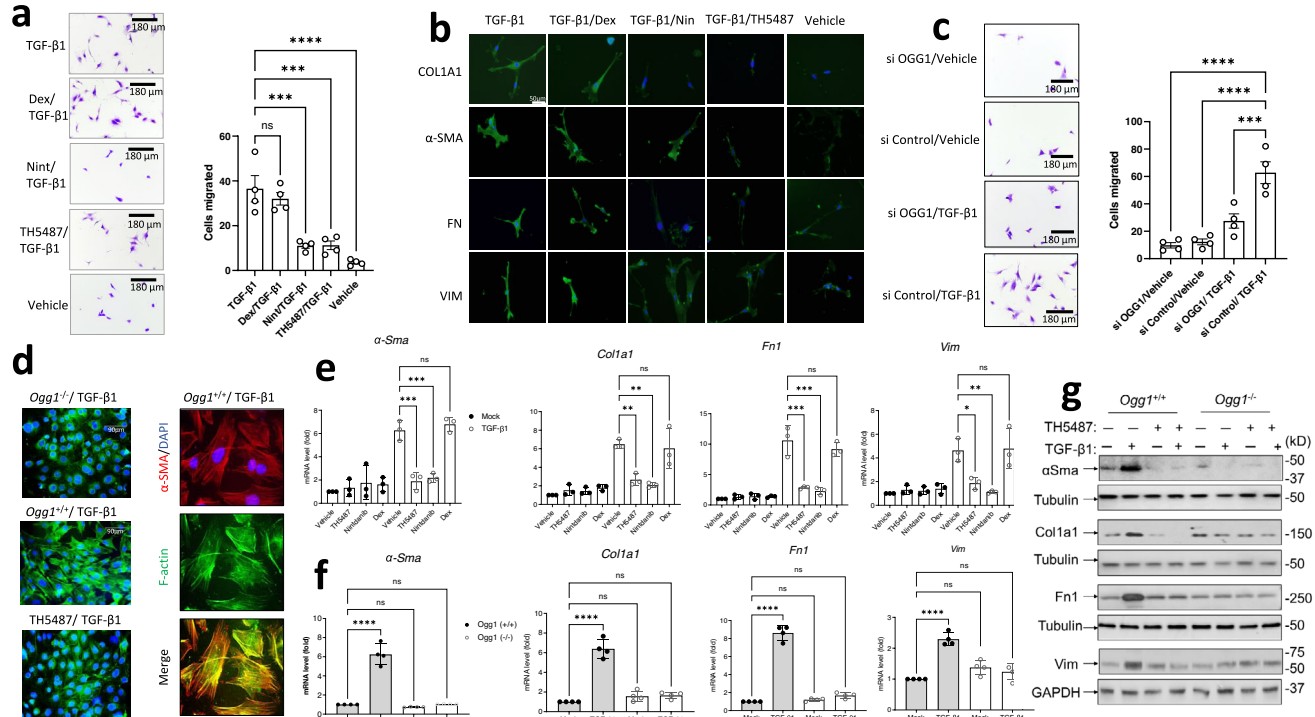

**Fig. 1 | OGG1-dependent cell migration and fibrotic gene expression. a** Migration of human primary lung fibroblasts (pHLF) post-TGF-β1 induction for 24 h, with significantly more myofibroblast cells appearing than in mock- and inhibitor(s)-treated wells. Data were analyzed using a one-way ANOVA followed by a Dunnett's post hoc test unless otherwise specified: TGF-β1 vs TGF-β1/TH5487 (P = 0.0001); TGF-β1 vs TGF-β1/Dex (P = 0.6954); TGF-β1 vs TGF-β1/Nin (P = 0.0001); TGF-β1 vs TGF-β1/Veh (P < 0.0001). ns: not significant. Source data are provided as Source data file. Data are representative of 4 independent experiments containing 3 biological replicates (scale bar = 180 μm). Data are presented as the mean ± standard error of the mean (**a, c, e, f**). **b** Immunostaining of pHLF cells (green) following 24 h of TGF-β1 treatment, with TH5487 treatment displaying visually reduced levels of collagen (COL1A1), fibronectin (FN), vimentin (VIM), α-smooth muscle actin (α-SMA). Scale bar = 50 μm. Results shown from 3 independent experiments. **c** Migration of pHLF post-TGF-β1 induction for 24 h, and siRNA transfection targeting *OGG1* or scrambled sequence (control). Data were analyzed using a one-way ANOVA followed by a Dunnett's post hoc test: si Control/TGF-β1 vs si OGG1/TGF-β1 (P = 0.0008); si Control/TGF-β1 vs si OGG1/vehicle and si Control/TGF-β1 vs si

Control/vehicle (P < 0.0001). Data are representative of 4 independent experiments containing 3 biological replicates (scale bar = 180 μm). Data are presented as the mean ± standard error of the mean. **d** Stress fiber formation was seen in response to TGF-β1 in *Ogg1*$^{+/+}$, but not *Ogg1*$^{-/-}$ MF cells (left panel, F-actin). Expression and colocalization of α-SMA (red) with F-actin (green) are shown (right panels, scale bar = 90 μm). Results shown from 3 independent experiments. **e** The effects of TH5487 on transcription of *α-Sma*, *Fn1*, *Vim*, and *Col1A1* in TGF-β1-stimulated MF cells as determined by qRT-PCR. TH5487 (10 μM), significantly decreased mRNA levels of all genes with compiled data representative of 3 independent experiments. Data were analyzed by one-way ANOVA followed by a Dunnett's post hoc test. **f** TGF-β1-induced expression of αSma, Col1a1, Fn1, and Vim was decreased in *Ogg1*$^{-/-}$ but not in *Ogg1*$^{+/+}$ MF cells. Data representative of 3 independent experiments. **g** Immunoblot analysis of TGF-β1-stimulated pHLF cells showing decreased levels of α-SMA, COL1A1, FN1, and VIM levels in OGG1-depleted cells by siRNA. TGF-β1, 2 ng/mL; TH5487, 10 μM; Nintedanib (Nin), 10 μM; Dexamethasone (Dex), 10 μM. Results shown from 3 independent experiments.

## Intraperitoneal TH5487 maintained murine bodyweight following bleomycin administration

We next investigated TH5487-mediated *Ogg1* inhibition in vivo using a murine model of pulmonary fibrosis. Male C57BL/6J mice received a once-off intratracheal administration (Supp. Fig. 26) of bleomycin (2.5 U/kg) and were dosed (i.p.) with TH5487 or dexamethasone 1 h post-bleomycin administration, followed by additional dosing once per day, in 5-day intervals, over a total period of 21 days (Fig. 3a). Body weights were recorded as a proxy for overall health status (Fig. 3b). Mice in the bleomycin/vehicle group lost 17.98 ± 3.12% total bodyweight, while mice in remaining groups either maintained their bodyweight or gained significant weight compared to the bleomycin/vehicle group over the experimental period. Representative lungs are shown in Fig. 3c alongside with total lung weight from each animal, where significant differences can be seen. Bleomycin-administered mice showed visual lung damage and significantly increased total lung weights compared to all other groups, but not in mice that received TH5487 (P < 0.0001) or the clinically used drugs, nintedanib (P = 0.0015) and pirfenidone (P < 0.0001), respectively.

## TH5487 treatment reduced cytokine levels and lung damage

The effect of TH5487 treatment on inflammatory cytokine levels was investigated using a 23-cytokine multiplex assay (Supp. Figs. 8–10). Cytokines were assessed in lung tissue homogenate, BALF, and plasma (Fig. 4a–c). TH5487 treatment significantly reduced the levels of several bleomycin-induced inflammatory cytokines, including those in the BALF such as IL-9 (P = 0.0478), eotaxin (P < 0.0001), MIP-1α (P = 0.0035), MIP-1β (P = 0.0010) and IL-5 (P = 0.0004), and KC (P < 0.0001), while cytokines from plasma displaying a reduction include IL-5 (P = 0.0064), MCP-1 (P = 0.0216), IL-6 (P = < 0.0001), and eotaxin (P < 0.0001). Furthermore, similar reductions in the cytokine levels in the lung homogenate were seen following TH5487 administration, in particular, levels of MIP-1α (P = 0.0007), KC (P = 0.0092), and IL-6 (P = 0.0047) and IL-5 (P = 0.0249). In addition, TGF-β1 levels were measured in the BALF, plasma, and lung homogenate of murine samples (Fig. 4d). TH5487-treated samples displayed significantly less TGF-β1 than in vehicle/bleomycin samples (BALF and plasma: P < 0.0001; lung: P = 0.0022). Plasma leakage into the BALF as a proxy for lung damage was assessed by BCA assay (Fig. 4e). TH5487-treated mice displayed significantly lower albumin levels compared to vehicle/

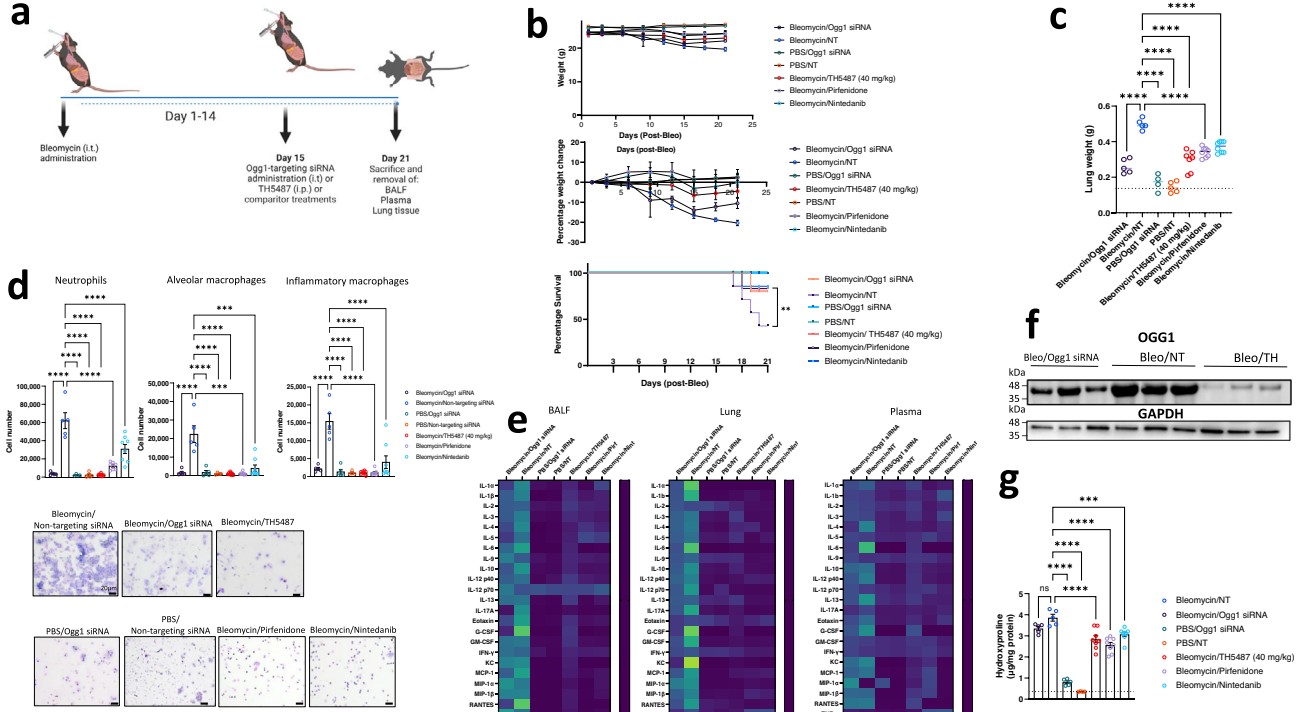

**Fig. 2 | In vivo bleomycin model highlighting OGG1 as a therapeutic target.**
**a** Bleomycin (Bleo, 2.5 U/kg) was intratracheally-administered to C57Bl/6J mice, with mice left to develop fibrosis over a period of 21 days. Subsequent *Ogg1*-targetting or non-targeting siRNA (NT) administration occurred at day 14, with the inclusion of TH5487 (TH; 40 mg/kg; i.p.; daily for 5 days), Nin (60 mg/kg; p.o.; 5 daily doses), and pirfenidone (300 mg/kg; p.o.; daily for 5 days) treatment groups. Data are presented as means ± SEM (**b**, **d**, **g**). ns: not significant. **b** Murine weights, survival, and percentage total weight loss for the Bleo administered groups displayed similar trends until day 14, thereafter the *Ogg1* siRNA and TH5487-treated groups displayed maintenance of weight until the end of the study (Bleo/ Ogg1 siRNA *n* = 5, Bleo/NT *n* = 5, Phosphate buffered saline (PBS)/Ogg1 siRNA *n* = 5, PBS/NT *n* = 5, Bleo/TH5487 *n* = 9, Bleo/Pirfenidone *n* = 8, Bleo/Nintedanib *n* = 8). **c** Murine total lung weight. Significant differences were seen between TH5487 and *Ogg1* siRNA-treated groups compared to the NT siRNA group (*P* < 0.0001). **d** Flow cytometry conducted on murine bronchoalveolar lavage fluid (BALF) indicates a

significant decrease in inflammatory cell recruitment following *Ogg1* siRNA or TH5487 treatment (*P* < 0.0001; (Bleo/Ogg1 siRNA *n* = 5, Bleo/NT *n* = 5, PBS/ Ogg1 siRNA *n* = 4, PBS/NT *n* = 5, Bleo/TH5487 *n* = 9, Bleo/Pirfenidone *n* = 8, Bleo/ Nintedanib *n* = 8)). Murine BALF samples showing cell morphology for each treatment condition. Scale bar = 20 μm. **e** Cytokine expression (day 21) following Bleo administration and subsequent treatment (mean-normalized values shown, with yellow indicating high values and blue indicating low values). **f** Lung homogenate samples were analyzed by SDS-PAGE (*n* = 3; 1 from each independent experiment), followed by immunoblotting using an antibody specific to OGG1. Results shown from 3 independent experiments. **g** Hydroxyproline levels display significant decreases in both siRNA and drug (TH5487, Nin, pirfenidone)-treated groups, indicative of reduced levels of collagen in these lungs (Bleo/Ogg1 siRNA *n* = 5, Bleo/ NT *n* = 5, PBS/Ogg1 siRNA *n* = 5, PBS/NT *n* = 5, Bleo/TH5487 *n* = 9, Bleo/Pirfenidone *n* = 8, Bleo/Nintedanib *n* = 8). Source data are provided as a Source data file. Elements of this figure were created with BioRender.com.

bleomycin mice (*P* < 0.0001). Further, lactate dehydrogenase (LDH) was assessed as a marker for tissue damage in the lungs, with TH5487 (*P* = 0.0022) and dexamethasone (*P* = 0.0018) significantly decreasing these levels (Fig. 4f). In addition, hydroxyproline content (Fig. 4g) was assessed in murine lung tissue and found to be significantly lowered following TH5487 treatment (*P* < 0.0001) but not following dexamethasone treatment (*P* = 0.1236), indicative of the distinct anti-fibrotic nature of TH5487 versus the anti-inflammatory comparator. This was further supported by fibrosis-specific PCR arrays conducted on pirfenidone, TH5487, and nintedanib-treated samples (Fig. 4h–j). More specifically, TH5487 displayed greater reductions in the expression of key fibrotic genes, TIMP metallopeptidase inhibitor 1 (*Timp-1*) and *Smad2* in comparison to both nintedanib and pirfenidone. While pirfenidone significantly reduced (<Log2 fold) a greater number of genes compared to TH5487 and nintedanib, the comparison between nintedanib and TH5487 revealed additional significant reductions in Gremlin-1, *Tgf*-β1, *Tnf*, and other genes associated with pro-fibrotic function (Supp. Figs. 23–25).

## TH5487 treatment reduced immune cell infiltration into the airways in vivo
We next investigated the effects of TH5487 treatment on immune cell infiltration into the airways by performing flow cytometry on BALF

obtained from the in vivo bleomycin studies. Decreases in neutrophil count (*P* < 0.0001) and inflammatory macrophages (*P* = 0.0450) were seen following TH5487 administration compared to vehicle/bleomycin samples (Fig. 5a, b). Giemsa-Wright-stained BALF samples showed bleomycin-treated macrophages were enlarged, displaying an inflammatory phenotype not seen in mice treated with TH5487 (Fig. 5c). The inflammatory phenotype of these macrophages was confirmed using CD206/F4/80 immunofluorescence, with TH5487 treatment significantly decreasing CD206 staining in cells obtained from both murine BALF (Fig. 5d) and lung tissue (Supp. Fig. 11), with decreased neutrophil-specific staining in murine BALF following TH5487 treatment (Fig. 5e).

## Mass spectrometry reveals fibrotic-related protein decreases following TH5487 treatment
To further elucidate the mechanisms by which TH5487 decreases bleomycin-induced fibrosis, LC-MS/MS DIA analyses were conducted, comparing proteomic profiles in both, lung tissue and BALF between treatment groups. Animals within groups were considered biological replicates of the respective treatment condition. Protein isolates per animal and compartment were analyzed via an optimized tissue label-free proteomic workflow based on nanoflow reversed phase chromatography and deep data-independent mass spectrometry. To support

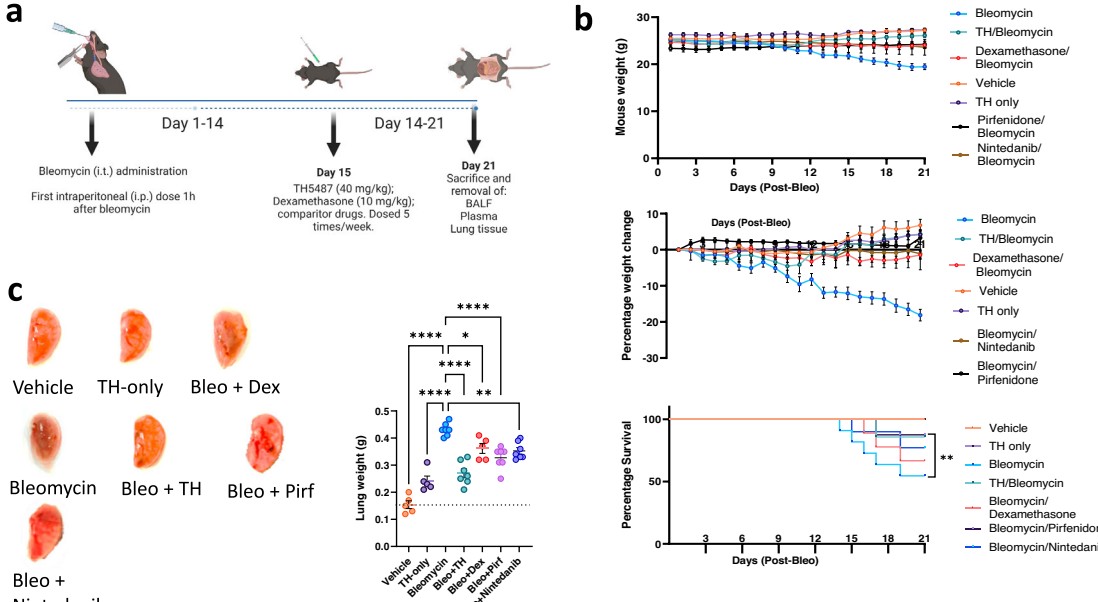

**Fig. 3 | TH5487 murine dosing strategy and weights. a** Mice received intratracheally-administered bleomycin (Bleo; 2.5 U/kg) and were subsequently dosed intraperitoneally (i.p.) with TH5487 (TH), nintedanib (Nin), pirfenidone (Pirf), or dexamethasone (Dex) 1 h post-Bleo administration. Drug treatment occurred five times per week, over the course of 21 days, followed by euthanasia and removal of BALF, plasma, and lung tissues (Bleo $n = 15$, Bleo/TH $n = 14$, Bleo/Dex $n = 9$, Vehicle $n = 9$, TH only $n = 9$, Bleo/Pirf $n = 7$, Bleo/Nin $n = 7$). Data are presented as means ± SEM (**b, c**). Data were analyzed using a one-way ANOVA followed by a Dunnett's post hoc test unless otherwise specified. **b** Mice receiving Bleo showed

weight loss up until day 10, where after those dosed with TH5847 picked up significant amounts of weight compared to the vehicle/bleomycin group. **c** Representative images of murine lungs (right lobes) removed after 21 days, with lung weights shown alongside, Bleomycin vs Bleo+TH ($P < 0.0001$), Bleo vs Bleo +Pirf ($P < 0.0001$), Bleo vs Bleo+Nintedanib ($P = 0.0056$) (Bleo $n = 8$, Bleo/TH $n = 8$, Bleo/Dex $n = 5$, Vehicle $n = 5$, TH only $n = 5$, Bleo/Pirf $n = 7$, Bleo/Nin $n = 7$). Source data are provided as a Source data file. Elements of this figure were created with BioRender.com.

unbiased protein identification, a spectral-library free approach based on deep learning based spectrum predictions was employed[41]. Proteomic patterns were compared in all 5 treatment conditions in both BALF and lung tissue, comparing treatment schemes bleomycin combined with TH5487 (BTH), dexamethasone, TH5487 alone (TH), and vehicle control (V) relative to bleomycin alone (B) as control (Fig. 6a). A total of 9872 proteins were identified across lung and BALF samples, with 5629 proteins observed exclusively from lung (57%), 495 proteins identified exclusively in BALF and 3748 proteins observed across both compartments (Fig. 6b). On average, $7682 ± 1074$ protein groups were identified per DIA-MS run of lung homogenates and $3024 ± 1083$ of bronchoalveolar lavage samples, forming a comprehensive basis for the evaluation of proteomic alterations incurred by the investigated treatment schemes (Fig. 6b). Figure 6c provides an overview of protein intensities observed across the lung and BALF compartments across the cohort. Emphasis was placed on elucidating pathways associated with murine lung environment. Significant proteome alterations were determined from normalized precursor-level abundances via t-statistics, summary to protein level and multiple hypothesis testing correction (Significance criteria; fold-change ≥ 2, Benjamini–Hochberg-corrected protein-level $P$-value ≤ 0.05 (Fig. 6d, Supp. Fig. 12, Supp. reports 01_diffTests*.pdf & "Methods" for details). Pathway analysis of the regulated protein sets using StringDB and PantherDB revealed a number of enriched pathways (Supp. Figs. 13–17). Particular focus was placed on processes related to fibrotic changes as compared to the bleomycin control.

Interestingly, lung proteins related to the gene-ontology (GO) terms collagen biosynthesis process (GO:0032964), wound healing involved in inflammatory response (GO:0002246), endothelial cell proliferation (GO:0001935), collagen metabolic process (GO:0032963), response to fibroblast growth factor (GO:0071774), regulation of cytokine production (GO:0001818), wound healing

(GO:0042060), response to wounding (GO:0009611) were decreased following TH5487/bleomycin treatment, as compared to the bleomycin alone treatment condition (Fig. 6d).

Furthermore, proteins of interest that appeared downregulated following TH5487 treatment included tenascin-C (TNC; $P = 0.0374$), carbamoyl phosphate synthetase 1 (CPS1; $P = 0.03978$), matrix metalloproteinase 19 (MMP19; $P = 0.0402$), tissue inhibitor of metalloproteinase 1 (TIMP1; $P$ value = 0.0460), Janus kinase 3 (JAK3; $P = 0.0301$), cAMP-dependent protein kinase catalytic subunit (PRKX; $P = 0.0152$), lysyl oxidase homolog 2 (LOXL2; $P$ value = 0.0208), whereas these changes did not attain statistical significance in light of strict multiple testing correction and limited statistical power constrained by cohort size and biological and technical variability.

Other proteins involved with fibrotic pathways which experienced Log two-fold decreases following TH5487 treatment as compared to bleomycin control samples included hyaluronidase 1 (HYAL1; Log2FC = 1.691), secreted frizzled-related protein 1 (SFRP1; 3.142), cathepsin S (CTSS; 1.830), A disintegrin and metalloproteinase with thrombospondin motifs 15 (ADAMTS15; 3.135), C-type lectin domain family 10 member A (CLEC10A; 2.132); heme oxygenase 2 (HMOX2; 2.991), collagen 1A1 (COL1A1; 2.241), arginase 1 (ARG1; 2.266), and other proteins (Fig. 6f).

In addition, to explain the mechanism by which TH5487 reduces fibrotic damage, the SMAD family of proteins were compared to matching bleomycin control proteins (Fig. 6f). SMAD 1 and 5 were increased in response to TH5487 treatment (0.3082 and 0.9730, respectively) with SMAD 2/3 and 4 decreased following treatment (0.2830 and 0.4000, respectively). Next, the effect of TH5487 was further elucidated by analyzing proteins known to contribute to DNA BER (Fig. 6f, lower panel). This is shown by Log two-fold increases of greater than 1.5 in samples from mice treated with bleomycin alone versus mice treated with bleomycin/TH5487 for cyclin-H (CCNH), CDK-

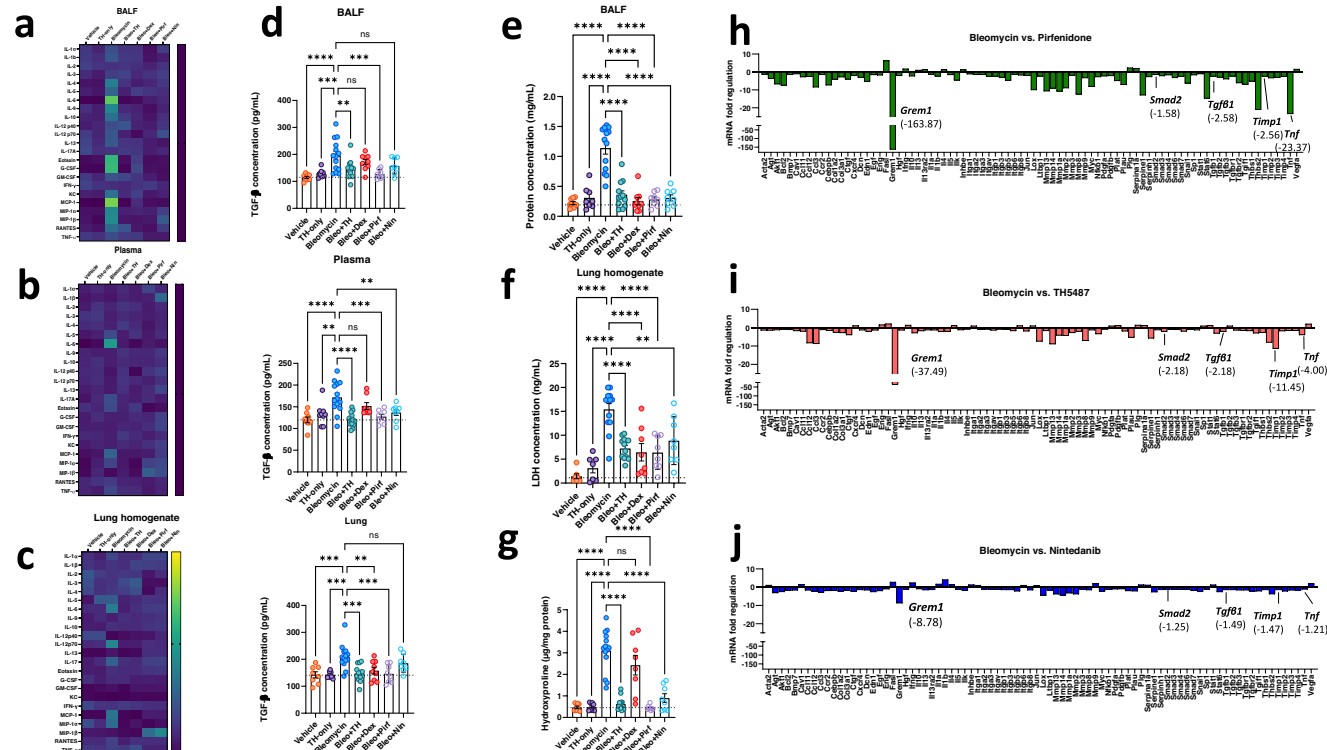

**Fig. 4 | Significantly decreased cytokine levels following TH5487 administration, with accompanying reduction in markers of lung damage.** Heatmaps (a–c) showing the differences in cytokine levels, as measured by multiplex assay, in murine bronchoalveolar lavage fluid (BALF), plasma, and lung homogenate (mean-normalized values shown, with yellow indicating high values and blue indicating low values). Cytokine values were compared to the vehicle/Bleo group using a one-way ANOVA (*$P < 0.05$; **$P < 0.01$; ***$P < 0.005$; ****$P < 0.0001$). ns: not significant. TGF-β1 ELISA (d) conducted on murine BALF, plasma, and lung homogenate revealed significantly decreased TGF-β1 levels in all three murine sample types with values compared to the vehicle/Bleo group using a one-way ANOVA: Bleo vs Bleo +TH (BALF: $P = 0.0022$; plasma: $P < 0.0001$, Lung homogenate: $P = 0.0001$). Data

are presented as means ± SEM (d–g). Murine albumin content (e) and Lactate dehydrogenase (LDH) (f) were measured in BALF samples as markers for lung damage and plasma leakage, with TH5487 treatment significantly decreasing both albumin content ($P < 0.0001$) and LDH levels in the BALF ($P = 0.0022$) compared to the vehicle/Bleo group. **g** Murine lung collagen content was measured using a hydroxyproline assay with TH5487 significantly reducing collagen levels ($P < 0.0001$) compared to Bleo control lungs (d–g: (Bleo $n = 14$, Bleo/TH $n = 13$, Bleo/Dex $n = 8$, Vehicle $n = 8$, TH only $n = 8$, Bleo/Pirf $n = 8$, Bleo/Nin $n = 8$). **h–j** qRT-PCR arrays specific for mouse fibrotic genes were used to analyze murine lung samples ($n = 5$ pooled samples) following treatment using TH5487 (TH), nintedanib (Nin), and pirfenidone (Pirf). Source data are provided as a Source data file.

activating kinase assembly factor MAT1 (MNAT1), DNA repair protein complementing XP-C cells (XPC), X-ray repair cross-complementing protein 1 (XRCC1), DNA polymerase delta subunit 2/3 (POLD 2/3), DNA ligase 1 (Lig1), and G/T mismatch-specific thymine DNA glycosylase (TDG) indicating upregulation of back-up repair proteins.

To further elucidate the differences seen in BALF and lung tissues, cluster analysis was conducted using the gap statistics method, identifying 8 discrete clusters (Supp. Fig. 17). Using this analysis, proteins which were associated to the assigned clusters were assessed using the StringDB and PantherDB. These analyses revealed enriched reactome pathways in the lung samples associated with cluster 1 (IL-6 signaling), 3 (DNA repair), 4 (degradation of extracellular matrix), and 5 (TCA cycle and mitochondrial respiratory and electron transport), indicative of a significant role for the lung proteome in this model.

### TH5487 treatment decreased bleomycin-induced lung damage
Next, TH5487's impact on bleomycin-induced lung damage was investigated using histological analysis of whole lung sections obtained from in vivo studies (Supp. Figs. 18 and 19). Bleomycin-treated control mice displayed significant lung damage when compared to saline-treated controls (Fig. 7a, b). Bleomycin-treated mice that received bleomycin/TH5487 treatment showed reduced lung damage ($P < 0.0001$), with lesser degrees of alveolar structural remodeling and less immune cell influx as indicated by the H&E stain (Fig. 7a). Picrosirius red staining revealed a reduced level of collagen

deposition in bleomycin/TH5487-treated versus control-treated ($P < 0.0001$) mice (Fig. 7b, Supp. Fig. 19). Positive pixel analysis was used to quantify lung damage and showed significantly less H&E and picrosirius red staining in the TH5487-administered lungs, with no significant decreases in either staining reported for dexamethasone-treated lungs (Fig. 7a, b). Further, scanning electron microscopy analysis (SEM) revealed collagen deposition surrounding the alveolar walls of bleomycin control mice was reduced in response to TH5487 treatment (Fig. 7c). Immunofluorescent staining of murine lung sections revealed decreased myeloperoxidase (MPO), fibronectin, COL1A1, and OGG1 staining compared to the vehicle bleomycin control group (Fig. 7d). To further support the involvement of OGG1 in fibrotic-related lung damage, co-staining was carried out on murine lung sections, revealing increased COL1A1/OGG1 fluorescence in similar lung areas (Fig. 7e).

### OGG1 and SMAD involvement in IPF patients
Given SMAD protein concentration measured in the bleomycin lung samples via proteomic mass spectrometry, confirmatory OGG1/SMAD7 immunofluorescence was measured in murine lung samples. Lungs treated with bleomycin appeared to co-stain in similar areas with both OGG1 and SMAD7, indicating potential interaction, with significantly reduced signal in the TH5487-treated samples (Fig. 8a). Confirmatory SDS-PAGE displayed lower levels of OGG1 and SMAD7 following TH5487 treatment ($P < 0.0001$) as compared to untreated samples (Fig. 8b).

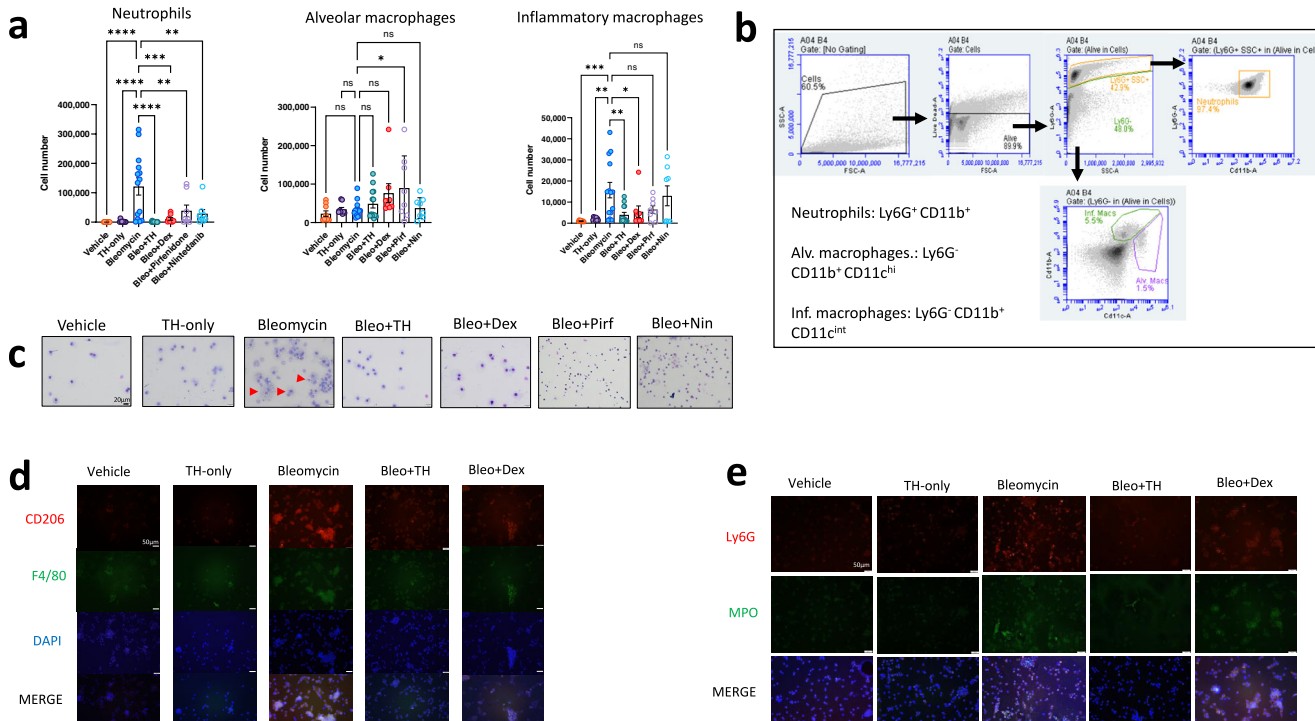

Fig. 5 | Inflammatory cell influx measured in murine BALF. Murine BALF was assessed for neutrophils, alveolar macrophages, and inflammatory macrophages (**a**) using flow cytometry, with the representative gating (**b**) strategy depicted alongside (Bleo $n = 14$, Bleo/TH $n = 13$, Bleo/Dex $n = 8$, Vehicle $n = 8$, TH only $n = 8$, Bleo/Pirf $n = 8$, Bleo/Nin $n = 8$). Data are presented as means ± SEM. ns: not significant. **a** Decreased numbers of neutrophils ($P < 0.0001$) and inflammatory macrophages ($P = 0.004$) were detected in response to TH5487 (i.p.) treatment (TH), with no significant difference reported between the neutrophils and inflammatory macrophages of mice treated with dexamethasone (Dex) or nintedanib (Nin). Almost no significant differences were seen for alveolar macrophage numbers, aside from pirfenidone (Pirf; $P = 0.0237$). Inflammatory cell numbers were compared to the vehicle/Bleo group using a one-way ANOVA (*$P < 0.05$). **c** Giemsa-Wright stained cytospin slides showing vehicle/Bleo BALF samples containing enlarged inflammatory macrophages, with TH5487 treatment reducing the presence of inflammatory macrophages, while the corticosteroid dexamethasone similarly reduced inflammatory macrophage influx comparable to vehicle-treated control samples. Results shown from 3 independent experiments. Scale bar = 20 μm. Immunofluorescent staining of murine BALF samples measured inflammatory macrophage (**d**) and neutrophil (**e**) content using CD206 (red)/F4/80 (green) and LY6G (red)/MPO (green), respectively. (DAPI, blue; scale bar = 50 μm. Source data are provided as a Source data file.

To further assess the translational utility of TH5487 for IPF treatment IPF and healthy lung controls (Supp. Table 1) were assessed for immunoreactivity measuring OGG1 and SMAD7. IPF patient lung immunoreactivity was significantly higher than healthy lung controls for both antibodies (SMAD7: $P = 0.0002$ and OGG1: $P = 0.0024$), indicating the potential role of OGG1 and SMAD7 in IPF disease (Fig. 8c, d; Supp. Figs. 20 and 21).

## Discussion

IPF is an interstitial lung disease characterized by dysregulated inflammation, progressive lung scarring, and eventual death due to respiratory complications[1,42]. Ensuing TGF-β production following lung injury results in the upregulation of tissue repair genes, including the DNA repair genes[16,43,44]. ROS generate localized increases in levels of oxidatively modified DNA base lesions, particularly 8-oxoG in guanine-rich gene regulatory regions, resulting in recruitment of OGG1, transcriptional effectors and pro-inflammatory gene expression[30,45,46]. Redox homeostasis imbalances have been implicated in several diseases, with reported deficiencies in glutathione and superoxide dismutase in the lower respiratory tracts of patients with IPF[47–49]. However, therapeutic interventions targeting oxidative DNA damage repair have received relatively little attention for fibrosis treatment despite the implication of mitochondrial and NADPH oxidase-derived ROS in IPF progression[50].

Current therapies, nintedanib and pirfenidone, significantly delay lung function decline[51,52]. However, no therapeutic solution exists which halts disease progression, necessitating the development of novel therapeutic interventions[53]. The approach reported in this study utilized a potent and selective small molecule, TH5487, employing a distinct mechanism of action, preventing OGG1 from binding damaged DNA, recruiting transcription factors, and upregulating pro-inflammatory and pro-fibrotic pathways required for epigenetic reprogramming and consequent airway remodeling[20,22]. Herein, we report several arguments supporting OGG1 as an appealing target for IPF treatment.

Reported data indicate that OGG1 binds 8-oxoG at regulatory gene domains, mediating transcriptional activation of inflammatory responses[13,31,34,35,54–60]. Our study confirms the OGG1-dependent ability of SMAD3 to bind to promoter regions of pro-fibrotic genes. OGG1 is increased in lung epithelial cells and fibroblasts following TGF-β1 addition, with IPF-derived fibroblasts generating higher levels of ROS[36,61]. Furthermore, abnormal wound healing of the alveolar epithelium in response to micro-injuries plays a crucial role in IPF progression[62–64]. Our study found OGG1 inhibition by TH5487 significantly decreased migration compared to TGF-β1 controls in all tested cell types, suggesting OGG1 inhibition suppresses the initiation of wound healing. While recent efforts have focused on removal of senescent or damaged fibroblasts[65,66], our approach advantageously suppresses expression of pro-fibrotic mediators and mitigates aberrant myofibroblast and epithelial cell involvement.

Importantly, uncontrolled lung injury is a hallmark of IPF initiation and progression, resulting in pro-inflammatory and pro-fibrotic cytokine release driving further fibrosis-related immune cell influx and ECM remodeling[18,67,68]. Our data demonstrate, targeting OGG1 suppresses

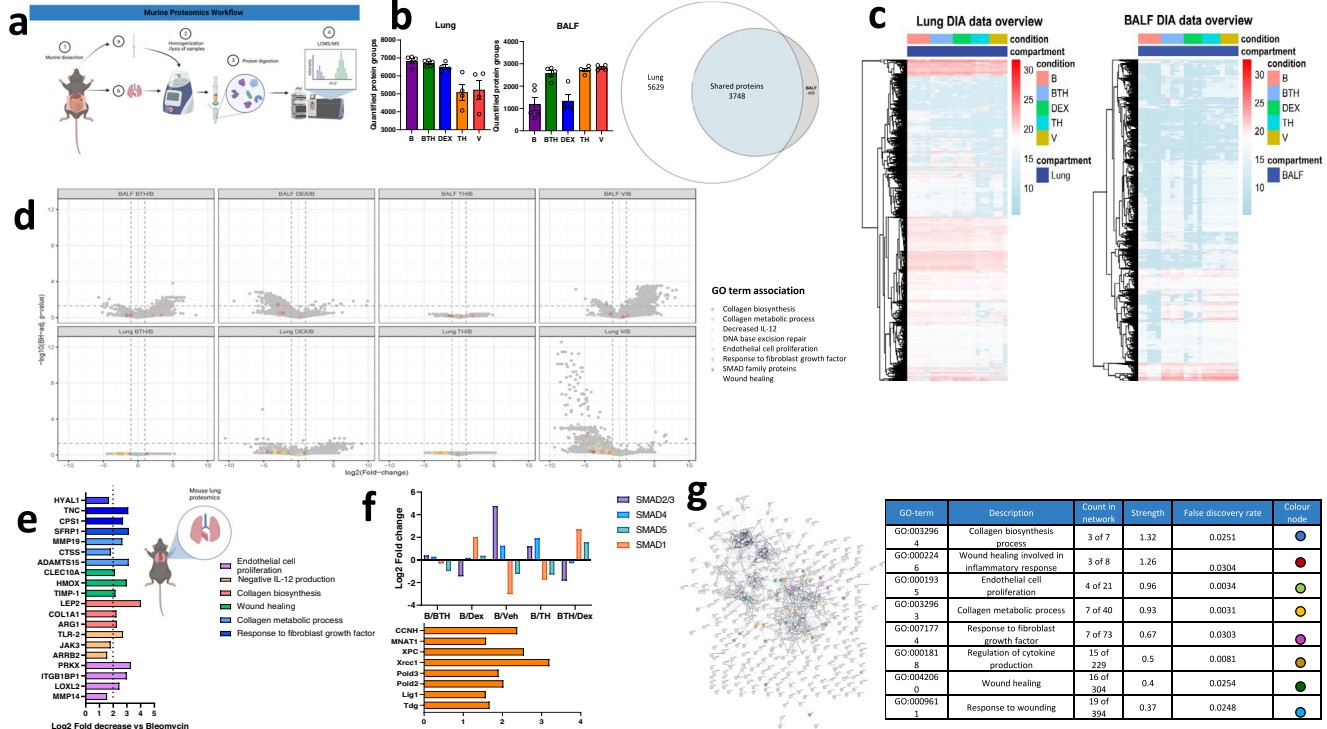

**Fig. 6 | Proteome of lungs and BALF. a** Experimental workflow showing analysis of protein extracts by mass spectrometry. **b** Unique and overlapping protein identifications subdivided by treatment condition in lungs and BALF by biological fluid. **c** Response profile heatmaps displaying protein expression across treatment conditions in each biological fluid. **d** Volcano plot depicting differentially expressed proteins in BALF (top panel) and lung (bottom panel) with proteins associated with significant gene ontology (GO) terms highlighted in each plot. **e**, **f** Specific lung proteins downregulated following Bleo/TH5487 treatment compared to the Bleo alone condition. Proteins highlighted in these plots are specifically associated with **e** fibrotic-related changes or **f** SMAD (top panel) and nucleotide base repair

(bottom panel). **g** Network analysis (StringDB) comparing the Bleo/Bleo+TH treatment condition displaying significant biological GO terms, protein counts in the network, strength, and FDR. For LCMS/MS analyses, murine samples were analyzed by single-shot DIA-MS for cohorts Bleo (B) $n = 5$, bleomycin/TH5487 (BTH) $n = 5$, bleomycin/dexamethasone (DEX) $n = 4$, TH5487 only (TH) $n = 4$ and vehicle only (V) $n = 4$, respectively. Significance criteria; absolute fold-change ≥2 & FDR, Benjamini−Hochberg-corrected $p$-value ≤0.05 unless indicated otherwise. Source data are provided as a Source data file. Elements of this figure were created with BioRender.com.

TGF-β1 and several key immune modulatory cytokines and chemokines in murine BALF, lungs, and plasma. Treatment with both TH5487, and to a lesser extent, *Ogg1* siRNA resulted in decreased pro-fibrotic cytokine expression and diminished immune cell recruitment to the lung compared to clinically used nintedanib and pirfenidone, both resulting in decreased lung collagen accumulation. Previous studies have described macrophage involvement in IPF[69–71], with particular emphasis on M2 macrophages[72,73] due to their involvement in TGF-β production[74]. Our study describes significant reduction of the M2 macrophage population following OGG1-targeting, supporting another beneficial aspect to this approach over currently used therapies[75].

To elucidate the mechanism by which these effects were mediated, we examined murine and human lung samples. OGG1 has displayed a crucial role in the TGF-β/SMAD axis, modulating EMT/FMT through the phosphorylation of SMAD2/3 by SMAD7[36]. This is supported in this study by proteomic analysis, immunostaining, and immunofluorescence which show that decreases in both SMAD7, SMAD2/3, and OGG1, as well as increases in several DNA repair-associated proteins, resulting in amelioration of fibrotic damage in both human and murine samples. Gene array assays positioned TH5487 as a superior treatment compared to nintedanib in the downregulation of several key fibrotic genes. In addition, proteins with known roles in fibrosis development including ARG1, TIMP1, COL1A1, and others[76] were significantly decreased following TH5487 treatment, confirming OGG1 as a tractable target for pharmaceutical intervention. In addition, 8-

oxoG, OGG1, and other base/nucleotide excision repair proteins may serve as biomarkers in patients with IPF, offering an alternative to clinical diagnoses.

Important limitations addressed in this study include, whether treatment using TH5487 can display utility in human IPF pathologies. While the translational aspect of the study is suggested using human lung sections, it is important to demonstrate that TH5487 treatment successfully decreases OGG1 levels and subsequent IPF lung damage in human clinical trials. Additional limitations in this study include the lack of monitoring of potential off-target effects induced by targeting OGG1. While no obvious reductions in key murine health status measures were observed in this or other studies[22], any small-molecule utilization should be accompanied by long term monitoring of adverse effects to ensure safe therapeutic usage. Furthermore, *Ogg1*[−/−] mice display no deleterious pathological changes[77], suggesting specific *Ogg1*-targeting may be safe. In any case, further efforts are required before progressing this compound to clinical trials.

Together, our findings demonstrate that TH5487 possesses a mechanism of action targeting OGG1 to suppress IPF, which is distinct from currently employed therapeutic interventions. This study further elucidates the downstream effects of this approach, decreasing myofibroblast transition, fibroblast migration, inflammatory cell recruitment, and eventual inhibition of fibrotic-related lung remodeling. These data show promising therapeutic effects of TH5487 in a mouse model of IPF and provide motivation for OGG1 as a tractable drug target for IPF.

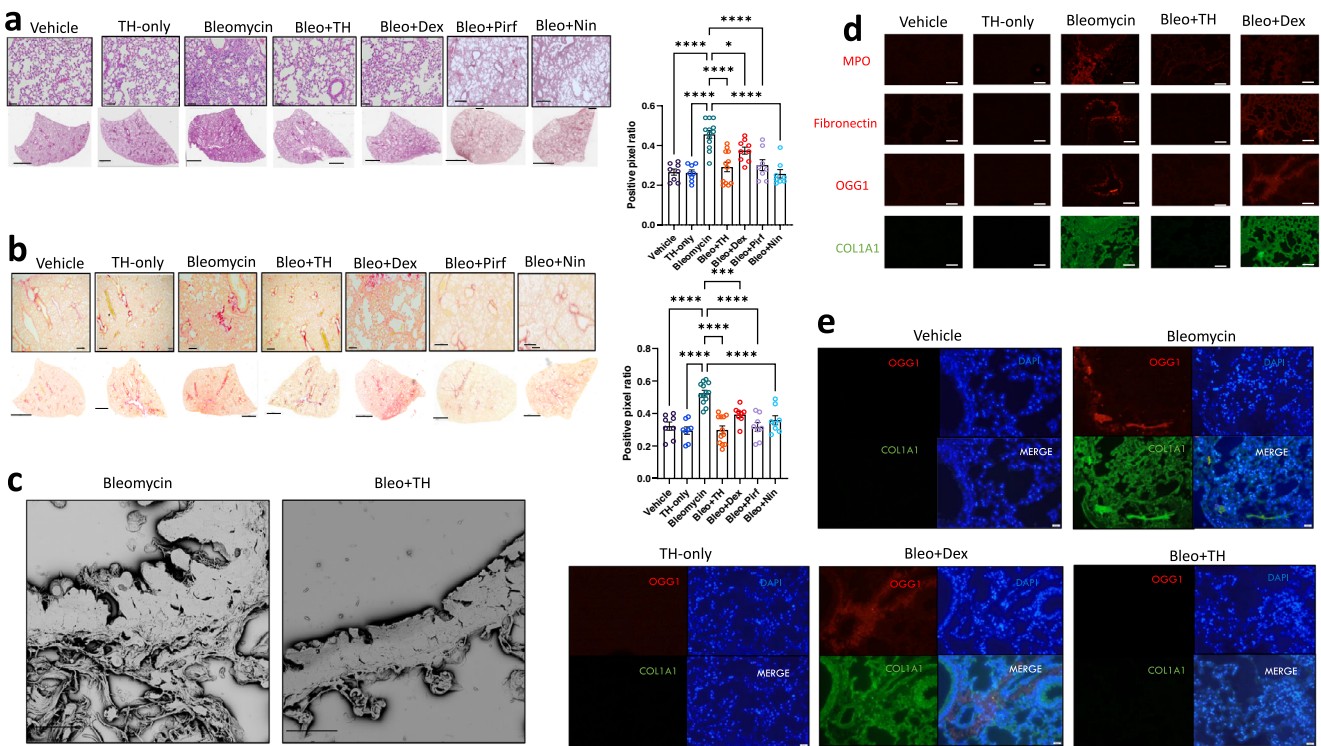

**Fig. 7 | Murine lung staining, scanning electron microscopy, and immuno-fluorescence show reduced levels of fibrotic-related lung damage following Bleo/TH5487 treatment compared to Bleo/vehicle samples.**
**a** TH5487 significantly decreased lung damage in Bleo-treated mice (H&E) and **b** collagen deposition (picrosirius red) in both macroscopic and microscopic structures compared to vehicle/Bleo lungs and was confirmed by positive pixel analysis of whole-lung scanned images (scale bar of microscopic image =100 μm; scale bar of whole lung scan = 2 mm). Results shown from 3 independent experiments. Statistical analyses were conducted using a one-way ANOVA (*$P < 0.05$; **$P < 0.01$; ***$P < 0.005$). ns: not significant. Murine samples included in these analyses: Bleo $n = 13$, Bleo/TH $n = 13$, Bleo/Dex $n = 8$, Vehicle $n = 8$, TH only $n = 9$, Bleo/Pirf $n = 7$, Bleo/Nin $n = 7$). Bleo vs Bleo+TH displayed significantly reduced levels of

H&E and picrosirius red staining ($P < 0.0001$). Data are presented as means ± SEM (**a**, **b**). **c** TH5487 /Bleo scanning electron microscopy images show reduced collagen deposition in the alveolar borders compared to Bleo-treated controls (scale bar = 20 μm). Immunofluorescent staining of murine lung sections (**d**) revealed decreased levels of MPO (red), fibronectin (red), OGG1 (red), and COL1A1 (green) following TH5487 treatment compared to both vehicle/Bleo and dexamethasone/Bleo groups (scale bar = 50 μm). **e** Co-stained murine lung sections revealed corresponding increases in OGG1 (red) and COL1A1 (green) following Bleo administration (DAPI counterstain, blue), with reduced levels of both OGG1 and COL1A1 in TH5487-treated samples (scale bar = 50 μm). Source data are provided as a Source data file.

## Methods

### Study design

The goal of this study was to test a novel pharmaceutical approach to inhibit OGG1, ultimately leading to the inhibition of fibrosis-related progression. Initial in vitro experiments in small airway epithelial and fibroblast cells displayed decreased fibrosis-related phenotypic features. Subsequent in vivo murine studies, using intratracheally-administered bleomycin to male C57BL/6J mice was chosen as a well-established and relevant model of experimental lung fibrosis. IPF is a disease predominantly affecting male patients and, similarly, male mice have been shown to produce a greater response to bleomycin compared to female mice[78]. Sample sizes were calculated by power analysis (G Power Software) based on previous pilot studies, feasibility, and to conform to the ARRIVE guidelines[79]. For lung fibrosis experiments testing the viability of *Ogg1* as a drug target in siRNA experiments and for TH5487 experiments mouse groups were determined a priori (with details provided in Supp. Table 2). The general humane endpoint was indicated by specific signs including a loss of 20% of original mouse weight and changes in mouse behavior. Downstream analyses were conducted with the investigator blinded to the treatment groups, and no animals were excluded as outliers from the reported dataset. All in vitro and in vivo experiments were performed in two to four technical replicates. Human resected lung sections were obtained with informed consent, with a statistically significant $n = 4$ samples used for IPF disease (3 male and 1 female) and control samples (3 female and 1 male).

### Materials availability

TH5487 utilized in these experiments was obtained from the Helleday Laboratory.

### Cell cultures

The following cell lines were utilized in this study pHLF (Lonza Biosciences), hSAEC (Lonza Biosciences), MEF (BNCC100518, ATCC), PC3 cells (CRL-1435™, ATCC), HaCat cells (AddexBio T0020001). Primary human lung fibroblast cells (pHLF) and human small airway epithelium cells (hSAECs) were cultured according to supplier's details (Lonza, Basel, Switzerland), Small airway epithelial cell growth medium (ProMoCell, C-21070), respectively. *Ogg1*[+/+] and *Ogg1*[-/-] mouse fibroblast (MF) cells were kindly provided by Dr. Deborah E. Barnes (Imperial Cancer Research Fund, Clare Hall Labs, United Kingdom). MF cells were maintained in DMEM/Ham's F-12 (3:1) supplemented with 10% fetal bovine serum (FBS), glutamine, penicillin (100 U), and streptomycin (100 μg/mL). PC3 cells (CRL-1435™) maintained in F-12K medium, supplemented with 10% FBS, glutamine, penicillin (100 U), and streptomycin (100 μg/mL). HaCaT cells were maintained in Dulbecco's modified Eagle's medium (DMEM) supplemented with 10% FBS, glutamine, penicillin (100 U), and streptomycin (100 μg/mL). Cells were routinely tested for mycoplasma contamination. Ogg1 expression in *Ogg1*[+/+] and *Ogg1*[-/-] MF cells were characterized by qRT-PCR. Exon 3-4F: 5′-TGGACCTC-GACTCATTCAGC-3′; R: 5′-CTTCGAGGATGGCTTTGGCA-3′; Exon 5-6F: 5′-TATGGCAGATTGCCCATCGT-3′; R: 5′-CCAGCATAAGGTCCCCACAG-3′.

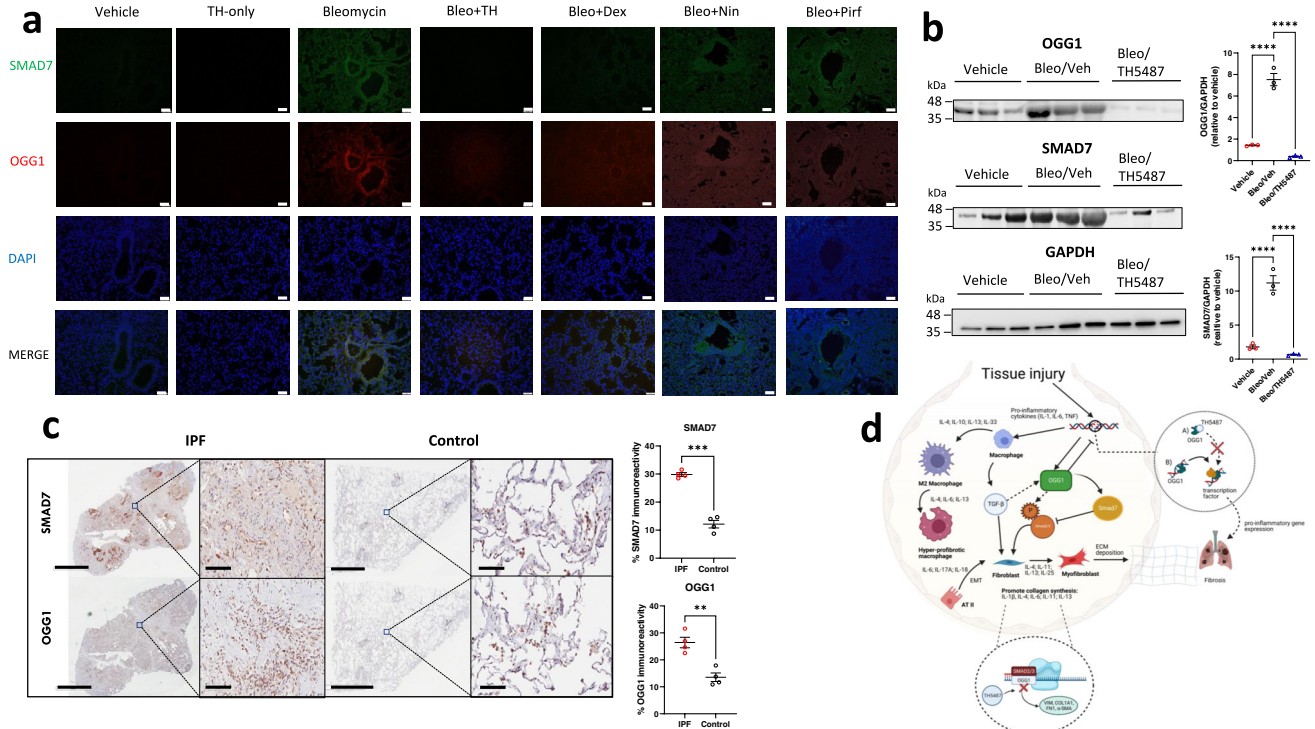

**Fig. 8 | Lower expression of OGG1 and SMAD7 are seen in both TH5487-treated mice and human explanted control lungs. a** Murine lung immunofluorescence analysis showed decreased levels of immunoreactivity for OGG1 (red)/SMAD7 (green) in TH5487 (TH)-treated mice compared to bleomycin (Bleo) controls (DAPI, blue; scale bar = 50 μm), results shown from 3 independent experiments, **b** with mouse lung homogenate samples analyzed by SDS-PAGE, followed by immuno-blotting using rabbit antisera specific to OGG1 and SMAD7 (protein levels quanti-fied inset *n* = 3 per treatment; statistical analyses were conducted using a one-way ANOVA (****$P$ < 0.0001 for comparisons between all groups tested)). Data are pre-sented as means ± SEM (**b**, **c**). **c** This result translated into human patient samples, with healthy control lung tissue displaying significantly decreased levels of OGG1 and SMAD7 immunoreactivity (brown staining; $P$ = 0.0024 and $P$ = 0.0002,

respectively), as compared using an unpaired *t* test with Welch's correction (two-sided), *n* = 4 patients per group. Scale bar = 6 mm and 100 μm for inset images. **d** Excessive OGG1 production facilitates pro-inflammatory gene expression, pro-moting inflammatory cell recruitment, leading to further exacerbation of the fibrotic lung environment. In addition, OGG1 promotes the phosphorylation of SMAD 2/3 by SMAD7 interaction, promoting TGF-β1-driven FMT and EMT and excessive ECM deposition. Therefore, decreased OGG1 expression and binding to DNA by TH5487 inhibits downstream pro-inflammatory gene expression and pul-monary fibrosis. In this study, OGG1 and SMAD7 levels were both reduced after the administration of TH5487, as shown by SDS-page/immunoblotting. Source data are provided as a Source data file. Elements of this figure were created with BioRender.com.

## Treatment of cells

Recombinant human TGF-β1 (Cat # AF-100-21C; PeproTech, Inc., Cranbury, NJ) was re-constituted in 4.4 mM HCl supplemented with 0.1% BSA at 1 μg/ml, aliquoted and kept at −80 °C until use. TGF-β1 was used at concentration of 2 ng/mL based on cell type and specific experiments. TH5487 (Science for Life Laboratory, Department of Oncology-Pathology, Karolinska Institutet, SE-171 76 Stockholm, Swe-den), a selective active-site inhibitor of OGG1 was dissolved in DMSO and added to cell cultures at 10 μM concentration and utilized in ani-mal experiments at 40 mg/kg.

## Fibroblast transwell experiment

Fibroblast chemotaxis was measured using 24-well Nunc (8 μm pore size) transwell inserts (ThermoFisher, MA, USA). pHLF, *Ogg1*$^{+/+}$, *Ogg1*$^{-/-}$ MEF cells were seeded (5 × 10$^5$ cells/mL) into the upper chamber in FBS-free medium, while the lower chamber contained complete medium with additional 10% FBS as a chemoattractant. Medium con-taining TGF-β1 (2 ng/mL) was added to each well and allowed to equilibrate for 24 h. Cells were washed with PBS, followed by the addition of medium containing TH5487, dexamethasone, medium only, or vehicle only (DMSO and PBS pH 7.4). Following 24 h, medium was removed and cells in the lower chamber were stained (crystal violet) and imaged using a Nikon microscope (Nikon, Tokyo, Japan) with a ×10 objective.

## Wound healing assay

Wound healing assays were conducted using MF and pHFL cells. Cells grown to confluence in 24-well plates, starved in 0.5% FBS-containing medium for 24 h. Wounds were made in the confluent cell layer using sterile 200 μL pipette tips, followed by washing with PBS and incuba-tion with complete culture medium with or without TGF-β1 (2 ng/mL) at 37 °C in a 5% CO$_2$ incubator for 28 h (MF cells) 48 h (pHFL cells) post-scratching. Wound images were documented using an Olympus CKX41 microscope with Olympus SC30 camera and cellSens Entry software (Olympus, Tokyo, Japan). Images were analyzed using the wound healing tool in ImageJ (https://imagej.nih.gov/ij/). Data are presented as percentages of the initial wound area.

## Antibodies

The following antibodies were used during this study, with dilutions included alongside. APC Rat anti-CD11b (BD 553312; 1:200), PE CY 7 Rat anti-CD11c (BD 558 079; 1:200), PE Rat anti-LY6G (BD 551461; 1:200), PE CY7 Rat anti-SiglecF (BD 562680 (1:200), PE Rat anti-I-Ad/I-Ed (BD 558593; 1:200). Rabbit anti-fibronectin (ab268020; 1:500), Rabbit anti-LY6G (ab238132; 1:1000), Mouse anti-COL1A1 (ab88147; 1:200), Rabbit anti-myeloperoxidase (ab208670; 1:200), Rabbit anti-mannose recep-tor (ab64693; 1:500), Rat anti-f4/80 (ab6640; 1:500), Rabbit anti-MADH7/SMAD7 (ab216428; 1:500), Rabbit anti-alpha smooth muscle actin (ab5694; 1:300), Rabbit anti-OGG1 (PA1-31402; 1:500), Rabbit

anti-vimentin (ab92547; 1:300), Mouse anti-GAPDH (ab8245; 1:1000), Alexa fluor-conjugated (488) goat anti-mouse secondary (ab150113; 1:2000), Alexa fluor-conjugated (647) goat anti-rabbit secondary (ab150083; 1:1000).

## Phalloidin and immunofluorescence staining
Cells were fixed in buffered 4% formaldehyde at room temperature for 15 min and washed 3 times in PBS. Triton X-100 (0.1%) in PBS added for 5 min to increase permeability, then washed 3 times in PBS and incubated with anti-alpha smooth muscle actin antibody (ab5694; Abcam, Cambridge, United Kingdom) at 1:300 dilution in Tris-buffered saline supplemented with Tween-20 (0.05%) (TBST) overnight at 4 °C. After washing three times with TBST, cells were incubated with Goat anti-Mouse IgG1 Secondary Antibody, Alexa Fluor 594 (A-21125; Invitrogen, Waltham, MA, USA) at 1:200 dilution at room temperature for 1 h. For visualization of F-actin, cells were stained with 50 ng/mL of Phalloidin (Fluorescein isothiocyanate-labeled (Cat # P5282, Millipore-Sigma, Burlington, MA, USA) for 15 min at room temperature. DNA was counter-stained with 10 ng/mL DAPI (4′,6′-diamidino-2-phenylindole dihydrochloride) for 15 min and cells were washed. Cells were mounted in antifade medium (Dako, Carpinteria, CA). Fluorescent images were captured using OLYMPUS Microscope System (BX53P) with a built-in digital CCD color camera DP73WDR.

## Immunostaining of pHLF cells
pHLF cells were seeded (1 × 10⁴ cells/ml) into 24-well plates containing rounded glass cover slips. TGF-β1 (2 ng/mL) was added to each well, followed by the addition of medium containing TH5487, dexamethasone, nintedanib, medium only, or vehicle only. Following 24 h of treatment, cells were washed and fixed with ice-cold methanol, and then treated with 0.5% Triton X-100. Cells were blocked using Dako Protein Block (Agilent, CA, USA) for 1 h at room temperature and then primary antibodies, rabbit anti-COL I, rabbit anti-αSMA, rabbit anti-fibronectin, and rabbit anti-vimentin antibodies (Abcam, Cambridge, UK) were added for overnight. AlexaFluor 488-conjugated goat anti-rabbit secondary antibody (Invitrogen) was used. Glass cover slips were mounted onto glass slides, with nuclei counter-stained using DAPI-containing fluoroshield (Abcam). Images were visualized using a Nikon Confocal Microscope. Fluorescence was quantified using ImageJ software (Java 1.8.0_172).

## Ethical approval
All animal experiments were approved by the Malmö-Lund Animal Care Ethics Committee (M17009-18). Human lung tissue was obtained after written informed consent, approval by the Regional Ethical Review Board in Lund (approval no. LU412-03) and performed in accordance with the Declaration of Helsinki as well as relevant guidelines and regulations.

## Animal studies
10–12-week-old male C57Bl/6 mice (Janvier, Le Genest-Saint-Isle, France) were housed at least 2 weeks in the animal facility at the Biomedical Service Division at Lund University before initiating experiments and were provided with food (RM1 (P) 801151; SDS, Essex, UK) and water ad libitum throughout the study. Mice were housed within a facility utilizing 12 h light/dark cycles, with temperatures maintained between 23 and 25 °C (humidity 50%). Mice were randomly allocated into five groups: intratracheally (i.t.)-administered bleomycin (Apoex, Lund, Sweden; 2.5 U/kg) + vehicle intraperitoneal (i.p.), bleomycin (i.t.) + TH5487 (40 mg/kg; i.p.), bleomycin (i.t.) + nintedanib (60 mg/kg p.o.), bleomycin (i.t.) + pirfenidone (300 mg/kg, p.o.), bleomycin (i.t.) + dexamethasone (10 mg/kg; i.p.), saline (i.t.) + vehicle (i.p.), saline (i.t.) + TH5487 (40 mg/kg; i.p.). Nintedanib (Cat # SML2848, Sigma-Aldrich, Saint Louis, MO, dissolved in DMSO), Pirfenidone ((Cat # P2116, Sigma-Aldrich, Saint Louis, MO, dissolved in DMSO). Dexamethasone (Cat # D8893, Sigma-Aldrich, Saint Louis, MO) were prepared as above.

For the siRNA experiments, lung fibrosis was induced by i.t. introduction of bleomycin (2.5 U/kg) or saline as control. Following 14 days, siRNA targeting mouse Ogg1 (L-048121-01-0050) or non-targeting control (D-001810-01-50) siRNA (Horizon Discovery, UK) was administered i.t at either 25 μg per mouse (Ogg1 siRNA) or 50 μg of non-targeting siRNA. In addition, TH5487, nintedanib, or pirfenidone was administered daily i.p. or p.o. On day 21, mice were euthanized followed by collection of blood, lung tissue, and BALF.

## Blood collection
Blood was collected in 0.5 M EDTA tubes by cardiac puncture and centrifuged at 1000 × g for 10 min. Plasma was used for the analysis of inflammatory mediators using a multiplex assay (Bio-plex assay; Bio-Rad, Hercules, CA).

## Collection of lung tissue
Right lungs were collected in Eppendorf tubes on dry ice and stored at −80 °C. The snap-frozen lungs were thawed and homogenized in tissue protein extraction reagent (T-PER) solution (ThermoFisher) containing protease inhibitor (Pefabloc SC; Sigma-Aldrich) at a final concentration of 1 mM. Lung homogenates were centrifuged at 9000 × g for 10 min at 4 °C, and the supernatants were collected for multiplex analysis. Left lungs were collected in Histofix (Histolab, Göteborg, Sweden) and submerged in 4% buffered paraformaldehyde solution.

## Bronchoalveolar lavage fluid (BALF) collection
BAL was performed with a total volume of 1 mL PBS containing 100 μM EDTA. BALF was collected in Eppendorf tubes on ice, with aliquots made for flow cytometry, cytospin differential counts, and an aliquot transferred to −80 °C for multiplex cytokine analysis. Cytospin preparations of cells were stained with modified Wright-Giemsa stain (Sigma-Aldrich, St. Louis, MO).

## Flow cytometry
Flow cytometry was carried out using a BD Accuri C6 Plus (BD Biosciences, Franklin Lakes, NJ). The washed cells were incubated with Fixable Viability Stain 510 (FVS510; BD #564406) to differentiate live and dead cells. Cells were washed with Stain buffer 1x (BD #554656) and incubated with Lyse Fix 1x (BD #558049 (5x)). Fixed cells were washed with stain buffer and aliquoted into two samples incubated with either anti-CD11b (BD553312), anti-CD11c (BD558079), anti-Ly6G (BD551461), or anti-CD11c, anti-MHCII (BD558593), anti-SiglecF (BD562680) antibodies. Data analysis was completed using BD Accuri C6 Plus Software (v 1.0.27.1).

## Bioplex cytokine analysis
For the detection of multiple cytokines in BALF, plasma, and lung homogenate, the Bio-Plex Pro mouse cytokine assay (23-Plex Group I; Bio-Rad) was used on a Luminex-xMAP/Bio-Plex 200 System with Bio-Plex Manager 6.2 software (Bio-Rad, Richmond, CA). A cytometric magnetic bead-based assay was used to measure cytokine levels, according to the manufacturer's instructions. The detection limits were as follows: Eotaxin (4524.58–1.23 pg/mL), GCSF (99,318.6–7.3 pg/mL), GMCSF (6310.48–3.91 pg/mL), IFN-γ (16,114.01–0.87 pg/mL), IL-1α (10,055.54–0.54 pg/mL), IL-1β (31,512.04–1.75 pg/mL), IL-2 (19,175.48–1.24 pg/mL), IL-3 (7514.5–0.44 pg/mL), IL-4 (5923.58–0.34 pg/mL), IL-5 (12,619.59–0.78 pg/mL), IL-6 (9409.63–0.68 pg/mL), IL-9 (64,684.09 −2.41 pg/mL), IL-10 (77,390.75–4.18 pg/mL), IL-12p40 (144,560.15– 18.62 pg/mL), IL-12p70 (78,647.56–4.81 pg/mL), IL-13 (197,828.67– 11.16 pg/mL), IL-17 (8727.85–0.51 pg/mL), KC (23,001.9–1.4 pg/mL), MCP-1 (393,545.52–10.01 pg/mL), MIP-1α (14,566.62–0.63 pg/mL), MIP-1β (7023.87–0.34 pg/mL), RANTES (19,490.48–4.61 pg/mL), and TNF-α (74,368.54–51.69 pg/mL). Cytokine measurements for lung

homogenate samples were corrected for total protein concentration using a Pierce™ BCA Protein Assay Kit (ThermoFisher, Maltham, MA).

## Hydroxyproline assay

Hydroxyproline levels in murine lung tissues were determined using the QuickZyme Hydroxyproline Assay kit (Quickzyme Biosciences, Leiden, the Netherlands). Lung tissues were homogenized as described above. Homogenates were diluted (1:1 vol:vol) with 12 N HCl and hydrolyzed at 95 °C for 20 h. After centrifugation at 13,000 × $g$ for 10 min, 200 µL from the supernatant was taken and diluted 1:2 with 4 N HCl. Hydroxyproline standard (6.25–300 µM) was prepared in 4 N HCl and transferred to the microtiter plate. Following addition of a chloramine T-containing assay buffer, samples were oxidized for 20 min at RT. Detection reagent containing p-dimethylaminobenzaldehyde was added and after incubation at 60 °C for 1 h, absorbance was read at 570 nm with a VICTOR 1420 Multilabel plate reader (PerkinElmer). The hydroxyproline content in lung tissue is given as hydroxyproline (µg) per mg lung tissue, corrected using a Pierce™ BCA Protein Assay Kit (ThermoFisher).

## TGF-β1 ELISA

The Quantikine ELISA kit targeting TGF-β1 (R&D systems, Abingdon, UK) was used to assess TGF-β1 levels in the BALF, plasma, and lung homogenate of murine samples according to the manufacturer's instructions. Optical density was measured at 450 nm using a VICTOR 1420 Multilabel plate reader (PerkinElmer, Waltham, MA).

## Lactate dehydrogenase H assay

LDH levels were determined in homogenized murine lung tissues using an LDH ELISA kit for murine samples according to the manufacturer's instructions (Cloud-Clone Corporation, Katy, TX). Optical density was measured at 450 nm using a VICTOR 1420 Multilabel plate reader (PerkinElmer).

## RT-qPCR

Total RNAs were extracted using RNeasy total RNA purification kit according to the manufacturer's instructions (Cat # 74106; Qiagen). RNA concentrations were determined using a NanoDrop ND1000 (Saveen Werner, Limhamn, Malmö). Complementary DNA was synthesized from 1 µg of total RNA using an iScript cDNA synthesis kit with oligo (dT) and primers (Cat # 1708840; Bio-Rad Laboratories). Transcript levels of genes were quantitated by qPCR using iQ SYBR green supermix (Cat # 1725120; Bio-Rad Laboratories) with gene-specific primers from Integrated DNA Technologies (see below). Gene expression values were calculated using the 2 − ΔΔCT method and normalized with β-actin. The primers used are listed below:

αSma
F: 5′-CCCAGACATCAGGGAGTAATGG-3′
R: 5′-TCTATCGGATACTTCAGCGTCA-3′
Col1a1
F: 5′-GCTCCTCTTAGGGGCCACT-3′
R: 5′-CCACG CT ACCATTGGGG-3′
Fn1:
5′-ATGTGGACCCCTCCTGATAGT-3′
R: 5′-GCCCAGTGATTTCAGCAAAGG-3′
Vim1:
F: 5′-CGTCCACACGCACCTACAG-3′
R: 5′-GGGGGATGAGGAATAGAGGCT-3′
β-actin:
F: 5′-ATCTGGCACCACACCTTC-3′
R: 5′-AGCCAGGTCCAGACGCA-3′
FN1:
F: 5′-TCAGAGCTCCTGCACTTTTG-3′
R: 5′-GTAACGCACCAGGAAGTTG-3′
COL1A1:

F: 5′-CCAGAAGAACTGGTACATCAGCA-3′
R: 5′-CGCCATACTCGAACTGGAATC-3′
β-ACTIN:
F: 5′-ACAGAGCCTCGCCTTTGCCG-3′
R: 5′-ACATGCCGGAGCCGTTGTCG-3′

## Real-time PCR array

Total mRNA was extracted from lung tissue submerged in RNAlater using a RNeasy Mini Kit (Qiagen, Hilden, Germany) according to the protocol from the manufacturer. RNA concentrations were determined using a NanoDrop ND1000 (Saveen Werner, Limhamn, Malmö). Equal amounts of RNA were pooled from all animals in each group ($n = 6$). cDNA was synthesized with an iScript Advanced cDNA Synthesis Kit (Bio-Rad) and mixed with RT² SYBR® Green ROX™ qPCR Mastermix. A volume of 25 µL of the reaction mixture was added to each well of a RT² Profiler™ PCR Array Mouse Fibrosis (GeneGlobe, PAMM-120Z). The RT-PCR reaction was performed using a QuantStudio™ 7 Flex system (Thermo Fisher Scientific) and data analysis was performed using the manufacturer's web-based software (https://geneglobe.qiagen.com/analyze). Normalization of gene expression was performed using the following house-keeping genes: *B2m*, *Actb*, *Gusb*, *Gapdh*, and *Hsp90ab1*.

## Sample preparation for LCMS

**Tissue homogenization.** Freshly harvested lungs in PBS were homogenized in MagNA Lyser (Roche) with silica beads (VWR) using 2 × 2 min cycles. Protein estimation in the homogenates was performed using a BCA kit (Thermo Fisher Scientific). 50 µg of protein from organs and 1 µL of plasma were utilized for digestion.

**Protein denaturation, tryptic digest & peptide solid-phase-extraction.** Proteins were resuspended in 8 M urea in 100 mM ammonium bicarbonate, reduced with 5 mM TCEP, pH 7.0 for 45 min at 37 °C, and alkylated with 25 mM iodoacetamide (Sigma, USA) for 30 min followed by dilution with 100 mM ammonium bicarbonate to a final urea concentration below 1.5 M. Proteins were digested by incubation with trypsin (1/100, w/w; Promega) for at least 9 h at 37 °C. Digestion was stopped using 5% trifluoracetic acid (Sigma) to pH 2–3. The peptides were cleaned up by C18 reversed-phase spin columns as per the manufacturer's instructions (Harvard Apparatus, Holliston, MA, USA). Samples were resuspended in 30 µL HPLC-water (Thermo-Fisher Scientific) with 2% acetonitrile, 0.2% formic acid (Sigma-Aldrich). Samples were spiked with iRT peptides (Biognosys, Schlieren, Switzerland).

## LC- Data-independent acquisition MS analysis

All peptide analyses were performed on a Q Exactive HF-X mass spectrometer (Thermo Fisher Scientific) connected to an EASY-nLC 1200 ultra-high-performance liquid chromatography system (Thermo Fisher Scientific). Peptides were trapped on pre-column (PepMap100 C18 3 µm; 75 µm × 2 cm; Thermo Fisher Scientific) and separated on an EASY-Spray C-18 reversed phase column with integrated ESI emitter (250 mm, ES902, column temperature 45 °C; Thermo Fisher Scientific). Equilibrations of columns and sample loading were performed per manufacturer's guidelines. Solvent A was used as stationary phase (0.1% formic acid), and solvent B (mobile phase; 0.1% formic acid, 80% acetonitrile) was used to run a linear gradient from 5% to 38% over 120 min at a flow rate of 350 nL/min. The 44 variable windows data-independent acquisition (DIA) acquisition method is derived from Bruderer et al.[80]. The mass range for MS1 was 350–1650 *m/z* with a resolution of 120,000 and a resolution of 30,000 for MS2 with a stepped normalized collision energy (NCE) of 25.5, 27, and 30. The 44 variably sized MS2 windows were 350–371, 370–387, 386–403, 402–416, 415–427, 426–439, 438–451, 450–462, 461–472, 471–483, 482–494, 493–505, 504–515, 514–525, 524–537, 536–548, 547–557,

556–568, 567–580, 579–591, 590–603, 602–614, 613–626, 625–638, 637–651, 650–664, 663–677, 676–690, 689–704, 703–719, 718–735, 734–753, 752–771, 770–790, 789–811, 810–832, 831–857, 856–884, 883–916, 915–955, 954–997, 996–1057, 1056–1135 and 1134–1650 $m/z$, resulting in a total cycle time of ~3.3 s and 6–8 points per chromatographic peak on average.

## Peptide and precursor identification and quantification from DIA-MS data

Protein sequences employed for proteomics data interpretation were downloaded from the UniProt database (1 sequence per gene, 17.09.2020). Peptide-centric analysis of DIA data was carried out using an in silico predicted spectral library using the DIANN Toolset (v1.7.12) and a two-pass analysis procedure. In the first pass, an empirically corrected, in silico predicted spectral library was created from the sample set of interest by querying the full proteome predicted library (with parameters Trypsin/P, up to 1 missed cleavage, excision of N-terminal M and carbamidomethylation of cysteine residues enabled as variable modification). The predicted library and initial search space thereby spanned 22,159 proteins, 33,323 protein groups and 4,141,742 precursors of 1,357,505 peptide species in distinct elution groups. DIANN was instructed to optimize mass accuracy and RT extraction windows separately for each MS run to maximize identification sensitivity and to then write out a new, constrained and empirically corrected spectral library (First-pass FDR was set to 1%. Protein inference was disabled to maintain protein grouping information and the option to reduce RAM usage was enabled). The produced libraries for lung/BALF/plasma datasets contained 10,121/5,017/941 protein groups, 97,775/33,857/6,025 precursors and 78,267/28,672/4,335 peptides in distinct elution groups, respectively. In the second pass of the analysis, the three corrected libraries were then applied to the datasets of the respective tissue type to derive quantitative information (FDR set to 1 % and with high accuracy quantification strategy selected).

## Downstream analysis of DIA proteomics data

DIA quantitative data from DIANN (Pass2.tsv) were imported and further processed in the R environment for statistical computing for visualization and analysis (R version 4.0.2) employing packages including data.table, pheatmap, fviz, ggplot2, ggrepel, and limma.

To detect protein expression differences, quantitative data were scaled on precursor level, tested for statistically significant differences and summarized to protein level via a customized script. Specifically, raw peptide intensities ("Precursor.Quantity") were log2-transformed, scaled by quantile normalization per MS run, excluded from analysis if observed in ≤4 MS runs, and then pair-wise student's t-tests performed per precursor and pair of conditions/cohorts. Since DIANN 1.7.12 does not provide an alignment or background quantification module, prior to statistical tests, missing values were imputed, sampling from a standard distribution centered on the 0.1%-ile of the observed data points, with a standard deviation of 0.2, parameters chosen upon visual inspection of the observed data. Protein level metrics were derived from the mean of precursor-level metrics (both, log2 fold-change, and $P$-value). Multiple hypothesis testing correction via the Benjamini–Hochberg method was applied.

To interpret groups of co-regulated proteins, protein-level log2 fold-change profile similarity across the comparisons of interest BALF and lung tissue datasets was calculated (1- Euclidean distance), including only proteins that were regulated in at least one of the comparisons (|log2 fold-change| ≥1 and corrected protein level $P$-value ≤ 0.05). The number of clusters in the accordingly sub-setted data was determined via the Gap statistic method as implemented in the package fviz.

Functional annotation over- and under-representation testing among protein sets of interest (regulated in a particular comparison, members of a specific co-regulation cluster, etc.) was performed in the PANTHER classification system[81] (http://pantherdb.org/, Release 15.0) using the Statistical over representation test (Fisher's exact test), testing against the background of all proteins covered by the MS analysis.

## H&E and picrosirius red staining of lung tissue

Mouse left lungs were fixed in Histofix (Histolab Products AB, Askim, Sweden), paraffin-embedded and sectioned at 3 μm. The tissue sections were placed on slides (Superfrost Plus; Fisher Scientific) and deparaffinized in serial baths of xylene and ethanol followed by staining using Mayer hematoxylin and 0.2% eosin (Histolab Products AB, Askim, Sweden) or picrosirius red staining kit (Abcam). The stained slides were imaged using an Aperio CS2 image capture device.

## Scanning electron microscopy of lung sections

Lung tissue sections were fixed as reported above. After fixation, samples were washed and dehydrated in alcohol at increasing concentrations, dried, mounted on aluminum holders, and covered with 20 nm of gold. Samples were examined in a Philips XL30 FEG scanning electron microscope (Eindhoven, The Netherlands) operated at an acceleration voltage of 5 kV.

## Immunostaining of murine lung sections

Lung tissue sections were fixed as reported above. Lung samples underwent antigen retrieval (pH 9 buffer) using a Dako PT Link pretreatment module (Agilent, CA, USA). Samples were washed and blocked for 10 min (Dako protein block; Agilent, Santa Clara, CA) before being treated with primary antibodies overnight. Mouse anti-COL1A1, rabbit anti-fibronectin, mouse anti-ly6G, rabbit anti-MPO, rabbit anti-mannose receptor, rat anti-F4/80, and mouse anti-SMAD7 (Abcam, CAM, UK), and rabbit anti-OGG1 (Invitrogen, Carlsbad, CA) antibodies were used. Alexa Fluor 488-conjugated goat/anti-mouse and Alexa Fluor 647 goat/anti-rabbit (Invitrogen, CA, USA) were used as secondary antibodies. Glass cover slips were mounted with DAPI-containing fluoroshield (Abcam). Images were visualized using a Nikon Confocal Microscope and fluorescence was quantified using ImageJ software (Java 1.8.0_172).

## Immunohistochemistry (IHC) of human lung samples

Macroscopically normal, tumor-free lung tissue samples were obtained during transplantation or resection from patients undergoing cancer surgery or IPF lung resection. Patients were selected following high-resolution computed tomography positivity (opaque fibrosis-like areas and honeycombing in the lung parenchyma). Patients were also older than 55 and predominantly male given the disease prevalence. Control patients were also older than 50 and had undergone explant surgery in relation to lung transplantation. Immediately after collection, samples were placed in 4% buffered formaldehyde. Following dehydration and embedding in paraffin, sections (3 μm) were produced. A single staining protocol (EnVision™ Detection system, K5007, Dako, Glostrup, Denmark) was used for visualization of OGG1 and SMAD7. Briefly, after antigen retrieval, OGG1 and SMAD7 were detected using rabbit anti-Ogg1 (Abcam) and rabbit-SMAD7 (Sigma-Aldrich) antibodies (1:1000) and visualized using secondary goat anti-rabbit antibodies conjugated with peroxidase polymers (Dako). IHC protocols were performed using an automated IHC robot (Autostainer Plus, Dako). Sections were counter-stained with Mayer's hematoxylin for visualization of background tissue, dehydrated in alcohol/xylene, and mounted on Pertex (Histolab, Göteborg, Sweden). The stained slides were imaged using an Aperio CS2 image capture device. Positive staining was quantified manually (number of cells/area of tissue) or as positivity (positive brown pixels divided by all stained pixels) using computerized image analysis on blinded sections using ImageScope (Aperio).

## Statistical analysis

In this study, groups of three or more mice were compared using one-way analysis of variance (ANOVA) with Dunnett's post hoc test. In experiments using two groups, results were compared using unpaired $t$-test with Welch's correction. Results in this study are displayed throughout as mean ± SEM. Statistical testing was carried out using GraphPad Prism 9.1.1 (San Diego, USA) with statistical significance defined as $P < 0.05$.

## Reporting summary

Further information on research design is available in the Nature Portfolio Reporting Summary linked to this article.

## Data availability

The mass spectrometry proteomics and initial search results generated in this study have been deposited in the ProteomeXchange Consortium database via the PRIDE partner repository (http://proteomecentral.proteomexchange.org). The processed data are available using (dataset identifier PXD029625). The processed proteomic data generated in this study are provided in the Source data file. Source data are provided with this paper.

## Code availability

Downstream analysis R code is available at https://github.com/heuselm/DiffTestR/tree/Tanner2021. The code is also available at Zenodo (https://zenodo.org/record/7463478#.Y6GxRHbMLEY), https://doi.org/10.5281/zenodo.7463478; 2022). All remaining data are available in the main text or the supplementary materials.

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

## Acknowledgements

We would like to acknowledge the assistance received from Maria Baumgarten; Lund University. In addition, we would like to acknowledge the flow cytometry support received from Assoc. Prof. Oonagh Shannon (Lund University). Figures feature elements created in BioRender.com. The work was supported by grants from Swedish Research Council 2020–011166 (A.E.). The Swedish Heart and Lung Foundation 20190160 (A.E.). The Swedish Government Funds for Clinical Research 46402 (ALF; A.E.). The Alfred Österlund Foundation (A.E.). Vinnova Swelife 2, 2018–03232 (A.E., T.H., C.K.). Horizon 2020 ERC-PoC (T.H.). US NIH, National Institute of Allergy and Infectious Diseases, AI062885 (I.B.). Royal Physiographic Society of Lund (L.T.). Landshövding Per Westlings Minnesfond RMh 2020-0015 (L.T.). Tore Nilsons Stiftelse 2021-00936 (L.T.). US. NIH National Institute of Allergic and Infectious Diseases (NIAID)/AI062885 (I.B.) Lars Hiertas Minne Fund FO 2021-0284 (L.T.).

## Author contributions

Conceptualization: A.E., L.T., A.B.S., T.H., C.K., J.B., and I.B.; methodology: L.T., A.B.S., R.K.V.B., M.H., T.M., C.A.Q.K., J.M., R.M.O., C.C., C.K.A., L.P., K.W., J.S.E., O.W., T.H., J.B, and IB.; MS data management & analysis: M.H.; investigation: L.T., A.B.S., R.K.V.B., M.H., T.M., C.A.Q.K., R.M.O., C.C., C.K.A., J.S.E., J.B., L.P., K.W., and I.B.; funding acquisition: A.E., C.K., and T.H.; project administration: A.E., C.K., and T.H.; supervision: A.E., C.K., and T.H.; writing—original draft: L.T.; writing—review & editing: L.T., A.B.S., A.E., C.K., M.H., J.B., and I.B.

## Funding

## Competing interests

T.H. is listed as inventor on a provisional U.S. patent application no. 62/636983, covering OGG1 inhibitors. The patent is fully owned by a nonprofit public foundation, the Helleday Foundation, and T.H. is a member of the foundation board developing OGG1 inhibitors toward the clinic. An inventor reward scheme is under discussion. The remaining authors declare no other competing interests.
