## [Peer Review File · Nature Communications]

Small-molecule-mediated OGG1 inhibition attenuates pulmonary inflammation and lung fibrosisREVIEWER COMMENTS

Reviewer #1 (Remarks to the Author):

The manuscript assesses the role of targeting OGG1 as a potential strategy for treating IPF. The use a combination of in vitro cell biology, in vivo animal models and human tissue to undertake a wide-ranging series of experiments to assess its potential as a pre-clinical asset for treatment of IPF. The strength of this manuscript is the use of a range of models, targeting OGG1 by both siRNA and an inhibitor of OGG1-DNA interactions and the use of both prophylactic and therapeutic models. However, there are a number of concerns which dampen my enthusiasm for this manuscript.

- 1) The in vitro studies lack rigour. The concentration of TGF β used is supraphysiological by a considerable margin (the TGF β concentration curve usually plateaus at 2ng/ml), the cell lines used are not relevant to IPF A549 and BES2B are lung cancer cell lines and MEFs and HFL1s are embryonic and foetal lines and it is not clear that 10uM is the ideal concentration of TH5487.
- 2) The bleomycin studies are comprehensive but the design does not appear as statistically robust as stated in the methods. If the power calculations and numbers were defined a priori why were the numbers variable across groups and where is the power calculation and what was the primary endpoint? In Figure 2 the therapeutic dosing shows only a marginal effect with OGG1 siRNA (the labelling as *Ogg1*^{-/-} implies gene deleted mice were used I assume this is an error) and a larger (but still small) effect with TH5487 at 40mg/kg but only 5 animals per group were used. In contrast the prophylactic dosing studies used 13 mice per group. It is unsurprising that the prophylactic dosing had a bigger effect and suggests that TH5487 is working primarily as an anti-inflammatory agent.
- 3) The use of Dexamethasone as a comparator only makes sense if they are investigating the anti-inflammatory role of this pathways as Dex has been shown to have little effect in modifying fibrotic disease. They need to use nintedanib or pirfenidone to benchmark TH5487.
- 4) The human studies are descriptive at best and why they have focused on SMAD7 which is inhibitory for TGF β activation and fibrogenesis.

Reviewer #2 (Remarks to the Author):

Tanner et al Nature Communications

The manuscript by Tanner et al reports substantial data supporting a central role for OGG1 in airway inflammation and progression to pulmonary fibrosis. Overall the study builds off of over a decade of foundational work from the Boldogh laboratory in which that group pioneered the discovery and molecular mechanisms underlying OGG1's role in regulating the inflammatory cascade post ROS-inducing exposures. This contribution by Tanner represents a logical extension of work presented in Visnes et al 2018 Science and follows a disease progression model based on the principles previously established following ragweed pollen challenge. Although there are overall significant contributions in this manuscript, there are significant concerns about experimental design, data presentation, data interpretation, all of which are described below.

In the Introduction (lines 23-25), the authors suggest that the literature focuses on OGG1's role in carcinogenesis and Alzheimer's disease (references 18-22). Two of these references are severely out-of-date and the authors omit references to the significant role of OGG1 in airway inflammation and metabolic syndrome. References 23 & 24 should also be made current as there are multiple outstanding reviews from which to choose.

Lines 36-51 give a summary of the findings of the manuscript – the Introduction should be limited to introducing and justifying the study to follow, and not be a recapitulation of the results prior to any data presentation. This should be removed and the experimental rationale and hypotheses expanded. In Fig. 1a, the x-axis is not labeled for any of the cell lines. Also, it would be very helpful if these experiments would have been done with *Ogg1*^{-/-} MEFs or alveolar epithelium cells, etc. Obviously, the presumption is that the TH5487 is highly specific for OGG1, but off-target effects cannot be ruled out. Utilization of knockout and knockout-complemented cells would help establish the interpretation of these data.

Also, in Fig 1c, the mechanism accounting for the lack of elevated levels of OGG1 (as measured via

immunofluorescence) following treatment with TH5487 should be demonstrated whether at a transcriptional or enzyme stability perspective. Are there data to suggest that TH5487 binding to OGG1 changes the intracellular half-life of the enzyme or can this observation be fully accounted for by a lack of transcriptional up regulation following TGF-beta 1 treatment? These data should be straightforward since mRNA abundance has been measured in these samples for multiple other genes.

The experimental rationale for using bleomycin for pulmonary fibrosis induction needs justification. As an anticancer drug, it produces a host of both direct and indirect DNA damages, including double- and single-strand breaks, base loss, and ROS induction. Cellular DNA repair and tolerance responses to this plethora of damage are extremely and unnecessarily complex. Formation of 8-oxoG adducts in DNA following bleomycin treatment would seem to be a relatively minor issue since Ogg1 knockout mice tolerate well treatments with potassium bromate for extended amounts of time; this is well described in: Arai T, Kelly VP, Minowa O, Noda T, Nishimura S. *Toxicology*. 2006 Apr 17;221(2-3):179-86 and other subsequent publications from this group. Additional points concerning this figure: 1) the x-axis is not labeled in Fig 2b; 2) the symbols used in 2b for PBS/NT and Bleomycin/TH5487 are not sufficiently different to know which data set is which; 3) the use of the nomenclature "Ogg1^{-/-}" in the figure is deceptive since these are wild-type mice – this should indicate that these were siRNA treated. It is also unclear why there was a 14-day interval between bleomycin treatment and the delivery of the Ogg1 or control siRNAs. Further, were there empirical data to guide the timing of the harvesting of tissues one week after the siRNA treatments? In the intratracheal administration of bleomycin or siRNAs, are there estimates for the depth of penetration in the lungs – if this is known, it would be useful to add to help in the understanding of the extent of lung involvement with these treatments. As mentioned above, the authors should have used an Ogg1^{-/-} mouse as a control for these experiments – these data will serve as the proper controls from which to assess the role of TH5487.

In Fig 3b, again there is no label on the x-axis and in the 2nd panel, the x-axis is shifted into the middle of the data set. Concerning these data, the extent of weight loss in the bleomycin-treated mice is disturbing since it exceeds the >20% allowable by all IACUC protocols that I am aware of. What was the criteria used for euthanizing the mice? The "Ethical Approval" section needs to be expanded. The size of the figure panels and labels in Fig 4 is too small – expanding it to >200% makes it barely readable – to aid in these data presentation, inclusion of only selective data is recommended and table(s) can report the essential data. In prior studies (Visnes et al 2018 *Science*), Cxcl2, IL6 and Tnf were described in detail as being modulated by TH5487 treatment. Of those, only IL6 is featured in these data – for ease of comparison with data presented in that prior publication, addition of Cxcl2 and Tnf should be included throughout.

In Fig 5a, the rationale and purpose for inclusion of dexamethasone should be included. In this regard, the authors do not address the very large differences in neutrophil numbers between the bleomycin alone and the bleomycin + dex control treatment; this trend is also very evident in the inflammatory macrophage data in which (with the exception of 1 mouse) the + dex treatment seems more effective than the TH5487 treatment.

Proteomic analyses are very solid.

Inclusion of human tissue analyses, with comparative murine data is a very positive contribution.

Reviewer #3 (Remarks to the Author):

This is a well-designed and executed study, and a well-written manuscript. The findings should be of interest to the readership of *Nature Communications*. The value of the proteomics dataset should also be of high value to the community. I recommend publication after the following issues are addressed.

- The authors report a decrease in the level of OGG1 protein upon inhibitor treatment. The reported inhibitor acts by blocking the catalytic action of OGG1: how do, then, the authors explain the observed decrease in OGG1 protein level? This question needs to be addressed in more detail. Stated differently, if the only action of the inhibitor was to interfere with the catalytic action of OGG1 as an enzyme, one would not expect to observe a decrease in OGG1 levels. Could it be that the OGG1-inhibitor complex formation somehow tags OGG1 for degradation or is this phenomenon strictly an

artifact of the IPF model system that utilizes bleomycin treatment? Have the authors examined the effect of inhibitor treatment on OGG1 levels in a system different from the IPF/bleomycin model?

- The use of the word "novel" in reference to the inhibitor itself is not warranted because the inhibitor was published 3 years ago in Science. The present study and its findings are novel but the inhibitor itself is not.

- Line 38: did the authors mean to use the word "resulting" instead of "resolving"?

- Lines 61 and 62 (and perhaps elsewhere): replace the Latin lowercase "u" with Greek mu symbol to denote micromolar.

- Line 333: the correct city name is Manassas.

Reviewer #4 (Remarks to the Author):

I have been asked to evaluate the proteomics aspects of this manuscript....

The manuscript describes an evaluation of a new small-molecule inhibitor of OGG, TH5487, for treating inflammation and fibrosis in the lung. The proteomic aspects of the manuscript investigate changes in protein expression after TH5487 treatment and the conclusion is that the changes observed are linked to known phenotypes of fibrosis.

My biggest concern with the proteomics is the number of animals in each treatment group. I could not actually find an explicit statement of how many were used but from the number of columns in Fig. 6C it appears that N=5 and N=4 (depending on condition) were used. Since each replicate was a different animal, these N values seem exceptionally low. I understand that a lot of animals were used overall but still, the low N makes the robustness of many of the changes suspect.

Apart from this, the rest of the statistics and data treatment seem in-line with expectations in the field. The one other gap, however, is that none of the custom scripts (e.g., that described in lines 563-565) or R code are available so evaluating much of what has been done here is difficult.

Minor points:

This statement seems inappropriate: "To our knowledge, this represents the deepest single-shot proteomic map of murine lung and BALF proteomes reported to date". For one thing this is not even a true statement since those numbers reported are from the combination of MANY experiments, not a single shot. And each of the 9,000+ proteins was not detected in each sample. I would just remove it because it does not add anything here.

Line 539: should be 6-8 POINTS, not peaks

Line 554: 10.121 should be 10,121

REVIEWER COMMENTS

Reviewer #1 (Remarks to the Author):

The manuscript assesses the role of targeting OGG1 as a potential strategy for treating IPF. The use a combination of in vitro cell biology, in vivo animal models and human tissue to undertake a wide-ranging series of experiments to assess its potential as a pre-clinical asset for treatment of IPF. The strength of this manuscript is the use of a range of models, targeting OGG1 by both siRNA and an inhibitor of OGG1-DNA interactions and the use of both prophylactic and therapeutic models. However, there are a number of concerns which dampen my enthusiasm for this manuscript.

1) The in vitro studies lack rigour. The concentration of TGF β used is supraphysiological by a considerable margin (the TGF β concentration curve usually plateaus at 2ng/ml), the cell lines used are not relevant to IPF A549 and BES2B are lung cancer cell lines and MEFs and HFL1s are embryonic and foetal lines and it is not clear that 10uM is the ideal concentration of TH5487.

Studies have shown that “the formation of actin stress fibers, as well as the downregulation and delocalization from the plasma membrane of epithelial markers such as TJP1 (ZO-1) and CDH1, which contribute to the tight junctions and adherence junctions,” “These effects are readily detectable after treatment of HaCaT cells for 48 h with a saturating dose (2 ng/mL) of TGF- β 1” (Miller et al., *Cell Reports* 25, 1841–1855 (2018; DOI: 10.1016/j.celrep.2018.10.056). In another example, PC3 and DU145 cells were used along with lower concentrations of TGF- β 1 than in our study (J-N Qi et al., *Front Oncol*, 9:1535 (2020; DOI: 10.3389/fonc.2019.01535).

However, other reports have utilized higher concentrations of TGF- β 1 in a context-dependent and cell-specific manner. For example, a) “... TGF- β (5 ng/ml) impacts cytokine-induced NO production in airway epithelial cells by reducing iNOS mRNA and protein levels through a ROCK-dependent pathway.” (Jiang et al. 2011; DOI: 10.1152/ajplung.00464.2010) b) “*Transforming growth factor beta* (TGF- β) plays major roles in ... treated with and without 10 ng/ml TGF- β 1, TGF- β 2 or TGF- β 3 for 72 h”. (Xu et al. 2018; DOI: 10.1038/s41413-017-0005-4) c) “To analyze the effect of *TGF- β* , cells were treated with 10 ng/mL TGF- β ... was based on two cell types (Tejera-Muñoz et al. 2021; DOI: 10.3390/ijms23010375; BEAS-2B and A549) to screen EMT and...” (Kim et al. 2020; DOI: 10.1038/s41598-020-67325-7) d) “Conversely, the treatment with TGF- β (10 ng/mL) induced a significant decrease in the T β RII levels”

To analyze responsiveness to TGF- β 1 of our model cell cultures (MEF, hSAEC), we performed two independent studies.

Study 1: We have determined the TGF- β 1 concentration dependent expression at the mRNA level of *aSma*, *Col1a1*, *Fn1* and *Vim* using MEF cells. As shown in Fig 1. (Supplementary Fig. 1a, c), 2 ng/mL and 5 ng/mL of TGF- β 1 did not significantly affect fibrotic gene expression, with 10 ng/mL significantly increasing transcript levels.

Fig. 1. Cells at 70% confluence were exposed to increasing concentrations (0, 2.5, 5, and 10 ng per mL) of TGF-β1 and harvested at day 4. mRNA levels were determined by qRT-PCR.

Study 2: In these experiments adult human small airway epithelial cells (hSAEC) were utilized to determine concentration dependent effects of TGF-β1 on development of mesenchymal phenotype. The figure below shows that hSAEC required 10 ng/mL TGF-β1 for development of myofibroblast morphology and actin polymerization to F-actin. Percentage of cells showing morphological changes and polymerized actin filaments were monitored after daily administration of TGF-β1 (0, 2.5, 5, 10 and 20 ng per mL) for 12 days. This is mentioned on line 59 and shown in Supplementary Figure 1b in the revised manuscript.

Fig. 2 TGF-β1 concentration-dependent development of mesenchymal phenotype of hSAEC. Upper panels. Phase contrast images were photographed using NIKON TE system (10x objective). Lower panels: Actin stress fibers are visualized by phalloidin-FITC staining of cells (images are acquired using NIKON TE system, 20x objective).

The cell lines used are not relevant to IPF A549 and BES2B are lung cancer cell lines and MEFs and HFL1s are embryonic and fetal lines

We have used primary human cell lines for more accurate readouts and, more specifically, have repeated the experiments using primary human small airway epithelial cells and human lung fibroblasts as well as *Ogg1*^{+/+} and *Ogg1*^{-/-} fibroblast cells (recommended by Reviewer #2). Results are shown in Fig. 1 of the revised manuscript and mentioned in lines 54-64 in the revised version of the manuscript.

It is not clear that 10uM is the ideal concentration of TH5487

TH5487 concentration utilized in these studies was dependent on **a)** inhibitory effect of TH5487 on gene expression and **b)** its solubility.

a) Effective concentrations of TH5487 were determined using *Ogg1* proficient MEF cells. In brief, MEF cells at 60% confluence were exposed to increasing doses of TGF- β 1 for 4 days and changes in mRNA levels of *α Sma*, *Col1a1*, *Fn1* and *Vim* were determined by RT-qPCR. Results show that 10 μ M of TH5487 were needed to inhibit *α Sma*, *Col1a1*, *Fn1* and *Vim* expression at the mRNA level (Fig. 3). Treatment of cells with 20 μ M of TH5487 did not further decrease expression of these profibrotic genes. Lower TH5487 concentrations (1.25 μ M, 2.5 μ M, 5 μ M) did not significantly inhibit gene expression. These results are also presented in Supplementary Fig. 1c).

Fig. 3. Concentration dependent inhibition by TH5487 of TGF- β 1-induced gene expression.

b) Solubility of TH5487 in physiological salt solution (e.g., PBS) at pH: 7.4 is \sim 12 μ M (room temperature). In complete cell culture media (10% FBS) TH5487 solubility is similar \sim 12 μ M (pH: 7.4, at 37 $^{\circ}$ C); (Visnes et al. 2018; doi: 10.1126/science.aar8048). Note: Solubility of TH5487 is similar to that of 8-oxoGua base (Boldogh et al. 2012; doi: 10.1074/jbc.C112.364620).

2) The bleomycin studies are comprehensive but the design does not appear as statistically robust as stated in the methods. If the power calculations and numbers were defined a priori why were the numbers variable across groups and where is the power calculation and what was the primary endpoint?

We appreciate the comments from the reviewer. The power calculations were determined *a priori* as follows (added to the reporting summary), which we believe to be statistically sound. From an initial pilot study involving 3 groups of mice ($N=5$ per group) we determined an initial effect size (f) of 0.6 allowing us to perform formal power and sample size calculations. Using the freely downloadable software G Power (Faul, Erdfelder, Lang and Buchner, 2007; DOI: 10.3758/bf03193146), we calculated

an *a priori* sample size based on the following values: $\alpha=0.05$; Power=0.9; number of groups=5. Based on the pilot study data, an ANOVA with fixed effects, omnibus, one-way was used to calculate the ideal total sample size value of 50 ($N=10$ per group).

However, the study was designed with the knowledge that the administration of bleomycin causes mortality in 10-20% of the mice before the crucial onset of fibrosis. With this knowledge, the corrected group sizes for the bleomycin administered groups would be calculated at a minimum of $N=12$. To account for unexpected murine death (due to intraperitoneal injection) we allowed for $N=13$ mice per bleomycin treated group. The remaining two groups, namely vehicle and TH5487 (TH)-only were decided upon considering the 3 R's in the ARRIVE guidelines for animal use. It was decided that with no changes expected within the vehicle only group, based on our preliminary studies, a smaller group size could be used. Finally, the TH-only group included the recommended number of animals ($N=10$) to achieve statistical power. This approach was sufficient to detect significant biological differences and offer translational value from this set of experiments. This is mentioned on L338-340 in the revised manuscript.

Secondly, the endpoint for these studies was set as follows. The mice were euthanized as soon as the weight crossed the 20% cutoff mark or if the mice displayed any signs of distress as listed in our ethical permit. The general humane endpoint is indicated by many signs including a loss of 20% of original mouse weight and changes in mouse behavior including:

- discomfort or stress: (e.g. dull/staring coat (hair erect); decreased or increased activity; avoidance behaviour; isolation from the group; depressed; decreased appetite)

- deterioration (e.g. Discharge from eyes; loss of general condition; anorexia; dehydration (tenting skin/sunken eyes); weakness; decreased motility)
- distress: (e.g. very weak; unresponsive to touch; unconscious; convulsing; difficulty breathing)
- If obvious distress (bleeding trachea, labored breathing etc.) following intratracheal administration of compound/bleomycin is seen, the mice will be immediately euthanized.
- Mice that experience severe adverse reactions to the test compound or anaesthetic drug and are deemed to be in distress will be euthanized. Animals showing neurological signs, that have convulsions, demonstrate severe weakness, avoidance behaviour, display signs of depression, show respiratory distress, or open-mouthed breathing will humanely euthanized prior to the end point.

3) In Figure 2 the therapeutic dosing shows only a marginal effect with OGG1 siRNA (the labelling as Ogg1^{-/-} implies gene deleted mice were used I assume this is an error) and a larger (but still small) effect with TH5487 at 40mg/kg but only 5 animals per group were used. In contrast the prophylactic dosing studies used 13 mice per group. It is unsurprising that the prophylactic dosing had a bigger effect and suggests that TH5487 is working primarily as an anti-inflammatory agent. The use of Dexamethasone as a comparator only makes sense if they are investigating the anti-inflammatory role of this pathways as Dex has been shown to have little effect in modifying fibrotic disease. They need to use nintedanib or pirfenidone to benchmark TH5487.

We thank the reviewer for this comment and agree that the labelling is confusing, this has been amended to “Ogg1 siRNA.” The reason for including the initial lung specific siRNA-targeting of Ogg1 in Figure 2 was to demonstrate that Ogg1 was a tractable target for pharmaceutical intervention, with the inclusion of a TH5487-treated group for further support of this point. We were exceptionally aware of the 3R's as laid out in the ARRIVE guidelines, and as such we reduced the group sizes as compared to the more-in depth studies presented later in the manuscript. We then proceeded to delineate the effects of Ogg1-targeting in subsequent prophylactic experiments, necessitating the use of more animals. We feel that the reduced number of animals does not highlight TH5487 as anti-inflammatory as the reviewer suggests. Instead, by comparing the anti-inflammatory dexamethasone to TH5487 we have established that despite similar reductions in immune cell recruitment, TH5487 displayed significant decreases in fibrotic lung damage whilst dexamethasone did not. This indicates that TH5487 reduces fibrotic lung damage by additional mechanisms as highlighted in the manuscript. This is further supported by the significantly reduced lung damage seen in the TH 5487/bleomycin and Ogg1 siRNA/bleomycin groups (Supp. Fig. 4). Similar efforts on which we based the siRNA administration included in these studies have successfully been undertaken (Haak et al. 2019; doi: 10.1126/scitranslmed.aau6296.);(Bacsı et al. 2013; DOI: 10.1016/j.dnarep.2012.10.002), before progressing to prophylactic treatment.

Furthermore, as the reviewer suggests, we have utilized nintedanib in cell culture and experimental animal studies. We found nintedanib to be more relevant to our studies as inhibits TGF- β 1 signaling, including tyrosine phosphorylation of the type II TGF- β receptor, phosphorylation of SMAD3, consequently down-regulates ECM production (Wollin et al. 2015; doi: 10.1183/09031936.00174914). We have also conducted benchmarking studies using pirfenidone and nintedanib. Pirfenidone (400 mg/kg, PO) and nintedanib (60mg/kg, PO) were dosed both prophylactically (n=8 per group) and therapeutically (n=8 per group) in a murine model of bleomycin to benchmark the TH5487 treatment as shown in other studies (Lv et al. 2013; DOI: 10.1371/journal.pone.0068631) and Wollin et al. 2014; DOI: 10.1124/jpet.113.208223). This is mentioned on line 84-86 and throughout the manuscript.

3) The human studies are descriptive at best and why they have focused on SMAD7 which is inhibitory for TGFb activation and fibrogenesis.

We thank the reviewer for this comment. We feel that the human studies add significant translational value to the study as indicated by the other reviewers. We have added a table describing the patient characteristics (Supp. Table 1).

Supplementary Table 1: Showing clinical data for patient samples included in this study. Healthy control samples were obtained from cancer patients undergoing lung resectioning (values not available-NA; F-female; M-male). FEV1- Forced expiratory volume in one second; FEV1%-percentage forced expiratory volume % of expected FEV1 (in relation to age, height, and sex); FEV1/FVC- represents the proportion of a person's vital capacity expired in the first second of forced expiration (FEV1) divided by the forced vital capacity (FVC); DLCO-diffusion capacity of the lungs to carbon monoxide (CO per unit time per mm of driving pressure of CO (cc of CO/sec/mm of Hg)).

Patient ID	Sex	Diagnosis	FEV1	FEV1%	FVC	FEV1/FVC	DLCO (%)
LURES 005	F	Adenocarcinoma	2.0	90	2.46	0.81	50
LURES 010	F	Atypical carcinoma	1.78	59	2.37	0.75	64
LURES 013	F	Adenocarcinoma	2.5	101	3.37	0.74	96

LURES 018	M	Squamous epithelial cell cancer	3.1	93	NA	NA	NA
LUEX 18	M	Lung Fibrosis	2.2	53	2.78	0.79	34
LUEX 25	M	Lung Fibrosis	1.22	41	1.56	0.81	NA
LUEX 55	M	Lung Fibrosis	2.2	73	2.52	0.87	26
LUEX 24	F	Lung Fibrosis	1.32	53	1.76	0.75	33

We agree with the reviewer to a degree as SMAD7 has been shown to be inhibitory of liver fibrosis. However, a recent publication (Wang et al. 2020; doi: 10.1096/fj.201901291RRRRR) observed that OGG1 promoted TGF- β 1-induced cell transformation and activated Smad2/3 by interacting with Smad7. The interaction between OGG1 and the TGF- β /Smad axis modulates cell transformation of lung epithelial cells and fibroblasts. Moreover, they demonstrated that *Ogg1* deficiency relieved pulmonary fibrosis in bleomycin-treated mice. *Ogg1* knockout decreased the bleomycin-induced expression of Smad7 and phosphorylation of Smad2/3 in mice (Wang et al. 2020; doi: 10.1096/fj.201901291RRRRR). Therefore, our logic to highlight the interaction between SMAD7 and OGG1 (Figure 8) is consistent with previous findings, offering a direct translational link into human IPF. This is mentioned on line 303-308 in the revised manuscript.

Reviewer #2 (Remarks to the Author):

Tanner et al Nature Communications

The manuscript by Tanner et al reports substantial data supporting a central role for OGG1 in airway inflammation and progression to pulmonary fibrosis. This contribution by Tanner represents a logical extension of work presented in Visnes et al 2018 Science and follows a disease progression model based on the principles previously established following ragweed pollen challenge. Although there are overall significant contributions in this manuscript, there are significant concerns about experimental design, data presentation, data interpretation, all of which are described below.

1) In the Introduction (lines 23-25), the authors suggest that the literature focuses on OGG1's role in carcinogenesis and Alzheimer's disease (references 18-22). Two of these references are severely out-of-date and the authors omit references to the significant role of OGG1 in airway inflammation and metabolic syndrome. References 23 & 24 should also be made current as there are multiple outstanding reviews from which to choose.

We thank the reviewer for these comments. We have updated the references in Line 24 with the following:

20. Brasier, A. R. & Boldogh, I. Targeting inducible epigenetic reprogramming pathways in chronic airway remodeling. *Drugs in Context* (2019). doi:10.7573/dic.2019-8-3
21. Vlahopoulos, S., Adamaki, M., Khoury, N., Zoumpourlis, V. & Boldogh, I. Roles of DNA repair enzyme OGG1 in innate immunity and its significance for lung cancer. *Pharmacol. Ther.* 194, 59–72 (2019).
22. Visnes, T. et al. Small-molecule inhibitor of OGG1 suppresses proinflammatory gene expression and inflammation. *Science* (80). 362, 834 LP – 839 (2018).
23. Ba, X. & Boldogh, I. 8-Oxoguanine DNA glycosylase 1: Beyond repair of the oxidatively modified base lesions. *Redox Biol.* 14, 669–678 (2018).
24. Sampath, H. & Lloyd, R. S. Roles of OGG1 in transcriptional regulation and maintenance of metabolic homeostasis. *DNA Repair (Amst)*. 81, 102667 (2019).
25. Burchat, N. et al. Maternal Transmission of Human OGG1 Protects Mice Against Genetically- and Diet-Induced Obesity Through Increased Tissue Mitochondrial Content . *Frontiers in Cell and Developmental Biology* 9, (2021).
26. Sampath, H. et al. 8-Oxoguanine DNA Glycosylase (OGG1) Deficiency Increases Susceptibility to Obesity and Metabolic Dysfunction. *PLoS One* 7, e51697 (2012).
27. Dizdaroglu, M., Coskun, E. & Jaruga, P. Repair of oxidatively induced DNA damage by DNA glycosylases: Mechanisms of action, substrate specificities and excision kinetics. *Mutat. Res. Mutat. Res.* 771, 99–127 (2017).
28. Beard, W. A., Horton, J. K., Prasad, R. & Wilson, S. H. Eukaryotic Base Excision Repair: New Approaches Shine Light on Mechanism. *Annu. Rev. Biochem.* 88, 137–162 (2019).

2) Lines 36-51 give a summary of the findings of the manuscript – the Introduction should be limited to introducing and justifying the study to follow, and not be a recapitulation of the results prior to any data presentation. This should be removed and the experimental rationale and hypotheses expanded.

We thank the reviewer for this comment and have summarized the closing of the introduction as follows (Lines 34-49).

‘In this regard, a small molecule inhibitor of OGG1-DNA interactions, TH5487, was shown to decrease in vivo levels of pro-inflammatory gene expression²². More specifically, TH5487 lowered DNA occupancy of OGG1 at guanine-rich promoter regions, subsequently impeding tumor necrosis factor- α (TNF α)-induced OGG1-DNA interactions, reducing immune cell recruitment²². In our study, a bleomycin-challenged murine model was used to assess TH5487’s anti-fibrotic efficacy. This model reproduces several phenotypic features of human IPF, including peripheral alveolar septal thickening, dysregulated cytokine production, and immune cell influx resulting in fibrosis^{37,38}. Herein we demonstrate that TH5487 inhibits migratory and proliferative capacities of lung-derived epithelial and fibroblast cells in

vitro. Additionally, intratracheal administration of *Ogg1*-targetting small interfering RNA (siRNA) or TH5487, mitigates distinct bleomycin-induced pulmonary immune cell recruitment and fibrotic lung damage. Finally, we confirm the involvement of base/nucleotide excision repair and Mothers against decapentaplegic homolog (SMAD) family of proteins, with OGG1 and several SMAD proteins shown to drive fibroblast proliferation and differentiation^{36,39}. Taken together, these data strongly suggest that TH5487 is a potent, specific, and clinically-relevant treatment for IPF.'

3) In Fig. 1a, the x-axis is not labeled for any of the cell lines. Also, it would be very helpful if these experiments would have been done with *Ogg1*^{-/-} MEFs or alveolar epithelium cells, etc. Obviously, the presumption is that the TH5487 is highly specific for OGG1, but off-target effects cannot be ruled out. Utilization of knockout and knockout-complemented cells would help establish the interpretation of these data.

We thank the reviewer for this comment and have labelled all x-axes in Figure 1a. As the reviewer suggests, we have utilized *Ogg1*^{+/+} and *Ogg1*^{-/-} MEF cells. Results show highly significant responses of *Ogg1*^{+/+} MEF cells to TGF-β1 at RNA and protein levels (*αSma*, *Col1a1*, *Fn*, and *Vim*), which was inhibited by TH5487 (Fig. 4a, b). Lack of OGG1 (*Ogg1*^{-/-} MEF) or treatment of *Ogg1*^{+/+} MEFs with TH5487 significantly lowered TGF-β1-induced cell migration (trans-well assays) (Fig. 4c, d). Because TH5487 has no significant effect on mRNA, protein, and cell migration in *Ogg1*^{-/-} MEF cells, one may conclude that its off-target effect is minimal if any, in this cell culture model. This is also presented on lines 54-64 in the revised manuscript.

Fig. 4. Effects of TH5487 on mRNA, protein levels of α Sma, Col1a1, Fn, and Vim and TGF-β1-induced cell migration

4) Also, in Fig 1c, the mechanism accounting for the lack of elevated levels of OGG1 (as measured via immunofluorescence) following treatment with TH5487 should be demonstrated whether at a transcriptional or enzyme stability perspective. Are there data to suggest that TH5487 binding to OGG1 changes the intracellular half-life of the enzyme or can this observation be fully accounted for by a lack of transcriptional up regulation following TGF-beta 1 treatment? These data should be straightforward since mRNA abundance has been measured in these samples for multiple other genes.

Currently, we do not have direct conclusive evidence to show that “TH5487 binding to OGG1 changes the intracellular half-life of the enzyme.” To test for potential mechanism(s), we first show that long-term (4 days) exposure of cells to TGF- β 1 increases in *Ogg1* mRNA (Fig. 5bc) and protein levels (Fig. 5d). We note that these data are in line with increased expression of OGG1 in bleomycin-induced fibrosis in the mouse model and fibrotic tissues of human lungs (Fig. 8b in the revised manuscript). TH5487 had no effect on *Ogg1* mRNA levels (Fig. 5c). These results suggest that TH5487 not only inhibits DNA substrate binding of OGG1 (documented previously Visnes et al. 2018; doi: 10.1126/science.aar8048) but may subject OGG1 for degradation in TGF- β 1-exposed cells (Fig. 5d) raising the possibility “TH5487 binding to OGG1 changes the intracellular half-life of the enzyme.” Future studies will determine OGG1 half-life *in vitro*. This is mentioned on lines 95-99 in the revised manuscript.

Fig. 5. mRNA Characterization of *Ogg1*^{+/+} and *Ogg1*^{-/-} MEF cells and effect of TH5487 on *Ogg1* expression in TGF- β 1 \pm TH5487-treated *Ogg1*^{+/+} MEF cells. (a) RT-PCR analysis of exon #3-4 and Exon #5-6 RNA levels in *Ogg1*^{+/+} and *Ogg1*^{-/-} MEF cells. (b) Time course analysis of *Ogg1* expression in mock- and TGF- β 1 (10 ng/mL)-treated *Ogg1*^{+/+} MEFs. (c) There are no effects of TH5487 (10 μ M) on *Ogg1* expression at mRNA levels. (d) Western blot analysis of *Ogg1* in *Ogg1*^{+/+} MEF cells, with representative Western blot shown for OGG1 and GAPDH.

5) The experimental rationale for using bleomycin for pulmonary fibrosis induction needs justification. As an anticancer drug, it produces a host of both direct and indirect DNA damages, including double- and single-strand breaks, base loss, and ROS induction. Cellular DNA repair and tolerance responses to this plethora of damage are extremely and unnecessarily complex. Formation of 8-oxoG adducts in DNA following bleomycin treatment would seem to be a relatively minor issue since *Ogg1* knockout mice tolerate well treatments with potassium bromate for extended amounts of time; this is well described in: Arai T, Kelly VP, Minowa O, Noda T, Nishimura S. Toxicology. 2006 Apr 17;221(2-3):179-86 and other subsequent publications from this group. Additional points concerning this figure: 1) the x-axis is not labeled in Fig 2b; 2) the symbols used in 2b for PBS/NT and Bleomycin/TH5487 are not sufficiently different to know which data set is which; 3) the use of the nomenclature "*Ogg1*^{-/-}" in the figure is deceptive since these are wild-type mice – this should indicate that these were siRNA treated. It is also unclear why there was a 14-day interval between bleomycin treatment and the delivery of the *Ogg1* or control siRNAs. Further, were there empirical data to guide the timing of the harvesting of tissues one week after the siRNA treatments?

We thank you for Reviewer comments suggestions. Our point-by-point responses are as follows

The experimental rationale for using bleomycin for pulmonary fibrosis induction needs justification.

The reviewer is entirely correct that bleomycin causes many DNA lesions, mostly those not relevant for OGG1 mediated DNA repair. We agree 8-oxoG would form only a fraction of lesions and those lesions are likely not toxic (DNA double-strand breaks are much more toxic) as documented by *Arai and colleagues* (Arai et al. 2002; DOI: 10.1093/carcin/23.12.2005; Arai et al. 2006; doi: 10.1016/j.tox.2006.01.004). *Mmh/Ogg1*-deficient mice accumulate 8-oxoG, cells whilst retaining their ability to proliferate (Arai et al. 2003; PMID: 12874039).

Rationale using bleomycin-induced lung fibrosis in testing potential clinical utility of the OGG1 inhibitor TH5497 can be summarized as follows:

a) As reviewed previously “The bleomycin animal model: a useful tool to investigate treatment options for idiopathic pulmonary fibrosis” (Tashiro et al. 2017; DOI: 10.3389/fmed.2017.00118).

b) Bleomycin causes robust inflammatory and fibrotic responses within a short period of time (14 to 21 days; especially after intratracheal instillation). The “switch” between inflammation and fibrosis occurs around day 9 after bleomycin, with a peak approximately day 14 (Moeller et al. 2008; doi: 10.1016/j.biocel.2007.08.011). Although bleomycin is highly genotoxic, generate oxidative stress (major source of cytotoxicity), inducer of robust innate immune responses as measured by increases in expression of pro-inflammatory cytokines (e.g., IL-1, IL-4, TNF- α , IL-6, MCP-1) chemokines (CC and CXC chemokines), inflammatory cell recruitment (neutrophils, inflammatory macrophages) followed by increased expression of pro-fibrotic mediators (e.g., TGF- β 1, EGF, FGF) and fibrotic markers (e.g., fibronectin, collagen; Shenderov et al. 2021; doi: 10.1172/JCI143226).

c) Bleomycin-induced dysregulated tissue repair and inflammatory responses lead to epigenetic reprogramming, which at least in part responsible for fibrotic gene expression, the development of pulmonary fibrosis (Shenderov et al. 2021; doi: 10.1172/JCI143226; Brasier and Boldogh 2019; doi: 10.7573/dic.2019-8-3; Brasier 2018; doi: 10.1080/17476348.2018.1526677).

d) In addition, treatment of human cancers with bleomycin clinically associated with deterioration of lung function parameters ~10% of patients and a large portion of these patients acquire fibrotic lung disease (Hay, Shahzeidi, and Laurent 1991; doi: 10.1007/BF02034932; Brasier and Boldogh 2019; doi: 10.7573/dic.2019-8-3; Aguilera-Aguirre et al. 2015; doi: 10.1016/j.freeradbiomed.2015.07.007).

Importantly, our analysis shows a link between expression of OGG1 and SMAD (Figures 7 and 8 in revised manuscript).

e) The reviewer is correct in stating that it is clear that 8-oxoG is not sufficient to cause fibrosis. It is the cytotoxicity caused by bleomycin causing scarring and resulting in the fibrotic response. We thank the reviewer for pointing this out. In the revised version we include a section on that it is not 8-oxoG caused by bleomycin that causes fibrosis as potassium bromide is not inducing fibrosis, and we discuss the mechanism leading to lung damage. (L39) ‘This model reproduces several phenotypic features of human IPF, including peripheral alveolar septal thickening, dysregulated cytokine production, and immune cell influx resulting in fibrosis.’

We completely agree that the nomenclature ‘Ogg1-/-’ is misleading and has been changed throughout the manuscript. Similar efforts on which we based the siRNA administration included in this study have successfully been undertaken (Bacsi et al. 2013; doi: 10.1016/j.dnarep.2012.10.002; Haak et al. 2019; doi: 10.1126/scitranslmed.aau6296), before progressing to prophylactic treatment.

6) In the intratracheal administration of bleomycin or siRNAs, are there estimates for the depth of penetration in the lungs – if this is known, it would be useful to add to help in the understanding of the extent of lung involvement with these treatments. As mentioned above, the authors should have

used an *Ogg1*^{-/-} mouse as a control for these experiments – these data will serve as the proper controls from which to assess the role of TH5487.

We have previously conducted a pilot study to examine the extent of aerosol distribution using a methylene blue solution (Supp. Fig. 26). Visible distribution can be seen throughout the entire murine lung, indicating sufficient distribution of both the bleomycin and siRNA solutions (Fig. 6). Moreover, siRNA technology is extensively applied therapeutic approach for treatment of a variety of diseases that are incurable by conventional drugs (Hu et al. 2020; doi: 10.1038/s41392-020-0207-x). Studies from our group have documented that down-regulation of OGG1 expression by siRNA (administered intranasally) in the airways, significantly decreases allergic lung inflammation (Bacsi et al. 2013; doi: 10.1016/j.dnarep.2012.10.002). Previous studies by Wang, Y. *et al.* 2020 (doi: 10.1096/fj.201901291RRRRR) have additionally shown decreased bleomycin-induced pulmonary fibrosis using *Ogg1*^{-/-} mouse model.

Fig. 6. Murine lungs in two representative administrations of methylene blue solution

7) In Fig 3b, again there is no label on the x-axis and in the 2nd panel, the x-axis is shifted into the middle of the data set. Concerning these data, the extent of weight loss in the bleomycin-treated mice is disturbing since it exceeds the >20% allowable by all IACUC protocols that I am aware of. What was the criteria used for euthanizing the mice? The “Ethical Approval” section needs to be expanded.

We thank the reviewer for this comment and agree that the extent of weight loss initially appears disturbing, exceeding the allowable 20% cutoff. However, the mice were euthanized as soon as the weight crossed the 20% cutoff mark or if the mice displayed any signs of distress as listed in our ethical permit. The general humane endpoint is indicated by many signs including a loss of 20% of original mouse weight and changes in mouse behavior including (amended in revised manuscript; L338):

- discomfort or stress: (e.g. dull/staring coat (hair erect); decreased or increased activity; avoidance behaviour; isolation from the group; depressed; decreased appetite)

- deterioration (e.g. Discharge from eyes; loss of general condition; anorexia; dehydration (tenting skin/sunken eyes); weakness; decreased motility)
- distress: (e.g. very weak; unresponsive to touch; unconscious; convulsing; difficulty breathing)
- If obvious distress (bleeding trachea, labored breathing etc.) following intratracheal administration of compound/bleomycin is seen, the mice will be immediately euthanized.
- Mice that experience severe adverse reactions to the test compound or anaesthetic drug and are deemed to be in distress will be euthanised. Animals showing neurological signs, that have convulsions, demonstrate severe weakness, avoidance behaviour, display signs of depression, show respiratory distress, or open-mouthed breathing will humanely be euthanized prior to the end point.

The weight loss was exceptionally rapid in a few of the mice and these mice were euthanized immediately upon the next inspection. The weight loss percentages were then recorded as final experimental weights. If the animals are removed from the weight loss calculation, the following figure is obtained (Fig. 7, upper panel). With the inclusion of the repeat experiment samples, the manuscript has been amended to read (L111-114) "Mice in the bleomycin/vehicle group lost 17.98 ± 3.12 % of total bodyweight, whilst mice in remaining groups either maintained their bodyweight or gained significant weight compared to the bleomycin/vehicle group over the experiment timeline."

Original weight loss Figure:

Amended figure:

Fig. 7. Weight loss percentage

8) The size of the figure panels and labels in Fig 4 is too small – expanding it to >200% makes it barely readable – to aid in these data presentation, inclusion of only selective data is recommended and table(s) can report the essential data. In prior studies (Visnes et al 2018 Science), Cxcl2, IL6 and Tnf were described in detail as being modulated by TH5487 treatment. Of those, only IL6 is featured in these data – for ease of comparison with data presented in that prior publication, addition of Cxcl2 and Tnf should be included throughout.

We thank the reviewer for this comment. We agree that the figure panels were too small and these have been increased in size to allow easier readability. Although the reviewer makes a good point regarding the Visnes et al. study, the primary goal of this study was to assess fibrotic changes in the lung environment rather than inflammatory changes, negating the need for direct comparison. All tested cytokines can be found in Supplementary Figure 3, 4, 5. In addition, the fibrosis panels used in this study (Figure 4) allow comparisons between the different treatments to be made.

9) In Fig 5a, the rationale and purpose for inclusion of dexamethasone should be included. In this regard, the authors do not address the very large differences in neutrophil numbers between the bleomycin alone and the bleomycin + dex control treatment; this trend is also very evident in the inflammatory macrophage data in which (with the exception of 1 mouse) the + dex treatment seems more effective than the TH5487 treatment.

The reviewer makes an important point here. The reason for including dexamethasone was to include a positive control for anti-inflammatory effects which have been shown to reduce fibrotic damage (Wigenstam et al. 2018; doi: 10.1080/15563650.2018.1479527). However, despite the reduction in neutrophil and inflammatory macrophage numbers in the dexamethasone group there was a significant increase in fibrotic lung damage in comparison to the TH/bleomycin group. This indicates that TH5487 treatment reduces fibrotic damage by decreasing immune cell recruitment whilst importantly targeting other mechanisms (fibroblast transition, ROS, TGF- β 1/SMAD). This is shown in Fig. 1-8 in the revised manuscript.

10) Proteomic analyses are very solid. Inclusion of human tissue analyses, with comparative murine data is a very positive contribution.

We thank the reviewer for this comment and agree that both analyses support the use of TH5487 in reducing IPF.

Reviewer #3 (Remarks to the Author):

This is a well-designed and executed study, and a well-written manuscript. The findings should be of interest to the readership of Nature Communications. The value of the proteomics dataset should also be of high value to the community. I recommend publication after the following issues are addressed.

We appreciate the assessment by the reviewer and agree that the proteomic data set will be of great relevance.

- The authors report a decrease in the level of OGG1 protein upon inhibitor treatment. The reported inhibitor acts by blocking the catalytic action of OGG1: how do, then, the authors explain the observed decrease in OGG1 protein level? This question needs to be addressed in more detail. Stated differently, if the only action of the inhibitor was to interfere with the catalytic action of OGG1 as an enzyme, one would not expect to observe a decrease in OGG1 levels. Could it be that the OGG1-inhibitor complex formation somehow tags OGG1 for degradation or is this phenomenon strictly an artifact of the IPF model system that utilizes bleomycin treatment? Have the authors examined the effect of inhibitor treatment on OGG1 levels in a system different from the IPF/bleomycin model?

Thank you for this important comment. As we described in our response to Reviewer 2, currently, we do not have direct evidence to show “that the OGG1-inhibitor complex formation somehow tags OGG1 for degradation.” To elucidate the effects of TH5487 on OGG1, we first show that TGF- β 1 increases in *Ogg1* mRNA levels (Fig. 5a, b) and protein level (Fig. 5c). Note: mRNA levels were determined by qRT-PCR of Exon 3-4. These data are in line with increased expression of OGG1 in fibrotic human lungs and bleomycin-induced experimental fibrosis in mouse model.

TH5487 had no effect on *Ogg1* mRNA levels (Fig. 5b); however, in multiple experiments TH5487 significantly lowered protein level (Fig. 5c, data of a representative experiment is shown). These results suggest that TH5487 not only inhibit DNA substrate binding of OGG1 but may subjects it to degradation in TGF- β 1-exposed cells as suggested by the reviewer. This is shown in Suppl. Fig. 1e, f, g and also mentioned on lines 95-99 in the revised manuscript.

Fig. 5. Effect of TH5487 on *Ogg1* expression in TGF- β 1 \pm TH5487-treated *Ogg1*^{+/+} MEF cells. (a) Time course analysis of *Ogg1* expression in mock- and TGF- β 1 (10 ng/mL)-treated *Ogg1*^{+/+} MEFs. (b) There are no effects of TH5487 (10 μ M) on *Ogg1* expression at mRNA levels. (c) Western blot analysis of *Ogg1* in *Ogg1*^{+/+} MEF cells. A representative Western blot is shown.

The use of the word "novel" in reference to the inhibitor itself is not warranted because the inhibitor was published 3 years ago in Science. The present study and its findings are novel but the inhibitor itself is not.

We acknowledge that the use of 'novel' may not be correct as related to the inhibitor itself. This has been amended throughout the manuscript.

- Line 38: did the authors mean to use the word "resulting" instead of "resolving"?

This change has been made.

- Lines 61 and 62 (and perhaps elsewhere): replace the Latin lowercase "u" with Greek mu symbol to denote micromolar.

L61/62 have been corrected.

- Line 333: the correct city name is Manassas.

This change has been made.

Reviewer #4 (Remarks to the Author):

I have been asked to evaluate the proteomics aspects of this manuscript....

The manuscript describes an evaluation of a new small-molecule inhibitor of OGG1, TH5487, for treating inflammation and fibrosis in the lung. The proteomic aspects of the manuscript investigate changes in protein expression after TH5487 treatment and the conclusion is that the changes observed are linked to known phenotypes of fibrosis.

My biggest concern with the proteomics is the number of animals in each treatment group. I could not actually find an explicit statement of how many were used but from the number of columns in Fig. 6C it appears that N=5 and N=4 (depending on condition) were used. Since each replicate was a different animal, these N values seem exceptionally low. I understand that a lot of animals were used overall but still, the low N makes the robustness of many of the changes suspect.

We thank the reviewer for these comments. To address these, we now include additional pointers to the N numbers employed in proteomics analysis (in main text, methods, figure legends). We would like to highlight that the main goal of the proteomics module was to obtain a broad, top-level view of strong changes to lung and BALF proteomes incurred by TH5487 treatment (BTH) relative to the disease model (B) which was therefore prioritized with 5 animals sacrificed for this main task.

We would also like to highlight that the biological inter-individual variability within one treatment group by far exceeds the technical variability of DIA-MS measurements (see, e.g. Bruderer et al. 2017; doi: 10.1074/mcp.RA117.000314) on which our proteomics pipeline is based) and we opted for single-shot

analysis per animal and analysis of differences between groups of animals, rather than technical replication per animal.

A secondary goal was to compare proteome responses across multiple treatment conditions to identify modules/clusters of co-responding proteins that are thus implicated as being of general relevance in this system and disease setting. The additional conditions were sampled with 4 animals per group. We want to highlight just one recent publication where N=4 mice were used for one of the comparison groups (Schaum et al. 2020; doi.org/10.1038/s41586-020-2499-y). We highlighted all N numbers underlying the presented comparisons. Robustness is evaluated based on statistics as indicated. A further increase of N numbers could lead to more sensitive recovery (statistical significance) of more subtle changes, a consideration of statistical power that needs to be balanced with ethical considerations on animal use in research. However, at the presented replication scheme, we observe significant changes in the Lung and BALF proteomes. A detailed report of the data analysis procedure leading to these results was generated and is now included in the supplement. Additionally, clarification on specific proteins of interest which did not reach statistical significance have been highlighted in the main text (L186) 'Furthermore, proteins of interest that appeared downregulated following TH5487 treatment included tenascin-C (TNC; p value: 0.0374), carbamoyl phosphate synthetase 1 (CPS1; 0.03978), matrix metalloproteinase 19 (MMP19; 0.0402), tissue inhibitor of metalloproteinase 1 (TIMP1; 0.0460), Janus kinase 3 (JAK3; 0.0301), cAMP-dependent protein kinase catalytic subunit (PRKX; 0.0152), lysyl oxidase homolog 2 (LOXL2; 0.0208), whereas these changes did not attain statistical significance in light of strict multiple testing correction and limited statistical power constrained by cohort size and biological and technical variability.'

Apart from this, the rest of the statistics and data treatment seem in-line with expectations in the field. The one other gap, however, is that none of the custom scripts (e.g., that described in lines 563-565) or R code are available so evaluating much of what has been done here is difficult.'

We thank the reviewer for this comment and have included the custom scripts and R code. The following changes have also been made in the main text (L621-627) 'Mass spectrometry proteomics data and initial search results have been deposited to the ProteomeXchange Consortium (<http://proteomecentral.proteomexchange.org>) via the PRIDE partner repository [1] with the dataset identifier PXD029625 (Username: reviewer_pxd029625@ebi.ac.uk ; Password: 5YIGmBWu ; will be made public upon acceptance of the manuscript). Downstream analysis R code is available at <https://github.com/heuselm/DiffTestR/tree/Tanner2021>. All remaining data are available in the main text or the supplementary materials.'

Minor points:

This statement seems inappropriate: "To our knowledge, this represents the deepest single-shot proteomic map of murine lung and BALF proteomes reported to date". For one thing this is not even a true statement since those numbers reported are from the combination of MANY experiments, not a single shot. And each of the 9,000+ proteins was not detected in each sample. I would just remove it

because it does not add anything here.

We removed this statement and added a different sentence, specifying the numbers of protein groups quantified per each animal/analysis run in order to present the data more clearly. (L173) 'A total of 9,872 protein groups were identified across lung and BALF samples, with 5,629 protein groups observed exclusively from lung (57 %), 495 protein groups identified exclusively in BALF and 3,748 protein groups observed across both compartments (Fig. 6B). On average, $7,682 \pm 1074$ protein groups were identified per DIA-MS run of lung homogenates and 3024 ± 1083 of bronchoalveolar lavage samples, forming a comprehensive basis for the evaluation of proteomic alterations incurred by the investigated treatment schemes (compare Fig. 6B).'

Line 539: should be 6-8 POINTS, not peaks

Line 554: 10.121 should be 10,121

Thank you, both changes have been made in the revised version of the manuscript.

REVIEWER COMMENTS

Reviewer #1 (Remarks to the Author):

The authors have undertaken considerable in vitro and in vivo work to try and improve their manuscript. They have added experiments using primary SAEC and also further murine experiments which have added some value to the manuscript unfortunately I remain underwhelmed by the results despite the obvious hard work put in by the authors.

The concentration of TGFbeta required to induce these effects is far too high and I suspect has nothing to do with the direct effects of TGFbeta signalling. Looking at figure 2 the readout for the TGFbeta concentration responses was at 4 DAYS. The mRNA response following TGFbeta would be expected to be maximal at 24 hours, and the lack of response to 2ng/ml is very concerning and I suspect they have just missed the peak. This unfortunately calls into question much of the in vitro data showed.

The in vitro data are improved and I am reassured by the authors response to my statistical concerns. However, if the authors are 'well aware' of the ARRIVE guidelines why are they not cited, and why are the power calculations included in the methods?

I also particularly like the whole lung micrographs which greatly reassure me that the authors have analysed the histology appropriately although the images are far too low in resolution for me to check.

Could the authors reassure me that the in vivo experiments including Pirfenidone and Nintedanib were performed at the same time as the experiments using TH5487 and siRNA rather than subsequently and plotted on the same graph as experiments performed at a different time?

Finally the weight loss curves in Figure 2b are unusual and confusing. Characteristically the weight loss observed by mice following intratracheal bleomycin is in the first seven days and then starts to recover by day 10-14 consistent with bleomycin induced lung injury followed by resolution of acute lung injury. However, in figure 2b the weight loss only starts after day 5 and is biased towards animals receiving no treatment or OGG1 siRNA (although before they received any therapy) suggesting that these mice had worse lung injury before administration of 'therapy' and therefore confounding the results. What is the explanation for the absence of lung injury in the mice receiving intratracheal bleomycin?

Why were no hydroxyproline experiments done for the prophylactic experiments comparing TH5487 with Deaxmethasone?

Finally I do take expectation to the final statement in the introduction "Final statement "Taken together, 72 these data strongly suggest that TH5487 is a potent, specific, and clinically-relevant treatment for 73 IPF." This is very much overselling their data, at best, if the data are replicated and the concerns answered this would be a reason to investigate further but there is nothing in these data that suggest it is 'clinically relevant for IPF'.

Reviewer #2 (Remarks to the Author):

The revised manuscript by Tanner et al represents a very substantial effort to address reviewers' concerns and recommendations, with the addition of significant new data, and clarification of the experimental rationale. As written, it will make a significant contribution to understanding the key regulatory role that OGG1 plays in modulating the magnitude of inflammatory cascades and will hopefully further bolster the development of clinically germane OGG1 inhibitors.

If accepted, as the manuscript proceeds toward a galley proof stage, the following improvements should be made:

1. abbreviations are sometimes repeated as reading through the manuscript

2. the superscripting of genotypes is variable
3. there are typos in the current version
4. the source of the TH5487 was not given - presumably it was from the Helleday lab, but if others wish to repeat or amplify this work, there should be a supplier listed for the compound.
5. the order of presentation of the subheadings in the Methods section should be organized with the order of presentation within the Results section - this would simply facilitate the reader to look at methodologies in a sequential order rather than scrolling through the entire methods looking for a method.

Reviewer #3 (Remarks to the Author):

The authors have addressed the Reviewers' questions/comments adequately. I recommend publication.

Reviewer #4 (Remarks to the Author):

The authors have addressed my concerns satisfactorily

Reviewer #1

The authors have undertaken considerable in vitro and in vivo work to try and improve their manuscript. They have added experiments using primary SAEC and also further murine experiments which have added some value to the manuscript unfortunately I remain underwhelmed by the results despite the obvious hard work put in by the authors.

We thank the reviewer for the acknowledgement of the hard work put into addressing these concerns.

The concentration of TGFbeta required to induce these effects is far too high and I suspect has nothing to do with the direct effects of TGFbeta signalling. Looking at figure 2 the readout for the TGFbeta concentration responses was at 4 DAYS. The mRNA response following TGFbeta would be expected to be maximal at 24 hours, and the lack of response to 2ng/ml is very concerning and I suspect they have just missed the peak. This unfortunately calls into question much of the in vitro data showed.

Response:

We are very appreciative of the Reviewer's comments and the valuable suggestions communicated with us in the first and second review. We have provided data showing functional inactivation of OGG1 by TH5487, that significantly decreases the effect of TGF- β 1 in terms of cell transformation and cell migration using various cell types. We agree with the Reviewer and have performed experiments to determine the effect of lower TGF- β 1 concentrations in terms of signaling (assessed by SMAD phosphorylation, receptor saturation) and transcriptional gene activation. In addition, as we described in our previous reply, it appears that more complex phenotypic changes (e.g., cell migration, stable EMT II phenotype) require higher TGF- β 1 dose as documented by publications (Jiang and George 2011, Tian, Zhao et al. 2016, Tian, Widen et al. 2018, Xu, Zheng et al. 2018, Kim, Ahn et al. 2020, Doolin, Smith et al. 2021).

In addition, we apologize for lack of sufficient data and brief information we provided in the legend to Supplementary Figure 1a and have updated it in the revised Supplementary Materials. In brief, results show that OGG1 proficient *Ogg1*^{+/+} MF cells, responded to acute TGF- β 1 (2 ng/ml) treatment as expected from data published previously (Vizan, Miller et al. 2013, Miller, Bloxham et al. 2018). However, *Ogg1*^{-/-} cells only responded to TGF- β 1 concentrations higher than 2 ng/mL. Please see **Fig. 1** and Supplementary **Figure 1a**.

Summary of data addressing Reviewer' comments

As advised by **Reviewer #1**, we have repeated and extended multiple experiments, to provide additional rigor to the studies in this submission. Specifically, we have performed TGF- β 1 dose response experiments (SMAD phosphorylation, gene expression and migration assays), using primary human fibroblast (pHLF), human small airway epithelial cells (hSAEC) and spontaneously immortalized isogenic pairs of *Ogg1*^{+/+}, *Ogg1*^{-/-} mouse fibroblasts (MF). Results are summarized as follows:

- 1.) TGF- β 1 has been shown to bind to its receptor (T β R), with Smad2/3 being phosphorylated by T β RI, where after it is translocated into the nucleus. Smad2/3 then initiates the transcription of alveolar epithelial-mesenchymal transition (EMT) and fibrosis-associated genes. To confirm the role of TGF- β 1 concentration dependence in SMAD3 phosphorylation, we conducted experiments using *Ogg1*^{+/+} and *Ogg1*^{-/-} MF cells and tested whether there were differences in response driven by the addition of TGF- β 1. Results are displayed in **Fig. 1 (Supplementary Fig 1c)**. No differences were

seen in TGF- β 1 signaling as assessed by **SMAD phosphorylation** between *Ogg1*^{+/+}, and *Ogg1*^{-/-} MF cells (**Fig.1**) and between our cell culture models (**Fig. 2**). Specifically, levels of p-SMAD3 were similar after TGF- β 1 (2, 5 and 10 ng/mL) exposure in all cell types (**Fig. 1** and **Fig. 2**). In addition, TH5487 had no effect on levels of **p-SMAD3** (**Fig. 2**) in hSAEC, *Ogg1*^{+/+}, and *Ogg1*^{-/-} MFs after exposure to 2 ng/mL of TGF- β 1.

Figure 1. Phosphorylation of SMAD3 in response to acute TGF- β 1 treatment of *Ogg1*^{+/+} and *Ogg1*^{-/-} cells.

Cell cultures were grown to ~70% confluency, kept in 0.5% FBS-containing medium for 24h and were exposed to TGF- β 1 or vehicle (one representative experiment). At 1h after TGF- β 1 addition, cells were lysed and levels of p-SMAD3 (Ser423/425) were determined by Western immunoblot analysis (antibody: Cat # 9520, Cell Signaling Technologies) essentially as described previously (Nicolas and Hill 2003, Vizan, Miller et al. 2013, Miller, Bloxham et al. 2018). Membranes were stripped and re-probed using SMAD3 (C67H9) antibody (Cat #: 9523, Cell Signaling Technologies). These data show no differences in levels of SMAD3 phosphorylation in response to 2 and 10 ng/mL of TGF- β 1. Kinetic changes in SMAD3 phosphorylation are well-documented showing that SMAD2/3 phosphorylation is maximized within 1h, and thereafter its level decreases over time but remains at a low steady state level as long as TGF- β 1 ligand is available (Vizan, Miller et al. 2013, Miller, Bloxham et al. 2018).

Figure 2. Effect of TH5487 on TGF- β 1 signaling as assessed by SMAD3 phosphorylation. hSAECs, *Ogg1*^{+/+} and *Ogg1*^{-/-} MF cells (as controls) were grown to 80% confluence in corresponding medium and cells were mock-treated (vehicle) or 2 ng/mL TGF- β 1 was added for 1 h. Cells were lysed and levels of p-SMAD3 were determined by Western immunoblot analysis (as described in legend of Fig. 1). These data are included in **Supplementary Figure 1e**. Data presented from one representative experiment.

- 2.) To determine whether cell-type specific differences in TGF- β 1-induced gene expression exist we treated cells with TGF- β 1 (2 ng/mL) and measured gene expression of two profibrotic genes (fold increases) in *Ogg1*^{+/+} MF cells, HaCat and PC3 cells. Results are summarized in **Fig. 3**, showing that

TGF- β 1 (2 ng/mL) induced significantly higher increases in RNA levels of *FN* and *COL1A1* in PC3 and HaCaT cells, compared to *Ogg1*^{+/+} MF. These results are shown in **Supplementary Fig. 1b** of the revised manuscript. These differences can partially be explained by constitutive levels of gene expression. In mock-treated HacaT and PC-3 cells basal expression of *COL1A1* and *FN1* were approximately 50% lower compared to those in *Ogg1*^{+/+} MF (data not shown). These observations are in line with those documented previously (constitutively high expression of *Vim* in fibroblasts) (Ostrowska-Podhorodecka, Ding et al. 2022). There are relatively high levels of expression of *Col1a1*, in fibroblasts compared to epithelial cells (Wei, Nguyen et al. 2021). Although it requires further investigation, these data suggest cell type-dependent (fibroblast vs. epithelial) expression of TGF- β 1-induced genes.

Figure 3. TGF- β 1-induced expression of *FN1* and *COL1A1* in *Ogg1*^{+/+} MF, PC3 and HaCat cells. Parallel cultures of cells were grown to ~70% confluency (~2 days), kept in 0.5% FBS-containing medium for 24h and were mock or TGF- β 1-treated with 2 ng/mL. Changes in mRNA levels were determined by qRT-PCR. Single experiment with 3 parallel replicates. Statistical comparisons were conducted using Student's t-test.

- Next, we aimed to elucidate whether TGF- β 1 concentration influences the expression of mRNA levels of *α -Sma*, *Vim*, *Fn1*, and *Col1A1* in isogenic pairs of *Ogg1*^{+/+}, *Ogg1*^{-/-} MF cells. Treatment with TGF- β 1 at a dose of 2 ng/mL induced significant increases in mRNA levels of *Fn1*, *Col1a1*, and *aSma*, with the exception of *Vim* in *Ogg1*^{+/+} MF. There were no significant differences between 5 ng/mL and 10 ng/mL TGF- β 1-induced gene expression in *Ogg1*^{+/+} MF. In contrast, *Ogg1*^{-/-} MF cells responded significantly to higher TGF- β 1 concentrations (**Fig. 4** and **Supplementary Fig. 1a**). These results were expected due to role of OGG1 in gene expression and airway remodeling (Pan, Wang et al. 2019, Pezone, Taddei et al. 2020, Wang, Chen et al. 2020). Because 10 ng/mL of TGF- β 1 induced significant increases in expression of all four genes in *Ogg1*^{-/-} MF cells, we utilized this concentration for wound healing and mRNA analysis in the main manuscript. Although MFs are different from comparator cells (e.g., HaCat or Colo-357) utilized in previous studies (Nicolas and Hill 2003, Vizan, Miller et al. 2013, Miller, Bloxham et al. 2018) the results obtained are similar (significant increases in mRNA levels occur at or after 24 h of TGF- β 1 treatment).

Figure 4. RNA expression levels of α -Sma, Col1A1, Fn1 and Vim in $Ogg1^{+/+}$ and $Ogg1^{-/-}$ MF cells in response to increasing doses of TGF- β 1. Parallel cultures of cells were grown to ~70% confluency (2 days), kept in 0.5% FBS-containing medium for 24h and were mock or TGF- β 1 treated (2, 5, 10 ng/mL). Twenty-four h post-treatment mRNA levels were determined by qRT-PCR. n = 3. Statistical comparisons were conducted using one-way ANOVA with Dunnet's post-hoc analysis.

Figure 5. Inhibition of OGG1 in $Ogg1^{+/+}$ MF cells decreases expression of α -Sma, Col1a1, Fn1 and Vim. Parallel cultures of $Ogg1^{+/+}$ and $Ogg1^{-/-}$ MF cells were grown to ~70% confluency (~2 days), starved for 24h. 2h prior to TGF- β 1 (2 ng/mL) addition TH5487 (10 μ M) was added. TH5487 was replaced by changing medium at 12h for spent medium containing 10 μ M fresh TH5487. Parallel cultures of cells were mock treated with equivalent volume of solvent (DMSO). Twenty-four h post-treatment with TGF- β 1, mRNA levels were determined by qRT-PCR (n=2 independent experimental replicates, with 2 technical replicates in each experiment).

Furthermore, we wanted to determine the role of TH5487 in suppressing profibrotic gene expression at 2 ng/mL TGF- β 1. We performed experiments using *Ogg1*^{+/+} and *Ogg1*^{-/-} MF cells treated with the highly specific OGG1 inhibitor TH5487 (Visnes, Cazares-Korner et al. 2018). Results show that TH5487 significantly decreased gene expression *Ogg1*^{+/+} MF cells (**Fig. 5a**). To provide evidence for specificity, we show that TH5487 had no significant effect on mRNA levels in *Ogg1*^{-/-} MF cells (**Fig. 5b**). This figure is included in **Supplementary Fig. 1f**. *Ogg1*^{-/-} and *Ogg1*^{+/+} MF cells are isogenic thus one of the most likely reasons for relative low level of gene expression (or lack of expression, see **Fig. 1**) is lack of OGG1 in *Ogg1*^{-/-} MF cells.

Figure 6. TGF- β 1-induced expression of *VIM* and *FN* at mRNA level. Parallel cultures of cells were grown to ~80% confluency (~3 days) in small airway epithelial cell serum-free growth medium (SAECGM, for hSAEC). At day 3, increasing concentration (2, 5 and 10 ng per mL) of TGF- β 1 was added for 24h. Changes in mRNA levels were determined by qRT-PCR. One experiment with 3 independent replicates. Statistical comparisons were conducted using one-way ANOVA with Dunnet's post-hoc analysis.

Finally, to see whether this trend was upheld in a different IPF-relevant cell type, hSAEC cells were TGF- β 1 (2, 5 or 10 ng/mL) treated and expression from *FN1* and *COL1A1* were determined by qRT-PCR. Similarly, the expression of *FN1* and *COL1A1* were significantly increased at TGF- β 1 concentrations above 2 ng/mL.

Data presented in **Fig. 4-6**, suggest that inhibition of OGG1 by TH5487 in *Ogg1*^{+/+} MF or absence of OGG1 protein in *Ogg1*^{-/-} MF resulted in low (or lack of response) gene expression induced by TGF- β 1. These data are supported by chromatin immunoprecipitation (ChIP) assays, showing significantly decreased binding of SMAD3 to promoter sequences in *Ogg1*^{-/-} MF cells or when cells are treated with the OGG1 inhibitor TH5487 (**Fig. 1g** of the main manuscript).

These results may be explained by lack of OGG1 protein that is needed for binding of SMAD complex(es) to DNA. It has been shown that affinity of SMADs for DNA is low (Kd ~ 1x 10⁷ M) (Shi, Wang et al. 1998). Therefore, specific SMAD-DNA interactions require cooperation with other sequence-specific transacting factors that promote their interaction with DNA (Feng and Derynck 2005, Ross and Hill 2008, Hill 2016). Facilitation of SMAD-DNA interactions by OGG1 may be due to the induced DNA structural changes at initial steps of DNA base excision repair (Bruner, Norman et al. 2000).

We note: OGG1 is a multifunctional protein that initiates DNA base excision pathways by excising the oxidatively modified guanine, 8-oxoguanine (David, O'Shea et al. 2007, Krokan and Bjoras 2013). Under

stress conditions (e.g., cytokine-receptor interaction), post-translationally modified OGG1 functions as transcriptional modulator of gene expression (Ba and Boldogh 2018, Pezone, Taddei et al. 2020). Studies have also identified 8-oxoguanine as an epigenetic-like mark (Fleming, Ding et al. 2017).

- 4.) Reviewer 1 suggested studying the effect of lower TGF- β 1 concentrations on directional cell migration. We tested varying concentrations of TGF- β 1 (0, 2, 5, and 10 ng/mL) in a wound healing assay using hSAEC cells (data included in **Supplementary Fig. 2 c**).

Figure 7. TGF- β 1 concentration-dependent directional migration of hSAECs. Cells were grown in standard serum-free basal medium. At 100% confluence cells were starved in basal medium (without supplements) for 24h and then monolayer scratched with a 200 μ L pipette tip, washed and basal SAECGM was added containing TGF- β 1 (2, 5 or 10 ng/mL). Cell cultures were photographed, and wound size analyzed. Representative images are shown at 0h and 48h. Data are presented from 2 experimental replicates with 3 technical replicates in each. Statistical comparisons were conducted using one-way ANOVA with Dunnet's post-hoc analysis.

Note: Using our cell culture models 5 ng/ mL, or 10 ng/ml TGF- β 1 was needed for significant changes. Our data are in line with those studies, using MCF-7 and MDA-MB-435S (human breast cancer cells), A549, H292, H226, and H460 (lung cancer cell lines) (Lv, Kong et al. 2013, Kim, Ahn et al. 2020). These data together show that cell migration as determined by wound healing (or those e.g., matrigel invasion, sphere formation, trans-well migration) requires higher TGF- β 1 concentration, compared to SMAD phosphorylation.

TGF- β 1 concentration-dependent cell migration as determined by “wound-healing” assays \pm inhibitor. Given that *Ogg1*^{-/-} MEF cells did not respond to lower (2 ng/mL) and partially responded to 5 ng/mL TGF- β 1 (**Fig. 1, Fig. 4**), these experiments were performed by using 10 ng/mL TGF- β 1 as in the original manuscript. Data summarized in **Fig. 8 (Supplementary Fig. 2 a,b)**, clearly show that the OGG1 inhibitor TH5487 is effective as Nintedanib (inhibits early events in TGF- β signaling (Wollin, Wex et al. 2015)., left and right panels).

Figure 8. TGF- β 1-induced directional migration of *Ogg1*^{+/+} and *Ogg1*^{-/-} cells as determined by wound healing assays. Parallel cultures of 24-well plates were grown to 100% confluence in complete medium, starved in basal medium (without FBS) for 24h and then wounds were made using 200 μ L pipette tip, monolayers were washed and then mock-treated or 10 ng/mL TGF- β 1 added \pm inhibitor(s), TH5487 (10 μ M); Nintedanib (10 μ M). Wound areas were photographed at 0, 14, 28h. Images were analyzed using the wound healing tool in ImageJ (<https://imagej.nih.gov/ij/>). Data are presented as percentages of the initial wound area. In the figure we have manually contrasted edges. Data are presented from 2 experimental replicates with 3 technical replicates in each. Statistical comparisons were conducted using one-way ANOVA with Dunnet’s post-hoc analysis.

Additionally, we assessed whether pHLF cells treated with 2 ng/mL TGF- β 1 induced some degree of wound healing. Despite inhibition of wound healing by TH5487 and nintedanib compared to the TGF- β 1 control, the results between all 5 conditions were not statistically significant (Fig. 9). This points to the requirement for TGF- β 1 concentrations greater than 2 ng/mL and we have therefore utilized 10 ng/mL for the main manuscript (Fig. 1b, c).

Figure 9. Wound healing assay of pHLF cells treated with TGF- β 1 (2 ng/mL) for 24h. pHLF cells were grown to confluence and then scratched to form a wound. The cells were washed and treated with TGF- β 1 (2 ng/mL) for 24h. Cells were treated with dexamethasone (10 μ M), nintedanib (10 μ M), and TH5487 (10 μ M). Wound areas were photographed at 0, and 24h using an Olympus CKX41 microscope with Olympus SC30 camera and cellSens Entry software (Olympus, Tokyo, Japan). Images were analyzed using the wound healing tool in ImageJ (<https://imagej.nih.gov/ij/>). Data are presented as percentages of the initial wound area. Data are presented from 3 experimental replicates with 3 technical replicates in each. Statistical comparisons were conducted using one-way ANOVA with Dunnet’s post-hoc analysis.

5.) To test whether TGF- β 1 concentrations influenced transwell migration, pHLF cells were exposed to varying concentrations of TGF- β 1 (0, 0.5, 2, 5 and 10 ng/mL). pHLF cells displayed significant increases in migration following 2, 5, and 10 ng/mL with 10 ng/mL achieving the greatest number of migrated cells (Fig. 10). Following a significant response to 2 ng/mL for the transwell migration assays, pHLF cells were treated with 2 ng/mL of TGF- β 1, along with drug-treatment (Fig. 11) and

Ogg1 siRNA targeting (**Fig. 12**) perturbations and monitored for 24h. Significant differences were seen between control TGF- β 1 only and TGF- β 1/TH5487 cells and between TGF- β 1 control and Ogg1 siRNA knockdown cells.

Figure 10. TGF- β 1 concentration-dependent effects on transwell migration of pHLF cells. Treatment of pHLF cells with increasing concentrations of TGF- β 1 (0, 0.5, 2, 5 and 10 ng/mL) induced significantly different numbers of migrated cells. There were no significant differences between 0 and 0.5 ng/mL, with significantly increased numbers of cells seen for 2 ng/mL, 5 ng/mL, and 10 ng/mL. As suggested by Reviewer 1, there was a TGF- β 1 concentration-driven effect. Therefore, we have repeated the *in vitro* transwell assays at 2 ng/mL in subsequent transwell assays. Data are presented from 4 experimental replicates with 3 technical replicates in each. Statistical comparisons were conducted using one-way ANOVA with Dunnet's post-hoc analysis.

Figure 11. Transwell assays conducted using TGF-β1 (2 ng/mL) for 24h- drug treatment. Transwell assays were conducted using pHLF cells treated with 2 ng/mL TGF-β1 over 24h. Cells were treated with dexamethasone (10 μM), Nintedanib (10 μM), and TH5487 (10 μM). Dexamethasone failed to suppress pHLF migration with no statistical differences seen when compared to the TGF-β1 control. Statistically significant reductions in transwell migration were reported for both Nintedanib and TH5487, with both treatments achieving similar levels of migratory inhibition. This has been included in the revised manuscript (**Fig. 1a**). Data are presented from 4 experimental replicates with 3 technical replicates in each. Statistical comparisons were conducted using one-way ANOVA with Dunnet’s post-hoc analysis.

Figure 12. Transwell assays conducted using TGF- β 1 (2 ng/mL) for 24h- siRNA treatment. To confirm the role of OGG1 in migration, ablation of *OGG1* was performed using scramble control or targeting smart-pool siRNA respectively against hOGG1, with both transfected into pHLF cells using Lipofectamine 3000 according to the manufacturer's instructions. Transwell assays were then conducted using pHLF cells treated with 2 ng/mL TGF- β 1 over 24h. OGG1 competent cells failed to suppress pHLF migration when compared to the siOGG1/TGF- β 1 samples. No differences were seen in migratory capacities in the vehicle-treated control samples indicative of the TGF- β 1-dependent migration of the pHLF cells which is certainly influenced by OGG1 expression. Data are presented from 4 experimental replicates with 3 technical replicates in each. Statistical comparisons were conducted using one-way ANOVA with Dunnet's post-hoc analysis.

To confirm whether protein expression was affected using 2 ng/mL of TGF- β 1 addition, Western blots were conducted following *Ogg1*-targetting siRNA administration to pHLF cells (24 h). *Ogg1* knockdown significantly attenuated fibrotic responses as measured by COL1A1, α -SMA, VIM, and FN protein expression (**Fig. 13**).

Figure 13. Western blotting of pHLF cells treated with TGF- β 1 (2 ng/mL) for 24h. To further support the role of OGG1 in the production of profibrotic proteins, we treated pHLF cells transiently transfected with *OGG1*-targetting siRNA and siRNA control. These cells were then exposed to 2 ng/mL of TGF- β 1 for 24h. Cells were then lysed and assessed by immunoblotting for alpha-smooth muscle actin (α -SMA), collagen 1A1 (COL1A1), vimentin (VIMENTIN), fibronectin (FN), and OGG1. Densitometry analyses were conducted, with data normalized to the loading control (GAPDH). Data are presented from 3 experimental replicates with 3 technical replicates in each (samples are pooled from each experiment). Statistical comparisons were conducted using one-way ANOVA with Dunnet's post-hoc analysis.

6.) Next, we further confirmed the expression of various fibrotic protein markers. Fluorescence microscopy confirmed decreased expression of COL1A1, α -SMA, FN, VIM, and OGG1 expression in TH5487-treated pHLF cells compared to those treated with TGF- β 1 (2 ng/mL for 24h; **Fig. 12 and 13**). Furthermore, cells treated with Nintedanib (10 μ M) showed decreased expression of COL1A1, VIM, and FN, with FN displaying some staining in the distal regions of cells. Conversely, dexamethasone (10 μ M) treatment failed to show a decrease in any of the proteins tested indicating that the effects of TH5487 are unlikely associated with anti-inflammatory activity. Finally, OGG1 protein levels were significantly decreased by the addition of TH5487 in comparison to the TGF- β 1 and vehicle control samples. This highlights the ability of TH5487 to specifically target the production of OGG1 within human fibroblast cells.

Figure 12) Immunofluorescent staining of FMT markers conducted on pHLF cells treated with TGF- β 1 (2 ng/mL) for 24h. To obtain a visual expression of selected profibrotic markers, we treated pHLF cells with TGF- β 1 (2ng/mL) for 24h. Cells were treated with dexamethasone (10 μ M), Nintedanib (10 μ M), and TH5487 (10 μ M). Increased expression of COL1A1, α -SMA, FN, and VIM was seen in TGF- β 1 control and dexamethasone-treated cells (TGF- β 1/Dex). Comparatively decreased expression of these markers was seen in TH5487, Nintedanib, and vehicle-treated samples. These data have been included in the revised manuscript (**Fig. 1d**). Data are presented from 1 experimental replicate with 3 technical replicates in each.

Comment: The in vitro data are improved and I am reassured by the authors response to my statistical concerns. However, if the authors are 'well aware' of the ARRIVE guidelines why are they not cited, and why are the power calculations included in the methods?

We have updated the manuscript to cite the ARRIVE guidelines (L338-340) 'Sample sizes were calculated by power analysis (G Power Software) based on previous pilot studies, feasibility, and to conform to the ARRIVE guidelines⁷⁸.'

Comment: I also particularly like the whole lung micrographs which greatly reassure me that the authors have analyzed the histology appropriately although the images are far too low in resolution for me to check.

The images have been included in greater resolution below with particular emphasis placed on increasing the size of the lungs in the Bleomycin and Bleo/TH5487-treated groups. These data have been updated in the Supplementary Materials (Supp. Fig. 7 and 8).

In addition, the workflow used to analyze samples has been included below (bottom panel). Morphometric analysis of the sections stained with H&E and picosirius red were captured by a ScanScope Aperio scanner, using a 20X objective. For each lung section, the medium-sized arteries were deleted manually before analysis (1). Automated algorithm tuning was utilized to determine positive areas of each section (2). This algorithm macro was then utilized throughout the batch in order to ensure consistent analysis (3). Data is presented as a ratio of positive to negative areas of the lung section.

Comment: Could the authors reassure me that the *in vivo* experiments including Pirfenidone and Nintedanib were performed at the same time as the experiments using TH5487 and siRNA rather than subsequently and plotted on the same graph as experiments performed at a different time?

In response to Reviewer #1, we state that experiments were conducted including the following groups: (i) Bleo/TH5487; (ii) Bleo/Pirfenidone; (iii) Bleo/Nintedanib; (iv) Bleo only; and (v) Bleo/Ogg1 siRNA. These experiments were then added to the existing data. In order to ensure reproducibility within these experiments, multiple experiments were conducted with appropriate control groups included as has been undertaken herein.

Comment: Finally the weight loss curves in Figure 2b are unusual and confusing. Characteristically the weight loss observed by mice following intratracheal bleomycin is in the first seven days and then starts to recover by day 10-14 consistent with bleomycin induced lung injury followed by resolution of acute lung injury. However, in figure 2b the weight loss only starts after day 5 and is biased towards animals receiving no treatment or OGG1 siRNA (although before they received any therapy) suggesting that these mice had worse lung injury before administration of ‘therapy’ and therefore confounding the results. What is the explanation for the absence of lung injury in the mice receiving intratracheal bleomycin?

The reviewer is correct that characteristically, weight loss is observed within 7-10 days followed by resolution of acute lung injury after 14 days. We note that there were differences in starting weight of mice; with percentage weight loss seen in all mice treated with bleomycin from day 2-3 (Fig. 2b middle panel). Please see Fig. 3b, which also shows weight loss from day 2 onwards in the bleomycin treated groups, with mice showing varying degrees of recovery following treatment.

Variable dose-dependent responses to bleomycin lung damage have been reported in several publications varying from no weight loss to significant weight loss after 7-14 days (Cowley et al., 2019, *Comp Med.* 69(2): 95–102; Manali et al., 2011, *BMC Pulm. Med* 11, Art # 33; Ravanetti et al., 2020 *Am J Physiol Lung Cell Mol Physiol* 318: L376–L385). However, there is weight loss due to lung injury in the mice receiving intratracheal bleomycin.

Note: Focusing directly on weight loss as the single predictor of murine lung health misses the point of the model. Patients diagnosed with IPF will have different trajectories related to their lung function decline and the same phenotypic features are observed in the bleomycin model. If reversion of bleomycin induced lung damage was seen following fibrotic induction without treatment, then the reviewer would be justified in her/his concerns. In addition, it can also be argued that the mice are all on a similar weight loss trajectory (from Day 12 to Day 15), with all bleomycin treated groups (Fig. 2b) losing approximately 5% bodyweight during this time. Should the writing in the Result section be modified to reflect the above?

Comment: Why were no hydroxyproline experiments done for the prophylactic experiments comparing TH5487 with Deaxmethasone?

Please see Figure 4g.

Comment: Finally I do take expectation to the final statement in the introduction "Final statement "Taken together, 72 these data strongly suggest that TH5487 is a potent, specific, and clinically-relevant treatment for 73 IPF." This is very much overselling their data, at best, if the data are replicated and the concerns answered this would be a reason to investigate further but there is nothing in these data that suggest it is 'clinically relevant for IPF'.

The statement has been amended to include the possibility of clinical relevance following further studies (L47-49). 'Taken together, these data strongly suggest that TH5487 mitigates numerous features of IPF progression and should be tested further in clinical trials.' Additionally, the final statement has also been amended to read (L328-329) 'These data show pre-clinical proof-of-concept of TH5487 and motivate for its use in clinical trials for IPF.'

REFERENCES

Ba, X. and I. Boldogh (2018). "8-Oxoguanine DNA glycosylase 1: Beyond repair of the oxidatively modified base lesions." *Redox Biol* **14**: 669-678.

Feng, X. H. and R. Derynck (2005). "Specificity and versatility in tgf-beta signaling through Smads." Annu Rev Cell Dev Biol **21**: 659-693.

Fleming, A. M., Y. Ding and C. J. Burrows (2017). "Oxidative DNA damage is epigenetic by regulating gene transcription via base excision repair." Proc Natl Acad Sci U S A **114**(10): 2604-2609.

Hill, C. S. (2016). "Transcriptional Control by the SMADs." Cold Spring Harb Perspect Biol **8**(10).

Kim, B. N., D. H. Ahn, N. Kang, C. D. Yeo, Y. K. Kim, K. Y. Lee, T. J. Kim, S. H. Lee, M. S. Park, H. W. Yim, J. Y. Park, C. K. Park and S. J. Kim (2020). "TGF-beta induced EMT and stemness characteristics are associated with epigenetic regulation in lung cancer." Sci Rep **10**(1): 10597.

Lv, Z. D., B. Kong, J. G. Li, H. L. Qu, X. G. Wang, W. H. Cao, X. Y. Liu, Y. Wang, Z. C. Yang, H. M. Xu and H. B. Wang (2013). "Transforming growth factor-beta 1 enhances the invasiveness of breast cancer cells by inducing a Smad2-dependent epithelial-to-mesenchymal transition." Oncol Rep **29**(1): 219-225.

Miller, D. S. J., R. D. Bloxham, M. Jiang, I. Gori, R. E. Saunders, D. Das, P. Chakravarty, M. Howell and C. S. Hill (2018). "The Dynamics of TGF-beta Signaling Are Dictated by Receptor Trafficking via the ESCRT Machinery." Cell Rep **25**(7): 1841-1855 e1845.

Nicolas, F. J. and C. S. Hill (2003). "Attenuation of the TGF-beta-Smad signaling pathway in pancreatic tumor cells confers resistance to TGF-beta-induced growth arrest." Oncogene **22**(24): 3698-3711.

Ostrowska-Podhorodecka, Z., I. Ding, M. Norouzi and C. A. McCulloch (2022). "Impact of Vimentin on Regulation of Cell Signaling and Matrix Remodeling." Front Cell Dev Biol **10**: 869069.

Pan, L., H. Wang, J. Luo, J. Zeng, J. Pi, H. Liu, C. Liu, X. Ba, X. Qu, Y. Xiang, I. Boldogh and X. Qin (2019). "Epigenetic regulation of TIMP1 expression by 8-oxoguanine DNA glycosylase-1 binding to DNA:RNA hybrid." FASEB J **33**(12): 14159-14170.

Pezone, A., M. L. Taddei, A. Tramontano, J. Dolcini, F. L. Boffo, M. De Rosa, M. Parri, S. Stinziani, G. Comito, A. Porcellini, G. Raugei, D. Gackowski, E. Zarakowska, R. Olinski, A. Gabrielli, P. Chiarugi and E. V. Avvedimento (2020). "Targeted DNA oxidation by LSD1-SMAD2/3 primes TGF-beta1/ EMT genes for activation or repression." Nucleic Acids Res **48**(16): 8943-8958.

Ross, S. and C. S. Hill (2008). "How the Smads regulate transcription." Int J Biochem Cell Biol **40**(3): 383-408.

Visnes, T., A. Cazares-Korner, W. Hao, O. Wallner, G. Masuyer, O. Loseva, O. Mortusewicz, E. Wiita, A. Sarno, A. Manoilov, J. Astorga-Wells, A. S. Jemth, L. Pan, K. Sanjiv, S. Karsten, C. Gokturk, M. Grube, E. J. Homan, B. M. F. Hanna, C. B. J. Paulin, T. Pham, A. Rasti, U. W. Berglund, C. von Nicolai, C. Benitez-Buelga, T. Koolmeister, D. Ivanic, P. Iliev, M. Scobie, H. E. Krokan, P. Baranczewski, P. Artursson, M. Altun, A. J. Jensen, C. Kalderen, X. Ba, R. A. Zubarev, P. Stenmark, I. Boldogh and T. Helleday (2018). "Small-molecule inhibitor of OGG1 suppresses proinflammatory gene expression and inflammation." Science **362**(6416): 834-839.

Vizan, P., D. S. Miller, I. Gori, D. Das, B. Schmierer and C. S. Hill (2013). "Controlling long-term signaling: receptor dynamics determine attenuation and refractory behavior of the TGF-beta pathway." Sci Signal **6**(305): ra106.

Wang, Y., T. Chen, Z. Pan, Z. Lin, L. Yang, B. Zou, W. Yao, D. Feng, C. Huangfu, C. Lin, G. Wu, H. Ling and G. Liu (2020). "8-Oxoguanine DNA glycosylase modulates the cell transformation process in pulmonary fibrosis by inhibiting Smad2/3 and interacting with Smad7." *FASEB J* **34**(10): 13461-13473.

Wei, K., H. N. Nguyen and M. B. Brenner (2021). "Fibroblast pathology in inflammatory diseases." *J Clin Invest* **131**(20).

Wollin, L., E. Wex, A. Pautsch, G. Schnapp, K. E. Hostettler, S. Stowasser and M. Kolb (2015). "Mode of action of nintedanib in the treatment of idiopathic pulmonary fibrosis." *Eur Respir J* **45**(5): 1434-1445.

Reviewer #2 (Remarks to the Author):

The revised manuscript by Tanner et al represents a very substantial effort to address reviewers' concerns and recommendations, with the addition of significant new data, and clarification of the experimental rationale. As written, it will make a significant contribution to understanding the key regulatory role that OGG1 plays in modulating the magnitude of inflammatory cascades and will hopefully further bolster the development of clinically germane OGG1 inhibitors.

If accepted, as the manuscript proceeds toward a galley proof stage, the following improvements should be made:

Comment 1. abbreviations are sometimes repeated as reading through the manuscript

We appreciate Reviewer 2's careful reading/editing of the manuscript.

Reply: Abbreviations have been included in the Abstract and Introduction and duplications have been removed. The following corrections have been made:

(L209-210) '...the Mothers against decapentaplegic homolog (SMAD) family...' has been corrected to '...the SMAD family of proteins...'

Comment 2. the superscripting of genotypes is variable

Reply: We made appropriate changes as follows

L78- '...OGG1' changed to *Ogg1*^{-/-}

L106- '...OGG1' to *Ogg1*

L293- '...OGG1' to *Ogg1*

L322- '...OGG1' to *Ogg1*

The following nomenclature conventions have been used throughout the manuscript:

- Human/non-human primates
- Gene symbols are italicized, all letters are in upper case (e.g: *IGF1*)
- Proteins designations

- same as the gene symbol, but not italicized and (depending on species) all in upper case e.g: IGF1
- mRNA and cDNA use the gene symbol and formatting conventions e.g: "... levels of *IGF1* mRNA increased when..."
- Mouse
 - Gene symbols are italicized, first letter upper case all the rest lower case e.g: *Igf1*
 - Proteins designations
 - same as the gene symbol, but not italicized and all upper-case e.g: IGF1
 - mRNA and cDNA use the gene symbol and formatting conventions e.g: "... levels of *Igf1* mRNA increased when..."
 - Mutant alleles should be defined when first mentioned
 - All letters and numbers are italicized and the allelic designation (*tm1Arge*) is a superscript

Comment 3. there are typos in the current version

Reply: Thank you. We corrected these and other typos throughout the manuscript

L16- Excisions to excision

L20- well established to well-established

L35- decreases to decrease

L77- Comma removed 'and' added

Comment 4. the source of the TH5487 was not given - presumably it was from the Helleday lab, but if others wish to repeat or amplify this work, there should be a supplier listed for the compound.

Reply: Indeed, the compound was obtained from the Helleday Lab and a statement confirming this has now been included (L365). 'TH5487 utilized in these experiments was obtained from the Helleday Laboratory.'

Comment 5. the order of presentation of the subheadings in the Methods section should be organized with the order of presentation within the Results section - this would simply facilitate the reader to look at methodologies in a sequential order rather than scrolling through the entire methods looking for a method.

Reply: We thank the reviewer for this comment and have reordered the results and methods sections accordingly.

REVIEWER COMMENTS

Reviewer #1 (Remarks to the Author):

Once again the authors have gone to great lengths to address my concerns. Unfortunately my concerns remain. There are 2 key issues that I can not reconcile, as well as a disregard for the ARRIVE guidelines and once again over-interpreting their findings.

1) The authors show beautifully in the heavily cropped western blot in figure 1 of the response to reviewers that the pSMAD3 signal is at its plateau at 2ng/ml as I stated in my initial concerns. 2ng/ml is at the top of the concentration response for TGFbeta signalling as the western they provide shows, and this is higher than physiological concentrations in tissue samples. So why do all the effects they report on mRNA and wound healing require concentrations above 2ng/ml? Whatever the reason it is NOT due to TGFbeta signalling because as the authors point out this is no different between 2 and 10ng/ml.

2) The Hydroxyproline results in figure 2g are unimpressive. I recognise that TH5847 reduces the Hydroxyproline, by maybe 20%, compared with control and this is similar to nintedanib and pirfenidone in their hands. However, the miRNA for OGG1 has no meaningful effect in terms of effect size or statistical evaluation. I would want to see these data replicated in another model before proceeding to further preclinical evaluation.

3) A key feature of the ARRIVE guidelines is the performance of an a priori power calculation which would generate a single number for assessment of each intervention. The authors continue to provide a range of number of animals used for each intervention. This suggests a failure to understand the point of the ARRIVE guidelines.

4) The final sentence is still inappropriate and does not really make sense "These data show pre-clinical proof-of-concept of TH5487 and motivate for its use in clinical trials for IPF."

The use of TH5487 is a long way from going into a clinical trial in my opinion. I certainly would not support it based on the current data, at best I would suggest further preclinical evaluation.

REVIEWER COMMENTS

Reviewer #1 (Remarks to the Author):

Once again the authors have gone to great lengths to address my concerns. Unfortunately my concerns remain. There are 2 key issues that I cannot reconcile, as well as a disregard for the ARRIVE guidelines and once again over-interpreting their findings.

We thank the reviewer for acknowledging the work that has gone into answering these comments. We have attempted to answer these comments previously, through an extensive set of repeat experiments which can be seen in past reviews.

1) The authors show beautifully in the heavily cropped western blot in figure 1 of the response to reviewers that the pSMAD3 signal is at its plateau at 2ng/ml as I stated in my initial concerns. 2ng/ml is at the top of the concentration response for TGFbeta signalling as the western they provide shows, and this is higher than physiological concentrations in tissue samples. So why do all the effects they report on mRNA and wound healing require concentrations above 2ng/ml? Whatever the reason it is NOT due to TGFbeta signalling because as the authors point out this is no different between 2 and 10ng/ml.

It must be kept in mind that the assays utilized are *in vitro* and represent preclinical predictions of isolated cell responses (migration, cell transformation, protein/mRNA expression). In this regard, it must be stated that the main goal of the manuscript is not to reduce TGF- β 1 signaling. TGF- β 1 has been utilized as a stimulatory agonist of the profibrotic cellular responses in these assays and has been shown in our own work (using an array of concentrations) and the work of others to range from 10 ng/mL to lower concentrations. The reviewer has taken an exceptionally strong stance on 2 ng/mL being the upper limit of TGF- β 1 addition and we must respect this. We have therefore removed the assays that do not show a response to 2 ng/mL, including Fig. 1b and c, with Fig. 1 revised in the main text.

2) The Hydroxyproline results in figure 2g are unimpressive. I recognise that TH5487 reduces the Hydroxyproline, by maybe 20%, compared with control and this is similar to nintedanib and pirfenidone in their hands. However, the miRNA for OGG1 has no meaningful effect in terms of effect size or statistical evaluation. I would want to see these data replicated in another model before proceeding to further preclinical evaluation.

The reviewer is correct that the results for the TH5487 treatment group in the hydroxyproline assay are comparable to those in the nintedanib and pirfenidone groups, with the siRNA groups displaying less of an effect (Fig. 2g). However, as we state in the manuscript, the goal of the siRNA knockdown in the lung was to demonstrate that the targeting of *Ogg1* in the lung would reduce fibrotic lung damage. This is evidently proven by the Western blot directly above this in Figure 2f, with TH5487 reducing the expression of OGG1 in murine lung tissue to a greater extent than the addition of siRNA. We have therefore shown that the targeting of *Ogg1* reduces the profibrotic effects in this model. To further support this, the data in Figure 4 (containing more mice, which are dosed with TH5487 from day 0), show a similarly comparable decrease in hydroxyproline content to the nintedanib and pirfenidone groups, reducing the hydroxyproline levels by approximately 80%. Again, this is not the only metric used

for determining fibrotic lung damage in these experiments and we have highlighted further benefits to utilizing TH5487.

3) A key feature of the ARRIVE guidelines is the performance of an *a priori* power calculation which would generate a single number for assessment of each intervention. The authors continue to provide a range of number of animals used for each intervention. This suggests a failure to understand the point of the ARRIVE guidelines.

We thank the reviewer for this comment. We must state that a very detailed reply was made in the first review and has been restated below. In addition, we have included an extensive commentary on the ARRIVE guidelines using the updated version of the document (du Sert et al. 2020).

The power calculations were determined *a priori* as follows (added to the reporting summary), which we believe to be statistically sound. From an initial pilot study involving 3 groups of mice ($N=5$ per group) we determined an initial effect size (f) of 0.6 allowing us to perform formal power and sample size calculations. Using the freely downloadable software G Power (Faul, Erdfelder, Lang and Buchner, 2007 <https://pubmed.ncbi.nlm.nih.gov/17695343/>), we calculated an *a priori* sample size based on the following values: $\alpha=0.05$; Power=0.9; number of groups=5. Based on the pilot study data, an ANOVA with fixed effects, omnibus, one-way was used to calculate the ideal total sample size value of 50 ($N=10$

per group).

However, the study was designed with the knowledge that the administration of bleomycin causes mortality in 10-20% of the mice before the crucial onset of fibrosis. With this knowledge, the corrected group sizes for the bleomycin administered groups would be calculated at a minimum of $N=12$. To account for unexpected murine death (due to intraperitoneal injection) we allowed for $N=13$ mice per bleomycin treated group. The remaining two groups, namely vehicle and TH5487 (TH)-only were decided upon considering the 3 R's in the ARRIVE guidelines for animal use. It was decided that with no changes expected within the vehicle only group, based on our preliminary studies, a smaller group size could be used. Finally, the TH-only group included the recommended number of animals ($N=10$) to

achieve statistical power. This approach was sufficient to detect significant biological differences and offer translational value from this set of experiments. This is mentioned on L338-340 in the revised manuscript.

Furthermore, we have addressed each point of the ARRIVE guidelines in the table below and included this as supplementary material (Supp. Table 2).

Table 1. ARRIVE guidelines in the context of the current data set

ARRIVE guidelines		
Guideline	Criteria	Data in this manuscript
1. Study design	For each experiment, provide brief details of study design including: a. The groups being compared, including control groups. If no control group has been used, the rationale should be stated. b. The experimental unit (e.g. a single animal, litter, or cage of animals).	Data were compared to the bleomycin control group. Control groups were also included for vehicle administration, TH5487 only administration, anti-inflammatory response (Dexamethasone/bleomycin), and clinically-utilized drugs (pirfenidone and nintedanib). The experimental unit in this case is the treatment group of mice included for each intervention.
2. Sample size	a. Specify the exact number of experimental units allocated to each group, and the total number in each experiment. Also indicate the total number of animals used. b. Explain how the sample size was decided. Provide details of any a priori sample size calculation, if done.	a. The following numbers of animals were allocated to each treatment and prophylactic group respectively, with the numbers representative of data included in the manuscript.  -Bleomycin (n=5 and n=13) -Bleomycin/Ogg1 siRNA (n=5; n=0) -PBS/Ogg1 siRNA (n=4; n=0) -Vehicle (n=4; n=8) -Bleomycin/TH5487 (n=8; n=13) -Bleomycin/Pirfenidone (n=8; n=8) -Bleomycin/Nintedanib (n=8; n=8) -Bleomycin/Dexamethasone (n=0; n=9) The total animals used amounted to n=148 and were allocated accordingly:  -Bleomycin (n=13 and n=18) -Bleomycin/Ogg1 siRNA (n=7; n=0) -PBS/Ogg1 siRNA (n=4; n=0)

		-Vehicle (n=4; n=8) -PBS/TH5487 (n=0; n=8) -Bleomycin/TH5487 (n=10; n=15) -Bleomycin/Pirfenidone (n=10; n=13) -Bleomycin/Nintedanib (n=10; n=13) -Bleomycin/Dexamethasone (n=0; n=15) b. The power calculations were determined a priori as follows (added to the reporting summary), which we believe to be statistically sound. From an initial pilot study involving 3 groups of mice ($N=5$ per group) we determined an initial effect size (f) of 0.6 allowing us to perform formal power and sample size calculations. Using the freely downloadable software G Power (Faul, Erdfelder, Lang and Buchner, 2007 https://pubmed.ncbi.nlm.nih.gov/17695343/), we calculated an a priori sample size based on the following values: $\alpha=0.05$; Power=0.9; number of groups=5. Based on the pilot study data, an ANOVA with fixed effects, omnibus, one-way was used to calculate the ideal total sample size value of 50 ($N=10$). However, the study was designed with the knowledge that the administration of bleomycin causes mortality in 10-20% of the mice before the crucial onset of fibrosis. With this knowledge, the corrected group sizes for the bleomycin administered groups would be calculated at a minimum of $N=12$. To account for unexpected murine death (due to intraperitoneal injection) we allowed for $N=13$ mice per bleomycin treated group. The remaining group, namely vehicle only was decided upon considering the 3 R's in the ARRIVE guidelines for animal use. It was decided that with no changes expected within the vehicle only group, based on our preliminary studies, a smaller group size could be used. This approach was sufficient to detect significant biological differences and offer translational value from this set of experiments.
3. Inclusion and exclusion criteria	a. Describe any criteria used for including and excluding animals (or experimental units) during the experiment, and data points during the analysis.	a. No data were excluded from the analysis but animals that did not survive to Day 15 of the model were not processed for further readouts. -Bleomycin (n=5 and n=13) -Bleomycin/Ogg1 siRNA (n=5; n=0) -PBS/Ogg1 siRNA (n=4; n=0) -Vehicle (n=4; n=8) PBS/TH5487 (n=0; n=8)

	Specify if these criteria were established a priori. If no criteria were set, state this explicitly. b. For each experimental group, report any animals, experimental units or data points not included in the analysis and explain why. If there were no exclusions, state so. c. For each analysis, report the exact value of n in each experimental group.	-Bleomycin/TH5487 (n=8; n=13) -Bleomycin/Pirfenidone (n=8; n=8) -Bleomycin/Nintedanib (n=8; n=8) -Bleomycin/Dexamethasone (n=0; n=9)
4. Randomization	a. State whether randomisation was used to allocate experimental units to control and treatment groups. If done, provide the method used to generate the randomisation sequence. b. Describe the strategy used to minimise potential confounders such as the order of treatments and measurements, or animal/cage location. If confounders were not controlled, state this explicitly.	a. Mice were randomly assigned to each group using a simple randomization approach. b. Confounders were reduced by randomizing treatment administration. Mice were also euthanized in groups based on a randomized order. The Excel function =INT(RAND()*group number) was used to calculate which groups would be treated or euthanized.
5. Blinding	Describe who was aware of the group allocation at the different stages of	Two of the investigators (JB/LT) were aware of which animals needed to be dosed with TH5487/bleomycin and as such were not truly blinded to the outcome of the study. However,

	the experiment (during the allocation, the conduct of the experiment, the outcome assessment, and the data analysis).	investigators not involved in the animal experimentation (RB) were blinded to the outcome and conducted downstream analyses in an unbiased manner. Therefore, we have not biased the study in terms of any of the results, but it is not correct to state that complete blinding was used.
6. Outcome measures	a. Clearly define all outcome measures assessed (e.g. cell death, molecular markers, or behavioural changes). b. For hypothesis-testing studies, specify the primary outcome measure, i.e. the outcome measure that was used to determine the sample size	a. Outcome measures in this study were mainly derived post-euthanization, with the exception of murine weight. End point-measurements included but were not limited to, lung/BALF inflammatory cell recruitment, cytokine production, collagen deposition, profibrotic gene regulation, profibrotic protein production, and histological damage. b. During the sample size calculation a pilot study was conducted and hydroxyproline measurements were used as the outcome measure for comparison.
7. Statistical methods	a. Provide details of the statistical methods used for each analysis, including software used. b. Describe any methods used to assess whether the data met the assumptions of the statistical approach, and what was done if the assumptions were not met.	In this study, groups of three or more mice were compared using one-way analysis of variance (ANOVA) with Dunnett's post hoc test. In experiments using two groups, results were compared using unpaired t test with Welch's correction. Results in this study are displayed throughout as mean ± SEM. Statistical testing was carried out using GraphPad Prism 9.1.1 (San Diego, USA) with statistical significance defined as $P < 0.05$.
8. Experimental animals	a. Provide species-appropriate details of the animals used, including species, strain and substrain, sex, age or developmental stage, and, if relevant, weight. b. Provide further	10-12-week-old male C57Bl/6 mice (Janvier, Le Genest-Saint-Isle, France) were housed at least 2 weeks in the animal facility at the Biomedical Service Division at Lund University before initiating experiments and were provided with food and water ad libitum throughout the study.

	relevant information on the provenance of animals, health/immune status, genetic modification status, genotype, and any previous procedures.	
9. Experimental procedures	For each experimental group, including controls, describe the procedures in enough detail to allow others to replicate them, including: a. What was done, how it was done and what was used. b. When and how often. c. Where (including detail of any acclimatisation periods). d. Why (provide rationale for procedures).	10-12-week-old male C57Bl/6 mice (Janvier, Le Genest-Saint-Isle, France) were housed at least 2 weeks in the animal facility at the Biomedical Service Division at Lund University before initiating experiments and were provided with food and water ad libitum throughout the study. In addition, mice underwent the following procedures: -Bleomycin administration: Bleomycin (i.t.; 2.5 U/kg) or saline control was administered once using a microsyringe device under anaesthesia (isoflurane). -Drug/siRNA treatment: Compounds were administered at different time points in the two different models with mice in the treatment group receiving treatment from Day 15 onwards and mice in the prophylactic group receiving treatment 1h after bleomycin administration. The treatment for each group is outlined below as follows: -Bleomycin: Mice received bleomycin (2.5 U/kg; i.t.) via microsyringe. Mice were subsequently administered vehicle (150 uL i.p.; once daily) at day 15-21 or day 1-21. -Bleomycin/Ogg1 siRNA: Mice received bleomycin (2.5 U/kg; i.t.) via microsyringe. Mice were subsequently administered siRNA (25 µg i.t.; once) at day 15. PBS/Ogg1 siRNA: Mice received saline (50 uL; i.t.) via microsyringe. Mice were subsequently administered siRNA (50 µg i.t.; once) at day 15.

		-Vehicle: Mice received saline (50 uL; i.t.) via microsprayer. Mice were subsequently administered vehicle (150 uL i.p.; once daily) at day 15-21. -PBS/TH5487: Mice received saline (50 uL; i.t.) via microsprayer. Mice were subsequently administered TH5487 (40 mg/kg i.p.; once daily) at day 15-21. -Bleomycin/TH5487: Mice received bleomycin (2.5 U/kg; i.t.) via microsprayer. Mice were subsequently administered TH5487 (40 mg/kg i.p.; once daily) at day 15-21 or day 1-21. -Bleomycin/Pirfenidone: Mice received bleomycin (2.5 U/kg; i.t.) via microsprayer. Mice were subsequently administered pirfenidone (300 mg/kg p.o.; once daily) at day 15-21 or day 1-21. -Bleomycin/Nintedanib: Mice received bleomycin (2.5 U/kg; i.t.) via microsprayer. Mice were subsequently administered nintedanib (60 mg/kg p.o.; once daily) at day 15-21 or day 1-21. -Bleomycin/Dexamethasone: Mice received bleomycin (2.5 U/kg; i.t.) via microsprayer. Mice were subsequently administered dexamethasone (10 mg/kg i.p.; once daily) at day 15-21 or day 1-21.
10. Results	For each experiment conducted, including independent replications, report: a. Summary/descriptive statistics for each experimental group, with a	 a. Each readout contains a description of the statistical method used (with appropriate P value, with results reported as means and SD's. b. Effect sizes have not been provided

	measure of variability where applicable (e.g. mean and SD, or median and range). b. If applicable, the effect size with a confidence interval.	
11. Abstract	Provide an accurate summary of the research objectives, animal species, strain and sex, key methods, principal findings, and study conclusions	All relevant experimental details are provided in the abstract, with key methods, findings and conclusions all highlighted.
12. Background	a. Include sufficient scientific background to understand the rationale and context for the study, and explain the experimental approach. b. Explain how the animal species and model used address the scientific objectives and, where appropriate, the relevance to human biology	a. The background to the study is supportive and explains the context of the research. b. The reasons for using the bleomycin model have been highlighted 'This model reproduces several phenotypic features of human IPF, including peripheral alveolar septal thickening, dysregulated cytokine production, and immune cell influx resulting in fibrosis.'
13. Objectives	Clearly describe the research question, research objectives and, where appropriate, specific hypotheses being tested.	The goal of this study was to test a novel pharmaceutical approach to inhibit OGG1, ultimately leading to the inhibition of fibrosis-related progression. Initial in vitro experiments in bronchial epithelial and fibroblast cells displayed decreased fibrosis-related phenotypic features. Subsequent in vivo murine studies using intratracheally-administered bleomycin were chosen as well-established and relevant models of experimental lung fibrosis.
14. Ethical statement	Provide the name of the ethical	All animal experiments were approved by the Malmö-Lund Animal Care Ethics Committee

	review committee or equivalent that has approved the use of animals in this study, and any relevant licence or protocol numbers (if applicable). If ethical approval was not sought or granted, provide a justification.	(M17009-18). Human lung tissue was obtained after written informed consent, approval by the Regional Ethical Review Board in Lund (approval no. LU412-03) and performed in accordance with the Declaration of Helsinki as well as relevant guidelines and regulations.
15. Housing and husbandry	Provide details of housing and husbandry conditions, including any environmental enrichment.	10-12-week-old male C57Bl/6 mice (Janvier, Le Genest-Saint-Isle, France) were housed at least 2 weeks in the animal facility at the Biomedical Service Division at Lund University before initiating experiments and were provided with food and water ad libitum throughout the study. Mice were provided with environmental enrichment which included cardboard tubes, chew sticks, and shredded paper.
16. Animal care and monitoring	a. Describe any interventions or steps taken in the experimental protocols to reduce pain, suffering and distress. b. Report any expected or unexpected adverse events. c. Describe the humane endpoints established for the study, the signs that were monitored and the frequency of monitoring. If the study did not have humane endpoints, state this.	Mice were anaesthetized during all invasive procedures (i.t. administration). In addition, the mice were euthanized as soon as the weight crossed the 20% cutoff mark or if the mice displayed any signs of distress as listed in our ethical permit. The general humane endpoint is indicated by many signs including a loss of 20% of original mouse weight and changes in mouse behavior including (amended in revised manuscript):  - discomfort or stress: (e.g. dull/staring coat (hair erect); decreased or increased activity; avoidance behaviour; isolation from the group; depressed; decreased appetite) - deterioration (e.g. Discharge from eyes; loss of general condition; anorexia; dehydration (tenting skin/sunken eyes); weakness; decreased motility) - distress: (e.g. very weak; unresponsive to touch; unconscious; convulsing; difficulty breathing) - If obvious distress (bleeding trachea, labored breathing etc.) following intratracheal administration of compound/bleomycin is seen, the mice will be immediately euthanized.

		 - Mice that experience severe adverse reactions to the test compound or anaesthetic drug and are deemed to be in distress will be euthanised. Animals showing neurological signs, that have convulsions, demonstrate severe weakness, avoidance behaviour, display signs of depression, show respiratory distress, or open-mouthed breathing will humanely be euthanized prior to the end point.
17. Interpretation	a. Interpret the results, taking into account the study objectives and hypotheses, current theory and other relevant studies in the literature. b. Comment on the study limitations including potential sources of bias, limitations of the animal model, and imprecision associated with the results.	Together, our findings demonstrate that TH5487 possesses a mechanism of action targeting OGG1 to suppress IPF, which is distinct from currently employed therapeutic interventions. This study further elucidates the downstream effects of this approach, decreasing myofibroblast transition, fibroblast migration, inflammatory cell recruitment, and eventual inhibition of fibrotic-related lung remodelling. These data show promising therapeutic effects of TH5487 in a mouse model of IPF, motivating further pre-clinical evaluation. Important limitations addressed in this study include, whether treatment using TH5487 can display utility in human IPF pathologies. Whilst the translational aspect of the study is suggested using human lung sections, it is important to demonstrate that TH5487 treatment successfully decreased OGG1 levels and subsequent IPF lung damage in human clinical trials. Additional limitations in this study include the lack of monitoring of potential off-target effects induced by targeting OGG1. Whilst no obvious reductions in key murine health status measures were observed in this or other studies, any small-molecule utilization should be accompanied by long term monitoring of adverse effects to ensure safe therapeutic usage. Furthermore, Ogg1^{-/-} mice display no deleterious pathological changes, suggesting specific Ogg1 targeting may be safe. In any case, further efforts are required before progressing this compound to clinical trials.
18. Generalisability/translation	Comment on whether, and how, the findings of this study are likely to generalise to other	Our data demonstrate targeting Ogg1 suppresses TGF-β1 and several key immune modulatory cytokines and chemokines in murine BALF, lungs, and plasma. Treatment with both TH5487 or Ogg1 siRNA resulted in decreased profibrotic cytokine

	species or experimental conditions, including any relevance to human biology (where appropriate).	expression and diminished immune cell recruitment to the lung. These findings are relevant for numerous diseases with an inflammatory related description.
19. Protocol registration	Provide a statement indicating whether a protocol (including the research question, key design features, and analysis plan) was prepared before the study, and if and where this protocol was registered.	The study plan was included as part of several grant applications which have been listed in the manuscript. The work was supported by grants from: Swedish Research Council 2020-011166 (AE) The Swedish Heart and Lung Foundation 20190160 (AE) The Swedish Government Funds for Clinical Research 46402 (ALF; AE) The Alfred Österlund Foundation (AE) Vinnova Swelife 2, 2018-03232 (AE, TH, CK) Horizon 2020 ERC-PoC (TH) US NIH, National Institute of Allergy and Infectious Diseases, AI062885 (IB)
20. Data access	Provide a statement describing if and where study data are available.	Mass spectrometry proteomics data and initial search results have been deposited to the ProteomeXchange Consortium (http://proteomecentral.proteomexchange.org) via the PRIDE partner repository [1] with the dataset identifier PXD029625 (Username: reviewer_pxd029625@ebi.ac.uk ; Password: 5YIGmBWu; will be made public upon acceptance of the manuscript). Downstream analysis R code is available at https://github.com/heuselM/DiffTestR/tree/Tanner2021 . All remaining data are available in the main text or the supplementary materials.
21. Declaration of interests	a. Declare any potential conflicts of interest, including financial and non-financial. If none exist, this should be stated. b. List all funding sources (including grant identifier) and the role of the funder(s) in the design, analysis	T.H. is listed as inventor on a provisional U.S. patent application no. 62/636983, covering OGG1 inhibitors. The patent is fully owned by a nonprofit public foundation, the Helleday Foundation, and T.H. is a member of the foundation board developing OGG1 inhibitors toward the clinic. An inventor reward scheme is under discussion. The remaining authors declare no competing financial interests. The work was supported by grants from: Swedish Research Council 2020-011166 (AE)

	and reporting of the study	The Swedish Heart and Lung Foundation 20190160 (AE) The Swedish Government Funds for Clinical Research 46402 (ALF; AE) The Alfred Österlund Foundation (AE) Vinnova Swelife 2, 2018-03232 (AE, TH, CK) Horizon 2020 ERC-PoC (TH) US NIH, National Institute of Allergy and Infectious Diseases, AI062885 (IB) Author contributions: Conceptualization: AE, LT, ABS, CK, JB, IB Methodology: LT, ABS, RKVB, MH, TM, CAQK, JM, RMO, CC, CKA, JSE, OW, TH, JB, IB MS data management & analysis: MH Investigation: LT, ABS, RKVB, MH, TM, CAQK, RMO, CC, CKA, JSE, JB, IB Funding acquisition: AE, CK, TH Project administration: AE, CK, TH Supervision: AE, CK, TH Writing – original draft: LT Writing – review & editing: LT, ABS, AE, CK, MH, JB, IB
--	----------------------------	--

4) The final sentence is still inappropriate and does not really make sense "These data show pre-clinical proof-of-concept of TH5487 and motivate for its use in clinical trials for IPF." The use of TH5487 is a long way from going into a clinical trial in my opinion. I certainly would not support it based on the current data, at best I would suggest further preclinical evaluation.

We agree with the reviewer and have changed the final sentence as follows (L330). "These data show promising therapeutic effects of TH5487 in a mouse model of pulmonary fibrosis and provide motivation for OGG1 as a tractable drug target for IPF."

REVIEWERS' COMMENTS

Reviewer #5 (Remarks to the Author):

This is a comprehensive preclinical study of OGG1 inhibition in an established mouse model of pulmonary fibrosis using both siRNA knockdown as well as small molecule approaches. The data is novel and of potential clinical reference and the authors' have adequately responded to the remaining reviewers comments.

Reviewer #5 (Remarks to the Author):

This is a comprehensive preclinical study of OGG1 inhibition in an established mouse model of pulmonary fibrosis using both siRNA knockdown as well as small molecule approaches. The data is novel and of potential clinical reference and the authors' have adequately responded to the remaining reviewers comments.

We thank the reviewer for this response and appreciate the time taken to assess our manuscript.